# Scalable Exploration via Ensemble++

**Yingru Li**[1,2*†]   **Jiawei Xu**[1*]   **Baoxiang Wang**[1]   **Zhi-Quan Luo**[1,2]
[1]The Chinese University of Hong Kong, Shenzhen
[2]Shenzhen Research Institute of Big Data

## Abstract

Thompson Sampling is a principled method for balancing exploration and exploitation, but its real-world adoption faces computational challenges in large-scale or non-conjugate settings. While ensemble-based approaches offer partial remedies, they typically require prohibitively large ensemble sizes. We propose Ensemble++, a scalable exploration framework using a novel shared-factor ensemble architecture with random linear combinations. For linear bandits, we provide theoretical guarantees showing that Ensemble++ achieves regret comparable to exact Thompson Sampling with only $\Theta(d \log T)$ ensemble sizes–significantly outperforming prior methods. Crucially, this efficiency holds across both compact and finite action sets with either time-invariant or time-varying contexts without configuration changes. We extend this theoretical foundation to nonlinear rewards by replacing fixed features with learnable neural representations while preserving the same incremental update principle, effectively bridging theory and practice for real-world tasks. Comprehensive experiments across linear, quadratic, neural, and GPT-based contextual bandits validate our theoretical findings and demonstrate Ensemble++'s superior regret-computation tradeoff versus state-of-the-art methods.

## 1 Introduction

Balancing *exploration* and *exploitation* is a core challenge in sequential decision-making problems, with applications ranging from online recommendation systems, automated content moderation to robotics, personalized healthcare and computer-using agents. A prominent Bayesian solution is *Thompson Sampling* (TS) [Thompson, 1933, Russo et al., 2018], which maintains a posterior over unknown parameters (or reward functions). At each step, it samples a model hypothesis from this posterior and selects the action appearing optimal under that hypothesis, elegantly balancing exploration of uncertain actions and exploitation of seemingly high-reward ones. Despite its elegant theory and strong empirical performance in simpler (conjugate) bandit scenarios, TS encounters serious *scalability* hurdles in modern settings with high-dimensional or non-conjugate (e.g., neural) models. Maintaining exact posterior samples can be computationally prohibitive, often requiring iterative approximation methods (Laplace, MCMC, or variational inference) that become expensive as the time horizon $T$ grows.

**Ensemble-Based Approximate Sampling.**   A widely adopted alternative to full Bayesian updates is *ensemble sampling*, which keeps $M$ model replicas in parallel and randomly picks one each round to act, thus approximating Thompson Sampling's "draw from the posterior" step [Osband and Van Roy, 2015, Osband et al., 2016, 2019]. However, prior theoretical results matching TS's optimal regret in linear bandits, such as Qin et al. [2022], require an ensemble size $M = \Omega(T \cdot |\mathcal{X}|)$, where $\mathcal{X}$ is the action space. This large-$M$ requirement is often infeasible in high-dimensional or long-horizon tasks. Moreover, many ensembles demand either repeated retraining or large architectural overhead, raising practical concerns in real-time or resource-constrained environments.

---

[*]Equal contribution. Code: `https://github.com/szrlee/Ensemble_Plus_Plus`
[†]Corresponding author: `szrlee@gmail.com`

39th Conference on Neural Information Processing Systems (NeurIPS 2025).

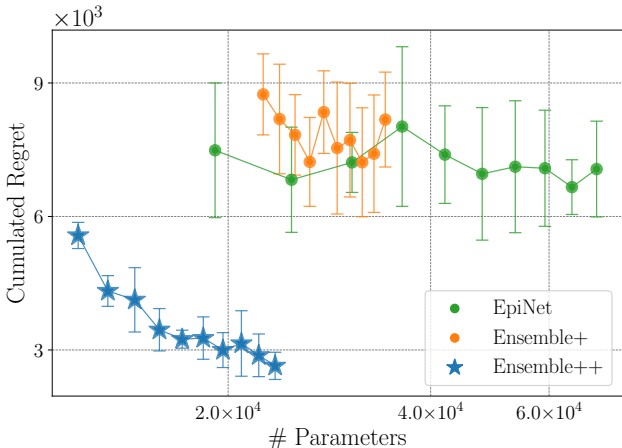

Figure 1: Preview of the regret-computation trade-off in a *nonlinear* bandit (unknown quadratic reward). The x-axis is the number of model parameters, serving as a proxy for computational cost. *Ensemble++* outperforms state-of-the-art baselines Ensemble+ and EpiNet. See details in Section 5.

Our approach to this challenge follows a principled strategy: we propose *Ensemble++*, a general framework using incremental updates suitable for complex models with non-conjugate posteriors. To rigorously analyze its core sampling mechanism and establish theoretical guarantees (particularly regarding ensemble size), we first analyze its instantiation in the tractable linear setting. This analysis provides foundational evidence for the mechanism's efficiency. We then demonstrate that the general method, using incremental SGD updates and neural network, achieves strong performance in the challenging non-conjugate settings it was designed for—thereby bridging theoretical understanding with practical applicability.

## 1.1 Key Contributions

In this paper, we propose *Ensemble++* for scalable approximate Thompson Sampling that obviates the need for large ensemble sizes or costly per-step retraining:

- *Novel Approximation Mechanism*: We introduce Ensemble++, which maintains a single shared ensemble matrix factor incrementally updated via a random linear combination scheme. This fundamental innovation enables efficient approximate posterior sampling without large ensembles or costly per-step retraining.
- *Theoretical Breakthrough in Linear Bandits*: We prove that *Linear Ensemble++ Sampling*, with an ensemble size of only $M = \Theta(d \log T)$, matches the regret order of exact Thompson Sampling. This is a foundational result for our broader approach. Specifically, it achieves regret of $\mathcal{O}(d^{3/2}\sqrt{T}(\log T)^{3/2})$ for compact action sets and $\mathcal{O}(d\sqrt{T \log |\mathcal{X}|} \log T)$ for finite action sets. Crucially, the *same algorithm* applies to both compact and finite action sets, whether in invariant or varying contexts, without changing algorithmic configurations. This dramatically improves on prior analyses requiring, for example, $M = \Omega(T \cdot |\mathcal{X}|)$ [Qin et al., 2022].
- *Neural Extension*: We extend these ideas by incorporating neural networks, replacing fixed linear features with a learnable neural feature extractor while keeping the same incremental-update principle. This yields a flexible approach for complex, high-dimensional reward functions.
- *Empirical Validation*: Through comprehensive experiments on synthetic and real-world benchmarks—including *quadratic* bandits and large-scale neural tasks involving GPTs—we demonstrate that Ensemble++ achieves superior regret-*vs*-computation trade-offs compared to leading baselines such as Ensemble+ [Osband et al., 2018, 2019] and EpiNet [Osband et al., 2023a,b] (see Fig. 1 and Section 5); and validate the theoretical results of linear Ensemble++ sampling.

This work both closes a longstanding theoretical gap in linear ensemble sampling and provides a flexible framework for deeper models. We describe linear Ensemble++ sampling and neural extension in Section 3, provide full theoretical analysis of our linear scheme in Section 4, and present empirical evaluations in Section 5, building on the foundational concepts in Section 2.

## 2 Background and Related Work

### 2.1 Sequential Decision Making under Uncertainty

We consider a sequential decision-making problem over a time horizon $T$. At each time step $t$:

- The agent observes a *decision set* $\mathcal{X}_t \subseteq \mathcal{X}$, which may change over time (e.g., due to evolving *context* or the appearance of new candidate actions).
- The agent selects action $X_t \in \mathcal{X}_t$, based on history $\mathcal{H}_t = \{\mathcal{X}_1, X_1, Y_1, \ldots, \mathcal{X}_{t-1}, X_{t-1}, Y_{t-1}, \mathcal{X}_t\}$.
- It then receives a noisy reward $Y_t = f^*(X_t) + \epsilon_t$, under unknown reward function $f^*$ and noise $\epsilon_t$.

The agent's *cumulative regret* measures how much reward is lost by not picking the best action:

$$R(T) \;=\; \sum_{t=1}^{T} \Big[ \max_{x \in \mathcal{X}_t} f^*(x) \;-\; f^*(X_t) \Big]. \tag{1}$$

A key challenge is *learning* unknown reward function $f^*$ (exploration) while simultaneously *selecting* good actions from $\mathcal{X}_t$ (exploitation) in $T$ time periods.

### 2.2 Thompson Sampling and the Scalability Dilemma

**Principled exploration with Thompson Sampling.** A popular Bayesian approach to sequential decision-making is *Thompson Sampling (TS)* [Thompson, 1933, Russo et al., 2018]. Given a posterior distribution over an unknown reward function $f^*$ (or unknown parameters $\theta^*$), Thompson Sampling operates as follows at each time $t$: (1) *Sample* a hypothesis $\theta_t$ from the current posterior, (2) *Select* the action $X_t = \arg\max_{x \in \mathcal{X}_t} f_{\theta_t}(x)$ under that hypothesis, (3) *Observe* the reward $Y_t$, and (4) *Update* the posterior distribution given $(X_t, Y_t)$. By sampling from its posterior, TS naturally balances exploration and exploitation.

**The scalability dilemma.** While elegant, TS faces scalability hurdles. When an environment model belongs to a *conjugate* family (e.g., linear–Gaussian), posterior updates are tractable. However, in high-dimensional or *non-conjugate* cases (e.g., neural nets), *exact* Bayesian updates become expansive or intractable [Russo et al., 2018]. *Approximate* methods (Laplace, MCMC, variational inference) can introduce large computational overheads and/or biased uncertainty estimates [MacKay, 1992, Welling and Teh, 2011, Blei et al., 2017, Xu et al., 2022]. These challenge motivate scalable approximate posterior approaches.

### 2.3 Approximation Techniques for Thompson Sampling

**Local perturbation.** For *linear–Gaussian* bandits, *local perturbation* [Papandreou and Yuille, 2010] offers an $\mathcal{O}(d^2)$ per-step update to emulate posterior samples. It incrementally maintains $\tilde{A}_t = \Sigma_t(\Sigma_{t-1}^{-1}\tilde{A}_{t-1} + X_t Z_t)$, where $\tilde{A}_0 \sim \mathcal{N}(0, \Sigma_0)$ and $Z_s \sim \mathcal{N}(0,1)$. If actions $\{X_s\}$ were non-adaptive, $\mu_t + \tilde{A}_t$ would be an exact posterior draw. However, adaptive action selection introduces *sequential dependencies*, biasing these draws as $\tilde{A}_t \mid \{X_s\}_{s \leq t}$ no longer matches $\mathcal{N}(0, \Sigma_t)$. Resampling all perturbations $\{Z_s\}$ from scratch at each step resolves this but is computationally prohibitive [Osband et al., 2019, Kveton et al., 2020a].

**Ensemble sampling.** *Ensemble sampling* [Osband and Van Roy, 2015, Osband et al., 2016, Lu and Van Roy, 2017] is a widely used alternative, maintaining $M$ models and randomly selecting one per round to act. While empirically effective with moderate $M$, theoretical guarantees often lag. For instance, Qin et al. [2022] showed that matching exact TS's $\sqrt{T}$-type regret in linear bandits requires $M = \Omega(T \cdot |\mathcal{X}|)$. Recent work by Lee and hwan Oh [2024] also explores ensemble methods for linear bandits, contributing to the understanding of ensemble size requirements under different analytical approaches. A key open question has been whether TS-comparable regret can be achieved with a practically small ensemble size, especially one that does not scale with $T$ or $|\mathcal{X}|$.

**Approaches for Non-Linear Models.** In non-linear or neural settings, methods like Ensemble+ [Osband et al., 2018, 2019] and EpiNet [Osband et al., 2023a,b] adapt ensemble ideas, often by training members on perturbed data or using architectural modifications to inject uncertainty. While empirically successful, these methods typically rely on large ensembles or lack the rigorous ensemble size and regret guarantees established in linear settings. The challenge remains to develop methods that

are both theoretically grounded in simpler settings and practically scalable to complex, non-conjugate domains. Our work, Ensemble++, is designed to address this gap.

## 3 Ensemble++ for Scalable Thompson Sampling

We now introduce *Ensemble++*, a *unified* and *scalable* approach to approximate Thompson Sampling in both *linear* and *nonlinear* bandit environments. The key technical novelty is to maintain a *shared ensemble factor* incrementally, thereby approximating posterior covariance (in the linear case) or capturing epistemic uncertainty (in the neural case) without requiring a large ensemble size or repeated retraining from scratch. We begin with *Linear Ensemble++ Sampling* (Section 3.1), describing its incremental matrix-factor updates and explaining how it approximates Thompson Sampling with only $M \approx d \log T$ ensemble directions. We then extend these ideas to general *Ensemble++* (Section 3.3), using the same *symmetrized regression* principle (Section 3.2) but replacing linear features with a trainable neural representation.

### 3.1 Linear Ensemble++ Sampling

Consider a *linear contextual bandit*, where each action $X_t \in \mathbb{R}^d$ and reward $Y_t = \langle \theta^*, X_t \rangle + \epsilon_t, \epsilon_t \sim \mathcal{N}(0, 1)$. Let $(\mu_t, \Sigma_t)$ be the usual ridge-regression posterior updates:

$$\Sigma_t^{-1} = \Sigma_{t-1}^{-1} + X_t X_t^\top, \quad \mu_t = \Sigma_t \left( \Sigma_{t-1}^{-1} \mu_{t-1} + X_t Y_t \right). \tag{2}$$

The $\mathcal{O}(d^3)$ cost for standard TS comes from matrix factorization for sampling. *Linear Ensemble++ Sampling* avoids this by maintaining a matrix $\mathbf{A}_t \in \mathbb{R}^{d \times M}$ that approximates $\Sigma_t^{1/2}$ incrementally:

**Initialization.** Construct $\mathbf{A}_0 = \frac{1}{\sqrt{M}} \left[ \tilde{A}_{0,1}, \ldots, \tilde{A}_{0,M} \right]$ with each ensemble $\tilde{A}_{0,m} \sim \mathcal{N}(0, \Sigma_0)$.

**Per-step procedure.** $(t = 1, \ldots, T)$:

1. *Action selection*: Sample a "reference" vector $\zeta_t \in \mathbb{R}^M$ from $P_\zeta$ (e.g., Gaussian; detailed in Appendices B.1 and E). Form

$$\theta_t(\zeta_t) = \mu_{t-1} + \mathbf{A}_{t-1} \zeta_t, \tag{3}$$

   via a random linear combination of the columns, then choose $X_t = \arg\max_{x \in \mathcal{X}_t} \langle x, \theta_t(\zeta_t) \rangle$.
2. Observe reward $Y_t$, sample a "perturbation" vector $\mathbf{z}_t \in \mathbb{R}^M$ from $P_\mathbf{z}$, and *update* $\mu_t$ via Eq. (2) and update ensemble matrix $\mathbf{A}_t = [\mathbf{A}_{t,1}, \ldots, \mathbf{A}_{t,M}]$ as following:

$$\mathbf{A}_{t,m} = \Sigma_t \left( \Sigma_{t-1}^{-1} \mathbf{A}_{t-1,m} + X_t \mathbf{z}_{t,m} \right), \forall m \quad \Leftrightarrow \quad \mathbf{A}_t = \Sigma_t \left( \Sigma_{t-1}^{-1} \mathbf{A}_{t-1} + X_t \mathbf{z}_t^\top \right), \tag{4}$$

**Approximate Posterior Sampling.** With $M = \Theta(d \log T)$, our analysis (see Section 4) shows that $\frac{1}{2} \Sigma_t \preccurlyeq \mathbf{A}_t \mathbf{A}_t^\top \preccurlyeq \frac{3}{2} \Sigma_t, \forall 0 \leq t \leq T$, with high probability. Hence, for $\zeta \sim \mathcal{N}(0, I_M)$, the random vector $\mu_t + \mathbf{A}_t \zeta$ serves as an *approximate* sample from $\mathcal{N}(\mu_t, \Sigma_t)$, enabling near-Thompson Sampling performance. The key insight of Linear Ensemble++ is that a relatively small number of properly updated ensemble directions can capture the essential uncertainty structure needed for effective exploration, without requiring full independence between ensemble members. By maintaining a shared factor incrementally, we avoid both large storage requirements and costly recomputation from scratch at each step. We emphasize that $P_\zeta$ and $P_\mathbf{z}$ can be chosen from distributions like Gaussian, uniform-on-sphere, coordinate or cube, each with different performance (Sections 4 and 5).

### 3.2 A Symmetrized Ridge-Regression View

An alternative perspective derives Ensemble++ from a ridge-regression objective. First, note that: (1) The *base* parameter $\mu_t$ solves the usual *ridge regression* objective $\min_b \sum_{s=1}^t \left[ Y_s - \langle b, X_s \rangle \right]^2 + \lambda \|b\|^2$. (2) Each column in $\mathbf{A}_t$ can be seen as a *perturbed* ridge solution that includes random offsets $\mathbf{z}_{s,m}$ for each data. Combining them yields a single objective for all parameters:

$$\min_{b, \{\theta_m\}} \sum_{s=1}^t \left( Y_s - \langle b, X_s \rangle \right)^2 + \sum_{m=1}^M \left( \mathbf{z}_{s,m} - \langle \theta_m, X_s \rangle \right)^2 + \lambda \left( \|b\|^2 + \sum_{m=1}^M \|\theta_m\|^2 \right). \tag{5}$$

Assuming $\mu_0 = 0$, $\Sigma_0 = \frac{1}{\lambda}I$ and $\tilde{\theta}_{0,m} = \frac{1}{\sqrt{M}}\tilde{A}_{0,m}$, the closed-form solution $(b_t^*, \{\theta_{t,m}^*\})$ coincides with the incremental updates in Eq. (2) and Eq. (4) when we identify

$$\mu_t = b_t^*, \quad \mathbf{A}_{t,m} = \theta_{t,m}^* + \tilde{\theta}_{0,m}. \tag{6}$$

**Symmetrized Loss.** Finally, from a random linear combination view (c.f. Eq. (3)) and Eq. (6), define the ensemble prediction function

$$f_\theta^{\text{linear}}(x, \zeta) = \Big\langle x, \ b + \sum_{m=1}^{M} \zeta_m \big(\theta_m + \tilde{\theta}_{0,m}\big)\Big\rangle, \tag{7}$$

and let $D = \{(X_s, Y_s, \mathbf{z}_s)\}_{s=1}^t$ with $\mathbf{z}_s = (\mathbf{z}_{s,1}, \ldots, \mathbf{z}_{s,M})$. A *symmetrized* objective includes both $(+\mathbf{z}_{s,m})$ and $(-\mathbf{z}_{s,m})$ for each data point can be defined:

$$L(\theta; D, f) := \sum_{m=1}^{M} \sum_{s \in D} \sum_{\beta \in \{\pm 1\}} \Big(Y_s + \beta\mathbf{z}_{s,m} - f\big(X_s, \ \beta e_m\big)\Big)^2 + \lambda\|\theta\|^2. \tag{8}$$

Minimizing Eq. (8) recovers the same solution as Eq. (5) since the symmetrized slack variable $\beta$ cancels out the cross-term. This "two-sided" perturbation perspective extends naturally to *Ensemble++*.

### 3.3 Ensemble++ for Nonlinear Bandits

Real-world tasks frequently require *nonlinear* function approximators (e.g., neural networks) for high-dimensional inputs or complex reward structures $f^*$. *Ensemble++* retains the same "shared ensemble factor" principle but replaces linear features with a learnable network.

**Model Architecture.** We generalize Eq. (7) by letting $h(x; w)$ be a *neural* feature extractor:

$$f_\theta(x, \zeta) = \Big\langle h(x; w), b + \sum_{m=1}^{M} \zeta_m \big(\theta_m + \tilde{\theta}_{0,m}\big)\Big\rangle,$$

where $\theta = (w, b, \{\theta_m\})$ are learnable parameters, and $\{\tilde{\theta}_{0,m}\}$ are fixed random "prior" directions. Different from Linear Ensemble++ Sampling, there is no closed-form update for $b$ and $\{\theta_m\}$.

**Symmetrized Loss and SGD.** Define the same symmetrized objective $L(\theta; D, f_\theta)$ as in Eq. (8), except that $f_\theta$ is now a neural mapping. At time step $t$, we store $(X_t, Y_t, \mathbf{z}_t)$ in a FIFO buffer $D$ (capacity $C$), then run a fixed number $G$ of SGD steps to update: $\theta \leftarrow \theta - \eta \nabla_\theta L(\theta; D, f_\theta)$.

---

**Algorithm 1** Ensemble++

---

1: Initialize $\theta = (w, b, \{\theta_m\})$, prior ensemble $\{\tilde{\theta}_{0,m}\}$, FIFO buffer $D$ of capacity $C$
2: **for** $t = 1$ to $T$ **do**
3:      Sample $\zeta_t \sim P_\zeta$; select action $X_t = \arg\max_{x \in \mathcal{X}_t} f_\theta(x, \zeta_t)$
4:      Observe reward $Y_t$; sample $\mathbf{z}_t \sim P_\mathbf{z}$; add $(X_t, Y_t, \mathbf{z}_t)$ to buffer $D$ (pop oldest if $|D| > C$)
5:      Perform SGD w.r.t. $L(\theta; D, f_\theta)$ (c.f. Eq. (8)) up to $G$ steps
6: **end for**

---

Algorithm 1 summarizes: by capping $C$ and $G$, the agent ensures constant-time updates even as $t$ grows. Through comprehensive empirical studies in Section 5 and Appendix F, Ensemble++ demonstrates strong performance on complex and high-dimensional reward functions. For clarity and reproducibility, a detailed implementation of Ensemble++ is provided in Appendix B.

## 4 Theoretical Analysis

We now analyze the Linear Ensemble++ Sampling to provide foundational theoretical justification. While derived for the linear setting, these results establish the fundamental efficacy of our core mechanism – the ensemble factor with incremental updates – which we leverage in non-conjugate environments in Section 3.3. W.L.O.G., we impose the following mild assumption.

**Assumption 4.1.** The random noise $\epsilon_t$ satisfies $\mathbb{E}\left[\exp\{s\,\epsilon_t\} \mid \mathcal{H}_t, X_t\right] \leq \exp\left(\frac{s^2}{2}\right)$, $\quad \forall s \in \mathbb{R}$, where $\mathcal{H}_t$ is the history up to time $t$. In addition, all actions satisfy $\|x\|_2 \leq 1$ for $x \in \mathcal{X}$.

## 4.1 Key Lemma: Covariance Tracking under Sequential Dependence

A critical step in analyzing Linear Ensemble++ Sampling is to ensure that its incremental updates accurately track the true posterior covariance $\boldsymbol{\Sigma}_t$. Specifically, recall the Ensemble++ update for the matrix $\mathbf{A}_t$ (Eq. (4)), which aims to approximate $\boldsymbol{\Sigma}_t^{1/2}$ even when actions $X_t$ are chosen adaptively based on prior $\{\mathbf{A}_s\}_{s<t}$. The following lemma, a variant of a sequential Johnson-Lindenstrauss type result, establishes that provided $M$ is on the order of $d \log T$, $\mathbf{A}_t \mathbf{A}_t^\top$ remains a constant-factor approximation to $\boldsymbol{\Sigma}_t$ (posterior covariance defined in Eq. (2)).

**Lemma 4.2** (Covariance Tracking via Sequential JL Variant). *Let $\{\mathbf{z}_t\}_{t=1}^T$ be random vectors in $\mathbb{R}^M$ such that they are conditionally $\sqrt{1/M}$-sub-Gaussian and are unit-norm almost surely. Define $\mathbf{A}_t$ via the recursive update Eq. (4) in Linear Ensemble++, and let $\boldsymbol{\Sigma}_t$ be the exact ridge posterior covariance from Eq. (2). If the ensemble size $M$ satisfies*

$$M \geq 320 \Big( d \log \Big( \tfrac{2 + \frac{96}{s_{\min}} \sqrt{s_{\max}^2 + T}}{\delta} \Big) + \log \Big( 1 + \tfrac{T}{s_{\min}^2} \Big) \Big) \simeq d \Big( \log \tfrac{1}{\delta} + \log T \Big), \quad (9)$$

*where $s_{\min}^2 = \inf_{\|a\|=1} \|a\|_{\boldsymbol{\Sigma}_0^{-1}}^2$ and $s_{\max}^2 = \sup_{\|a\|=1} \|a\|_{\boldsymbol{\Sigma}_0^{-1}}^2$. Then with probability at least $1 - \delta$,*

$$\forall t \leq T: \quad \tfrac{1}{2} \boldsymbol{\Sigma}_t \preccurlyeq \mathbf{A}_t \mathbf{A}_t^\top \preccurlyeq \tfrac{3}{2} \boldsymbol{\Sigma}_t.$$

**Significance.** Lemma 4.2 ensures that the Ensemble++ "covariance factor" $\mathbf{A}_t \mathbf{A}_t^\top$ tracks the true posterior $\boldsymbol{\Sigma}_t$ to within constant factors, *uniformly* for all $t \leq T$. Thus, the variance estimates used by Ensemble++ remain trustworthy at each decision point, despite the *sequential* dependencies in how actions are chosen.

**Proof sketch.** The central challenge is that standard dimensionality reduction techniques like the Johnson-Lindenstrauss (JL) lemma assume independent projections, but our bandit setting creates sequential dependencies between actions and parameters. Our approach introduces a novel sequential JL variant that accounts for this adaptive data collection process.

The proof proceeds in two main steps. First, by projecting the matrix updates onto an arbitrary direction, we can apply a sequential JL theorem to establish a concentration bound, guaranteeing the approximation holds for any single, fixed direction. Second, we employ a variance-aware covering argument to extend this per-direction guarantee to a uniform guarantee over the entire action space. This discretization step is crucial for bounding the operator norm of the error and ensuring the covariance approximation holds universally. Formal details are deferred to Appendix C.

## 4.2 Regret Bound for Linear Ensemble++ Sampling

Building on Lemma 4.2, we now show that *Linear Ensemble++ Sampling* attains near-optimal regret comparable to linear Thompson Sampling (TS). Let $P_\zeta$ be the *reference distribution* used in sampling $\zeta_t$ for action selection (Eq. (3)).

**Theorem 4.3** (Distribution-dependent Regret for Linear Ensemble++). *Suppose Assumption 4.1 holds, and Lemma 4.2 applies (i.e., $M$ satisfies Eq. (9)). Then Linear Ensemble++ Sampling, using reference vectors $\zeta_t \sim P_\zeta$, achieves the following regret bound with probability at least $1 - 2\delta$:*

$$\mathrm{Regret}(T) \leq \mathcal{O} \left( \left( \frac{\rho(P_\zeta)}{p(P_\zeta)} + \rho(P_\zeta) \right) \sqrt{T \cdot \left( \log \frac{1}{\delta} + d \log \frac{T}{d\lambda} \right) \cdot d \log \frac{T}{d\lambda}} + \frac{1}{p(P_\zeta)} \sqrt{T \log \frac{1}{\delta}} \right)$$

*where $\rho(P_\zeta)$ and $p(P_\zeta)$ are distribution-dependent constants related to the concentration and anti-concentration properties of $P_\zeta$. See Appendix D for full proof details.*

This result demonstrates that our approximation mechanism achieves regret comparable to exact Thompson Sampling while maintaining computational efficiency—a key insight that guides our approach in more complex settings. The specific big-O scaling of the regret depends on the interplay of $d, T, |\mathcal{X}|$ and the choice of $P_\zeta$, as detailed in Table 2.

**Reference Distribution Choices.** A crucial design element in Ensemble++ is the choice of sampling distribution $P_\zeta$. We can sample $\zeta_t \sim P_\zeta$ from, e.g., Gaussian distribution $\mathcal{N}(0, I_M)$, Sphere distribution $\sqrt{M} \cdot \mathcal{U}(\mathbb{S}^{M-1})$, Cube distribution $\mathcal{U}(\{\pm 1\}^M)$, or Coordinate distribution $\mathcal{U}(\{\pm\sqrt{M}e_1, \ldots, \pm\sqrt{M}e_M\})$ (scaled for unit variance projection). Each choice impacts $\rho(P_\zeta)$ and $p(P_\zeta)$; see Table 1 for examples and Appendix E for formal definitions.

Table 1: Representative values of $\rho(P_\zeta)$ and $p(P_\zeta)$ (or their relevant scalings) for typical distributions. The ratio $\frac{\rho(P_\zeta)}{p(P_\zeta)}$ influences the regret constant in Theorem 4.3. Here, $a \wedge b$ denotes the $\min(a, b)$.

| $P_\zeta$ | $\mathcal{N}(0, I_M)$ | $\sqrt{M} \cdot \mathcal{U}(\mathbb{S}^{M-1})$ | $\mathcal{U}(\{\pm 1\}^M)$ | $\mathcal{U}(\{\pm\sqrt{M}e_i\})$ |
|---|---|---|---|---|
| $\rho(P_\zeta)$ | $\mathcal{O}(\sqrt{M} \wedge \sqrt{\log|\mathcal{X}|})$ | $\mathcal{O}(\sqrt{M} \wedge \sqrt{\log|\mathcal{X}|})$ | $\mathcal{O}(\sqrt{M} \wedge \sqrt{\log|\mathcal{X}|})$ | $\mathcal{O}(\sqrt{M})$ |
| $p(P_\zeta)$ | $\frac{1}{4\sqrt{e\pi}}$ | $\frac{1}{2} - \frac{e^{1/12}}{\sqrt{2\pi}}$ | $7/32$ | $\frac{1}{2M}$ |

**Discussion of Reference Distributions.** Continuous-support distributions (e.g., Gaussian or uniform on the sphere) often yield a more favorable ratio $\rho(P_\zeta)/p(P_\zeta)$ (e.g., $\mathcal{O}(\sqrt{M})$ or $\mathcal{O}(\sqrt{\log|\mathcal{X}|})$) than discrete distributions like uniform on coordinate vectors (which can lead to an $\mathcal{O}(M^{3/2})$ factor for this ratio). This can translate to tighter regret constants. For finite action sets, the $\sqrt{\log|\mathcal{X}|}$ term typically arises from $\rho(P_\zeta)$ via covering arguments.

## 4.3 Computational Cost, Regret Guarantees, and Comparative Analysis

**Computational Cost.** Linear Ensemble++ Sampling has a per-step computational cost of $\mathcal{O}(d^2M)$ for updating $\mu_t$ and $\mathbf{A}_t$. Given $M = \Theta(d \log T)$ from Lemma 4.2, the complexity is $\mathcal{O}(d^3 \log T)$. While standard Linear Thompson Sampling has an $\mathcal{O}(d^3)$ cost, the $\log T$ factor for Ensemble++ is a modest price for a mechanism that readily extends to non-conjugate settings using incremental SGD updates, as discussed in Section 3.3. In contrast, Langevin-based methods like LMC-TS [Xu et al., 2022] can incur per-step costs $\mathcal{O}(d^2T)$.

**Inflation for Frequentist Regret.** We note that our theoretical analysis, like other related works providing frequentist bounds for TS [Abeille and Lazaric, 2017], relies on an *inflation parameter* (detailed in Appendix Appendix D) to scale the sampled posterior variance. This scaling is crucial for ensuring sufficient exploration and achieving the stated regret guarantees, thus avoiding the potential for linear regret known to affect vanilla (non-inflated) TS [Hamidi and Bayati, 2020, Abeille et al., 2025]. This parameter is implicitly accounted for in the bounds presented in Table 2, and we make its role explicit here to clarify the theoretical basis of our guarantees.

**Regret and Ensemble Size Comparison.** Table 2 summarizes the regret bounds and ensemble sizes for Ensemble++ against prior analysis of ensemble sampling in linear bandits.

Table 2: Comparison of Regret Bounds and Ensemble Sizes for Linear Bandits. $\mathcal{X}$ denotes the action set, $d$ the dimension, and $T$ the horizon. "Inv." refers to invariant contexts and "Var." to varying contexts. Ensemble++ achieves its regret bounds with an ensemble size of $\Theta(d \log T)$ across all listed settings. Citations for Linear TS: Agrawal and Goyal [2013], Abeille and Lazaric [2017].

| Method | Inv. & Compact | Var. & Compact | Inv. & Finite | Var. & Finite | Ensemble Size |
|---|---|---|---|---|---|
| Linear TS | $\mathcal{O}(d^{3/2}\sqrt{T}\log T)$ | $\mathcal{O}(d^{3/2}\sqrt{T}\log T)$ | $\mathcal{O}(d\sqrt{T\log|\mathcal{X}|}\log T)$ | $\mathcal{O}(d\sqrt{T\log|\mathcal{X}|}\log T)$ | N/A |
| Qin et al. [2022] | N/A | N/A | $\mathcal{O}(\sqrt{dT\log|\mathcal{X}|}\log\frac{|\mathcal{X}|T}{d})$ | N/A | $\Omega(|\mathcal{X}|T)$ |
| Janz et al. [2024] | $\mathcal{O}\left((d\log T)^{\frac{5}{2}}\sqrt{T}\right)$ | $\mathcal{O}\left((d\log T)^{\frac{5}{2}}\sqrt{T}\right)$ | N/A | N/A | $\Theta(d\log T)$ |
| Lee and hwan Oh [2024] | N/A | N/A | $\mathcal{O}(d^{3/2}\sqrt{T}(\log T)^{3/2})$ | N/A | $\Omega(|\mathcal{X}|\log T)$ |
| **Ensemble++ (Ours)** | $\mathcal{O}(d^{3/2}\sqrt{T}(\log T)^{3/2})$ | $\mathcal{O}(d^{3/2}\sqrt{T}(\log T)^{3/2})$ | $\mathcal{O}(d\sqrt{T\log|\mathcal{X}|}\log T)$ | $\mathcal{O}(d\sqrt{T\log|\mathcal{X}|}\log T)$ | $\Theta(d\log T)$ |

**Discussion of Comparative Results.**

- **vs. Qin et al. [2022]**: Ensemble++ offers a significant improvement, particularly an exponential reduction in ensemble size ($\Theta(d\log T)$ vs. $\Omega(|\mathcal{X}|T)$). This makes Ensemble++ far more practical and scalable for large action spaces or long horizons, addressing a key limitation of prior ensemble sampling analysis to achieve TS-comparable regret.
- **vs. Ash et al. [2022]**: For the finite arm setting, Ash et al. [2022] provide an excellent specialized result. Our work complements theirs in two key ways: (1) We provide, to our knowledge, the first rigorous regret bounds for an *incremental* ensemble update in the linear bandit setting, resolving a conjecture posed by Ash et al. [2022] (who analyzed the incremental case only for MABs). This incremental nature is vital for practical, non-conjugate extensions. (2) Ensemble++ is a *single, unified algorithm* that applies to both finite and compact action sets, whereas their analysis is specialized for the finite-arm case.
- **vs. Janz et al. [2024]**: For the compact action set, our $\mathcal{O}(d^{3/2}\sqrt{T}(\log T)^{3/2})$ regret bound is a significant improvement over the $\mathcal{O}\left((d\log T)^{\frac{5}{2}}\sqrt{T}\right)$ bound from Janz et al. [2024], while using a

comparable ensemble size (i.e., $\Theta(d \log T)$ vs. their $\tilde{\mathcal{O}}(d)$). This theoretical improvement stems directly from our novel shared-factor architecture and the tighter concentration bounds derived from our sequential Johnson-Lindenstrauss variant (Lemma 4.2).

- **vs. Lee and hwan Oh [2024]**: This concurrent work also analyzes ensemble sampling for linear bandits. While employing different analytical techniques, Ensemble++ provides distinct advantages:
  - *Ensemble Size Scaling*: Our $\Theta(d \log T)$ scales with dimension, whereas their $\Omega(|\mathcal{X}| \log T)$ scales with action set size. When $d \ll |\mathcal{X}|$ (common in practice, e.g., recommender systems), Ensemble++ requires a significantly smaller ensemble.
  - *Regret for Finite Actions*: For finite action sets, our regret $\mathcal{O}(d\sqrt{T \log |\mathcal{X}|} \log T)$ is sharper than their $\mathcal{O}(d^{3/2}\sqrt{T}(\log T)^{3/2})$ (only applicable in the invariant-context case) whenever $|\mathcal{X}| < 2^d T$.
- **Generality and Strengths**: Ensemble++ is the first approximate TS method to achieve near-optimal regret matching exact linear TS across all four common setups (Invariant/Varying contexts & Compact/Finite action sets, as shown in Table 2) with a single, unified algorithm and analysis framework. This flexibility across action space structures and contexts, without requiring algorithmic changes, is a key strength. Furthermore, it achieves this with a computationally feasible ensemble size of $\Theta(d \log T)$ and a practical per-step complexity of $\mathcal{O}(d^3 \log T)$. The core mechanism's design also prioritizes extensibility to non-linear models, as demonstrated in Section 3.3.

Overall, these theoretical results for Linear Ensemble++ Sampling confirm that it effectively balances statistical efficiency (near-optimal regret) with computational tractability (small ensemble size, manageable per-step cost) and broad applicability. This provides a solid foundation for its extension to more complex, non-linear domains.

## 5  Experiments

In this section, we investigate the efficiency and scalability of Ensemble++ in varying *linear* and *nonlinear* contextual bandits as introduced in Section 2. To empirically validate our theoretical insights, we first evaluate its performance in linear bandit settings.

### 5.1  Empirical Study on Linear Ensemble++ Sampling

We construct the *Finite-action Linear Bandit* task following prior research [Russo and Van Roy, 2018]. In this setup, the finite decision set $\mathcal{X}$ is formed by uniformly sampling actions from $[-1/\sqrt{d}, 1/\sqrt{d}]^d$, where $d$ is the ambient dimension. The reward function is linear, perturbed by an additive Gaussian noise term. A detailed implementation is provided in Appendix F.1.

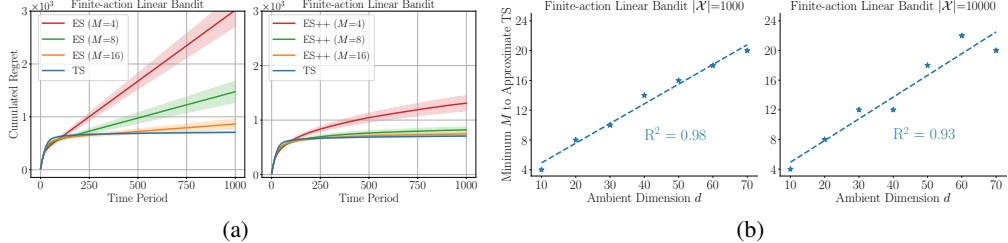

(a)                    (b)

Figure 2: Performance in Finite-action Linear Bandit. All experiments are repeated over 200 random runs to ensure robust results. (a) Comparison results under $d = 50$ and $|\mathcal{X}| = 10,000$. ES denotes Linear Ensemble Sampling, ES++ denotes Linear Ensemble++ Sampling, and TS refers to Thompson Sampling. (b) Minimum ensemble size $M$ required for Linear Ensemble++ Sampling to match the regret performance of TS.

**Advantage over Ensemble Sampling.** We consider a special case of Linear Ensemble++ Sampling using a coordinate reference distribution, which effectively performs uniform sampling among symmetrized ensemble members, akin to vanilla Linear Ensemble Sampling [Lu and Van Roy, 2017]. We compare the regret of Linear Ensemble++ Sampling (with a Gaussian reference distribution) against Linear Ensemble Sampling across different ensemble sizes $M$ in Fig. 2(a). For a fair comparison, both methods utilize the same spherical perturbation distribution. The results in Fig. 2(a) demonstrate that Linear Ensemble++ Sampling significantly outperforms Linear Ensemble Sampling across various

ensemble sizes. Notably, Linear Ensemble++ Sampling can nearly match the performance of TS with $M = 8$, achieving this with half the computational cost of Linear Ensemble Sampling.

**Optimal Ensemble Size Scaling.** To empirically validate our theoretical prediction of $M = O(d \log T)$, we investigate the minimal ensemble size $M$ required for Linear Ensemble++ Sampling to match the performance of TS. The minimal $M$ is determined by the criterion: $M = \min \left\{ M : \frac{|\text{Regret}(Ensemble++(M),T) - \text{Regret}(TS,T)|}{T} \le 0.02 \right\}$. We set a fixed, finite time horizon ($T = 1000$), which renders this metric a non-asymptotic goal. We evaluate this across varying decision set sizes $|\mathcal{X}|$ and ambient dimensions $d$. As depicted in Fig. 2(b), the minimal $M$ exhibits a *linear* relationship with $d$ and remains largely *unaffected* by $|\mathcal{X}|$, strongly supporting our theoretical claims.

## 5.2 Ensemble++ for Nonlinear Bandits

To evaluate Ensemble++ (c.f. Algorithm 1) in more complex scenarios, we consider several nonlinear contextual bandits: (1) *Quadratic Bandit*: Adapted from Zhou et al. [2020], with reward $f(x) = 10^{-2}(x^\top \Theta \Theta^\top x)$, where $x \in \mathbb{R}^d$ is the action feature and $\Theta \in \mathbb{R}^{d \times d}$ contains random variables from $\mathcal{N}(0,1)$. (2) *Neural Bandit*: A binary classification task adapted from Osband et al. [2022, 2023a], using 2-layer MLPs (50 units, ReLU) with two logit outputs. Bernoulli rewards $r \in \{0,1\}$ are sampled via softmax probabilities. (3) *UCI Shuttle*: Following Riquelme et al. [2018], Kveton et al. [2020b], we create contextual bandits for $N$-class classification using the UCI Shuttle dataset Asuncion et al. [2007]. (4) *Online Hate Speech Detection*: Built using a language dataset[3]. The agent decides to publish (reward 1 for "free", -0.5 for "hate") or block content (reward 0.5).

Detailed descriptions of these bandits are in Appendix F.2. For all algorithms, we use 2-layer MLPs (64 units) as the backbone for the first three tasks, and GPT-2[4] for the last. Implementation details for each algorithm are in Appendix A.7.1.

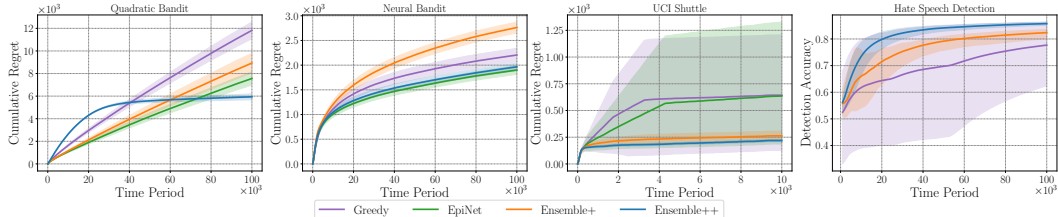

Figure 3: Comparison results across various nonlinear bandits.

**Comparison Results.** We benchmark Ensemble++ against Ensemble+ [Osband et al., 2018, 2019], EpiNet [Osband et al., 2023a,b], and a Greedy baseline (to highlight exploration needs). As shown in Fig. 3, Ensemble++ consistently achieves sublinear regret and higher accuracy. In the Quadratic Bandit, Ensemble++ converges rapidly while other baselines exhibit linear regret. For the Hate Speech Detection task, Ensemble++ outperforms Ensemble+ by 5%, showcasing its scalability with complex networks like Transformers. This task's framework, extendable to applications like recommendation systems and content moderation (see Appendix G), highlights Ensemble++'s real-world utility. Note that EpiNet could not be applied to the Hate Speech Detection task due to implementation details discussed in Appendix A.7.1. Moreover, comprehensive comparisons with LMCTS [Xu et al., 2022] detailed in Appendix F.2 confirm that Ensemble++ consistently achieves sublinear, smaller regret with bounded and lower per-step computation costs across a variety of nonlinear bandits.

**Regret vs. Computation Trade-off.** Ensemble++ achieves sublinear regret with moderate computation cost, as demonstrated in the Quadratic Bandit (results in Fig. 1). We measure computation cost by the number of network parameters and evaluate methods in the Quadratic Bandit ($d = 100, |\mathcal{X}| = 1000$), ensuring fair comparison by using identical hidden network architectures and optimization schedules for all algorithms. Additional results in the Neural Bandit (Fig. 14, Appendix F.2) corroborate this: across various ensemble sizes $M$, Ensemble++ surpasses baselines like EpiNet and Ensemble+ on the regret–compute frontier, affirming the cost-effectiveness of its

---

[3] https://huggingface.co/datasets/ucberkeley-dlab/measuring-hate-speech
[4] https://huggingface.co/openai-community/gpt2

random linear combinations plus a shared base. Appendix A.7.1 further discusses the relationship between ensemble size and network parameter size for Ensemble++ and baselines.

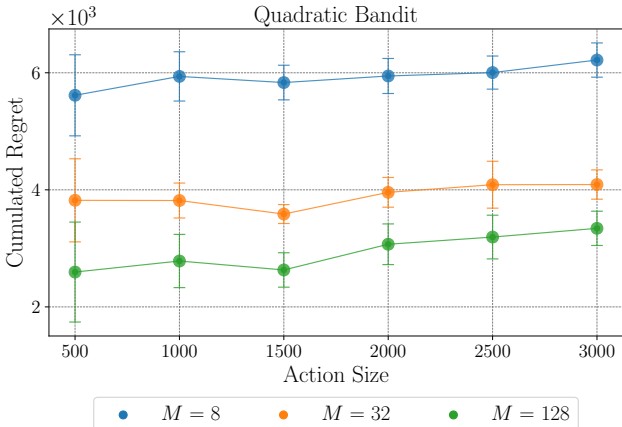

Figure 4: Scalability of Ensemble++ with varying decision set sizes ($|\mathcal{X}|$).

**Ablation Studies on Scalability.**    Consistent with linear bandit findings, the regret performance of Ensemble++ in nonlinear bandits is also largely *unaffected* by decision set size, as shown in Fig. 4. This further supports our theoretical analysis in Section 4. Larger ensemble sizes $M$ yield additional benefits, mirroring observations in linear bandits. Furthermore, we provide more detailed empirical studies in Appendix F. Specifically, we verify that Ensemble++ maintains comparable performance even with buffer capacity significantly smaller than the total time horizon in Fig. 15. Additionally, Fig. 16 provides guidance on choosing the sampling distribution $P_\zeta$ and suggests that the choice of perturbation distribution has a negligible impact on performance when using neural network.

## 6   Concluding Remarks

Thompson Sampling's principled treatment of exploration has inspired extensive research into Bayesian methods for sequential decision-making. Yet large-scale and non-conjugate contexts remained a hurdle: exact posteriors are infeasible, and naive ensemble approximations often demand huge ensemble sizes or high retraining cost. We introduced *Ensemble++*, showing how to:

- maintain a shared ensemble matrix $\mathbf{A}_t$ in the *linear* case, with ensemble size $M = \Theta(d \log T)$,
- achieve incremental $\mathcal{O}(d^2 M)$-time updates that approximate the posterior covariance square root $\Sigma_t^{1/2}$ despite adaptively collected data,
- unify base and ensemble parameters within a *symmetrized regression objective* that admits a closed-form solution (linear) or an SGD solution (neural).

As a result, *Linear Ensemble++ Sampling* achieves Thompson-like $\sqrt{T}$ regret across diverse linear bandit settings (c.f. Section 4.3 and Table 2) without the previous $M$-vs-$T$ scalability conflict typical of ensemble methods. This strong theoretical and practical foundation is seamlessly extended to the *neural* case, where the same symmetrized regression objective guides a trainable feature extractor, enabling broad applicability in complex, high-dimensional problems. Our experiments further demonstrate strong empirical performance, often superior to alternative ensemble methods, attributed to (i) a computationally feasible ensemble size and (ii) efficient incremental updates.

**Future directions.**    Ongoing work focuses on providing tighter theoretical bounds for the neural extension. A particularly promising direction is applying Ensemble++ to LLM agents. The sequential decision-making involved in multi-step reasoning and tool use presents a high-dimensional, non-conjugate challenge where managing uncertainty is critical. Naive ensembling of foundation models is computationally prohibitive, but the parameter-efficient, shared-architecture design of Ensemble++ offers a scalable method to approximate Thompson Sampling over an agent's policies. We are actively exploring how this framework can manage epistemic uncertainty in an LLM's reasoning paths, effectively allowing an agent to *explore* which strategies or tools lead to the best outcomes. By addressing the critical interplay between Bayesian exploration and scalable computation, Ensemble++ opens the door for more principled and effective online learning and decision-making in the next generation of AI agents.

## Acknowledgments and Disclosure of Funding

The work of Z.-Q. Luo was supported in part by the Guangdong Major Project of Basic and Applied Basic Research (No. 2023B0303000001), the Guangdong Provincial Key Laboratory of Big Data Computing, and the National Key Research and Development Project (No. 2022YFA1003900). The work of B. Wang was supported in part by the National Natural Science Foundation of China (No. 72394361) and an extended support project from the Shenzhen Science and Technology Program.

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

# NeurIPS Paper Checklist

1. **Claims**

   Question: Do the main claims made in the abstract and introduction accurately reflect the paper's contributions and scope?

   Answer: [Yes]

   Justification: The abstract and introduction (Section 1) clearly state the paper's main claims and contributions, which are directly supported by the theoretical analysis in Section 4 and empirical results in Section 5 and Appendix F. The claims about ensemble size scaling with $\Theta(d \log T)$, regret bounds, and empirical performance are all backed by rigorous proofs and experiments.

   Guidelines:

   - The answer NA means that the abstract and introduction do not include the claims made in the paper.
   - The abstract and/or introduction should clearly state the claims made, including the contributions made in the paper and important assumptions and limitations. A No or NA answer to this question will not be perceived well by the reviewers.
   - The claims made should match theoretical and experimental results, and reflect how much the results can be expected to generalize to other settings.
   - It is fine to include aspirational goals as motivation as long as it is clear that these goals are not attained by the paper.

2. **Limitations**

   Question: Does the paper discuss the limitations of the work performed by the authors?

   Answer: [Yes]

   Justification: The paper explicitly discusses limitations in multiple sections. In Appendix F, a gray boxed remark openly acknowledges a limitation of Theorem 2.1 regarding regret bounds when M increases beyond a threshold. The extension to neural networks in Section 3.3 also acknowledges the lack of closed-form updates for parameters. Additionally, Section 6 mentions future directions including developing tighter theoretical bounds for the neural extension.

   Guidelines:

   - The answer NA means that the paper has no limitation while the answer No means that the paper has limitations, but those are not discussed in the paper.
   - The authors are encouraged to create a separate "Limitations" section in their paper.
   - The paper should point out any strong assumptions and how robust the results are to violations of these assumptions (e.g., independence assumptions, noiseless settings, model well-specification, asymptotic approximations only holding locally). The authors should reflect on how these assumptions might be violated in practice and what the implications would be.
   - The authors should reflect on the scope of the claims made, e.g., if the approach was only tested on a few datasets or with a few runs. In general, empirical results often depend on implicit assumptions, which should be articulated.
   - The authors should reflect on the factors that influence the performance of the approach. For example, a facial recognition algorithm may perform poorly when image resolution is low or images are taken in low lighting. Or a speech-to-text system might not be used reliably to provide closed captions for online lectures because it fails to handle technical jargon.
   - The authors should discuss the computational efficiency of the proposed algorithms and how they scale with dataset size.
   - If applicable, the authors should discuss possible limitations of their approach to address problems of privacy and fairness.
   - While the authors might fear that complete honesty about limitations might be used by reviewers as grounds for rejection, a worse outcome might be that reviewers discover

limitations that aren't acknowledged in the paper. The authors should use their best judgment and recognize that individual actions in favor of transparency play an important role in developing norms that preserve the integrity of the community. Reviewers will be specifically instructed to not penalize honesty concerning limitations.

3. **Theory assumptions and proofs**

Question: For each theoretical result, does the paper provide the full set of assumptions and a complete (and correct) proof?

Answer: [Yes]

Justification: The paper clearly states all assumptions (see Assumption 4.1) for the theoretical results. Main theorems (Lemma 4.2 and Theorem 4.3) are presented with their assumptions and implications in Section 4. Complete proofs are provided in the appendix, with detailed derivations in Appendix C for the key lemma and Appendix D for regret bounds. The main paper provides intuitive proof sketches to accompany these formal proofs.

Guidelines:

- The answer NA means that the paper does not include theoretical results.
- All the theorems, formulas, and proofs in the paper should be numbered and cross-referenced.
- All assumptions should be clearly stated or referenced in the statement of any theorems.
- The proofs can either appear in the main paper or the supplemental material, but if they appear in the supplemental material, the authors are encouraged to provide a short proof sketch to provide intuition.
- Inversely, any informal proof provided in the core of the paper should be complemented by formal proofs provided in appendix or supplemental material.
- Theorems and Lemmas that the proof relies upon should be properly referenced.

4. **Experimental result reproducibility**

Question: Does the paper fully disclose all the information needed to reproduce the main experimental results of the paper to the extent that it affects the main claims and/or conclusions of the paper (regardless of whether the code and data are provided or not)?

Answer: [Yes]

Justification: The paper provides detailed experimental settings in Section 5 and Appendix F. For linear bandits, it describes the action range, dimensions, and problem scales. For nonlinear bandits, detailed environment specifications are given for each task (Appendix F.2). The model architecture, hyperparameters (ensemble size, learning rate, buffer capacity, etc.), and implementation details are specified in Appendix B. Importantly, the paper explicitly states that all experiments use P40 GPUs (except for GPT-2 experiments, which use V100s).

Guidelines:

- The answer NA means that the paper does not include experiments.
- If the paper includes experiments, a No answer to this question will not be perceived well by the reviewers: Making the paper reproducible is important, regardless of whether the code and data are provided or not.
- If the contribution is a dataset and/or model, the authors should describe the steps taken to make their results reproducible or verifiable.
- Depending on the contribution, reproducibility can be accomplished in various ways. For example, if the contribution is a novel architecture, describing the architecture fully might suffice, or if the contribution is a specific model and empirical evaluation, it may be necessary to either make it possible for others to replicate the model with the same dataset, or provide access to the model. In general. releasing code and data is often one good way to accomplish this, but reproducibility can also be provided via detailed instructions for how to replicate the results, access to a hosted model (e.g., in the case of a large language model), releasing of a model checkpoint, or other means that are appropriate to the research performed.
- While NeurIPS does not require releasing code, the conference does require all submissions to provide some reasonable avenue for reproducibility, which may depend on the nature of the contribution. For example

(a) If the contribution is primarily a new algorithm, the paper should make it clear how to reproduce that algorithm.

(b) If the contribution is primarily a new model architecture, the paper should describe the architecture clearly and fully.

(c) If the contribution is a new model (e.g., a large language model), then there should either be a way to access this model for reproducing the results or a way to reproduce the model (e.g., with an open-source dataset or instructions for how to construct the dataset).

(d) We recognize that reproducibility may be tricky in some cases, in which case authors are welcome to describe the particular way they provide for reproducibility. In the case of closed-source models, it may be that access to the model is limited in some way (e.g., to registered users), but it should be possible for other researchers to have some path to reproducing or verifying the results.

5. **Open access to data and code**

Question: Does the paper provide open access to the data and code, with sufficient instructions to faithfully reproduce the main experimental results, as described in supplemental material?

Answer: [Yes]

Justification: The paper provides the codebase link on the first page: `https://github.com/szrlee/Ensemble_Plus_Plus`. The datasets used in experiments are publicly available (UCI datasets, hate speech dataset from Hugging Face) with proper citations. For baselines comparison, the paper cites the original repositories used and mentions that some were reimplemented following those repositories' guidelines.

Guidelines:

- The answer NA means that paper does not include experiments requiring code.
- Please see the NeurIPS code and data submission guidelines (`https://nips.cc/public/guides/CodeSubmissionPolicy`) for more details.
- While we encourage the release of code and data, we understand that this might not be possible, so "No" is an acceptable answer. Papers cannot be rejected simply for not including code, unless this is central to the contribution (e.g., for a new open-source benchmark).
- The instructions should contain the exact command and environment needed to run to reproduce the results. See the NeurIPS code and data submission guidelines (`https://nips.cc/public/guides/CodeSubmissionPolicy`) for more details.
- The authors should provide instructions on data access and preparation, including how to access the raw data, preprocessed data, intermediate data, and generated data, etc.
- The authors should provide scripts to reproduce all experimental results for the new proposed method and baselines. If only a subset of experiments are reproducible, they should state which ones are omitted from the script and why.
- At submission time, to preserve anonymity, the authors should release anonymized versions (if applicable).
- Providing as much information as possible in supplemental material (appended to the paper) is recommended, but including URLs to data and code is permitted.

6. **Experimental setting/details**

Question: Does the paper specify all the training and test details (e.g., data splits, hyperparameters, how they were chosen, type of optimizer, etc.) necessary to understand the results?

Answer: [Yes]

Justification: The paper provides comprehensive experimental details. Hyperparameters are specified in Appendix B, including network architecture (2-layer MLP with 64 units), ensemble size (M=8), learning rate (0.0001), batch size (128), weight decay (0.01), and buffer capacity (C=10,000). For each environment, the paper describes in Appendix F.2 the task settings, dimensions, number of actions, and noise distributions. The choice of distributions for sampling (reference, update, perturbation) is discussed in Appendix B.1 with ablation studies in Fig. 16.

Guidelines:

- The answer NA means that the paper does not include experiments.
- The experimental setting should be presented in the core of the paper to a level of detail that is necessary to appreciate the results and make sense of them.
- The full details can be provided either with the code, in appendix, or as supplemental material.

7. **Experiment statistical significance**

Question: Does the paper report error bars suitably and correctly defined or other appropriate information about the statistical significance of the experiments?

Answer: [Yes]

Justification: The paper reports statistical significance by running multiple trials with different random seeds. As stated in Appendix F.1, the linear bandit experiments are repeated 200 times to ensure robust results. For nonlinear bandits (Appendix F.2), each experiment is repeated with 10 different random seeds. The figures (e.g., Figs. 2, 3 and 18) show error bands/bars representing the variability across these multiple runs.

Guidelines:

- The answer NA means that the paper does not include experiments.
- The authors should answer "Yes" if the results are accompanied by error bars, confidence intervals, or statistical significance tests, at least for the experiments that support the main claims of the paper.
- The factors of variability that the error bars are capturing should be clearly stated (for example, train/test split, initialization, random drawing of some parameter, or overall run with given experimental conditions).
- The method for calculating the error bars should be explained (closed form formula, call to a library function, bootstrap, etc.)
- The assumptions made should be given (e.g., Normally distributed errors).
- It should be clear whether the error bar is the standard deviation or the standard error of the mean.
- It is OK to report 1-sigma error bars, but one should state it. The authors should preferably report a 2-sigma error bar than state that they have a 96% CI, if the hypothesis of Normality of errors is not verified.
- For asymmetric distributions, the authors should be careful not to show in tables or figures symmetric error bars that would yield results that are out of range (e.g. negative error rates).
- If error bars are reported in tables or plots, The authors should explain in the text how they were calculated and reference the corresponding figures or tables in the text.

8. **Experiments compute resources**

Question: For each experiment, does the paper provide sufficient information on the computer resources (type of compute workers, memory, time of execution) needed to reproduce the experiments?

Answer: [Yes]

Justification: The paper explicitly states the computational resources used. As mentioned in Appendix F, "All experiments are conducted on P40 GPUs to maintain processing standardization." For the hate speech detection experiments with foundation models, the paper mentions that V100 GPUs were used. The paper also discusses computational complexity in Section 4.3, noting that Linear Ensemble++ has per-step complexity of $\mathcal{O}(d^3 \log T)$, and compares this to alternatives like LMC-TS with $\mathcal{O}(d^2 T)$ complexity.

Guidelines:

- The answer NA means that the paper does not include experiments.
- The paper should indicate the type of compute workers CPU or GPU, internal cluster, or cloud provider, including relevant memory and storage.
- The paper should provide the amount of compute required for each of the individual experimental runs as well as estimate the total compute.

- The paper should disclose whether the full research project required more compute than the experiments reported in the paper (e.g., preliminary or failed experiments that didn't make it into the paper).

9. **Code of ethics**

Question: Does the research conducted in the paper conform, in every respect, with the NeurIPS Code of Ethics https://neurips.cc/public/EthicsGuidelines?

Answer: [Yes]

Justification: The research in this paper conforms with the NeurIPS Code of Ethics. The paper uses publicly available datasets with proper citations. It addresses important scientific questions around computational efficiency and effectiveness of ensemble methods. The content moderation application in Appendix G directly discusses ethical considerations around using AI for content moderation, acknowledging the need for human feedback and the risks of over-blocking or under-blocking content.

Guidelines:

- The answer NA means that the authors have not reviewed the NeurIPS Code of Ethics.
- If the authors answer No, they should explain the special circumstances that require a deviation from the Code of Ethics.
- The authors should make sure to preserve anonymity (e.g., if there is a special consideration due to laws or regulations in their jurisdiction).

10. **Broader impacts**

Question: Does the paper discuss both potential positive societal impacts and negative societal impacts of the work performed?

Answer: [Yes]

Justification: The paper extensively discusses societal impacts, especially in Appendix G. Positive impacts include more efficient and accurate content moderation systems, reduced human reviewer workload, and better handling of ambiguous or borderline content. The paper explicitly discusses negative impacts like the risk of over-blocking benign content or under-blocking harmful content. The paper proposes that the uncertainty quantification in Ensemble++ helps mitigate these risks by focusing human review resources on truly ambiguous cases.

Guidelines:

- The answer NA means that there is no societal impact of the work performed.
- If the authors answer NA or No, they should explain why their work has no societal impact or why the paper does not address societal impact.
- Examples of negative societal impacts include potential malicious or unintended uses (e.g., disinformation, generating fake profiles, surveillance), fairness considerations (e.g., deployment of technologies that could make decisions that unfairly impact specific groups), privacy considerations, and security considerations.
- The conference expects that many papers will be foundational research and not tied to particular applications, let alone deployments. However, if there is a direct path to any negative applications, the authors should point it out. For example, it is legitimate to point out that an improvement in the quality of generative models could be used to generate deepfakes for disinformation. On the other hand, it is not needed to point out that a generic algorithm for optimizing neural networks could enable people to train models that generate Deepfakes faster.
- The authors should consider possible harms that could arise when the technology is being used as intended and functioning correctly, harms that could arise when the technology is being used as intended but gives incorrect results, and harms following from (intentional or unintentional) misuse of the technology.
- If there are negative societal impacts, the authors could also discuss possible mitigation strategies (e.g., gated release of models, providing defenses in addition to attacks, mechanisms for monitoring misuse, mechanisms to monitor how a system learns from feedback over time, improving the efficiency and accessibility of ML).

11. **Safeguards**

Question: Does the paper describe safeguards that have been put in place for responsible release of data or models that have a high risk for misuse (e.g., pretrained language models, image generators, or scraped datasets)?

Answer: [NA]

Justification: The paper does not introduce models or datasets that have a high risk for misuse. The method is an algorithmic approach for more efficient and effective exploration in contextual bandits. While the paper uses pretrained language models like GPT-2, these are used as feature extractors with standard published weights, rather than being novel models requiring safeguards.

Guidelines:

- The answer NA means that the paper poses no such risks.
- Released models that have a high risk for misuse or dual-use should be released with necessary safeguards to allow for controlled use of the model, for example by requiring that users adhere to usage guidelines or restrictions to access the model or implementing safety filters.
- Datasets that have been scraped from the Internet could pose safety risks. The authors should describe how they avoided releasing unsafe images.
- We recognize that providing effective safeguards is challenging, and many papers do not require this, but we encourage authors to take this into account and make a best faith effort.

12. **Licenses for existing assets**

Question: Are the creators or original owners of assets (e.g., code, data, models), used in the paper, properly credited and are the license and terms of use explicitly mentioned and properly respected?

Answer: [Yes]

Justification: The paper properly cites and credits all external assets. For datasets, it cites UCI repository [Asuncion et al., 2007] and the Hugging Face hate speech dataset (with URL). For baseline methods, it cites the original papers and repositories, specifying when code was used directly (e.g., LMCTS from `https://github.com/devzhk/LMCTS`) or reimplemented based on published descriptions. For pretrained models like GPT-2, the paper links to the Hugging Face model repository (`https://huggingface.co/openai-community/gpt2`).

Guidelines:

- The answer NA means that the paper does not use existing assets.
- The authors should cite the original paper that produced the code package or dataset.
- The authors should state which version of the asset is used and, if possible, include a URL.
- The name of the license (e.g., CC-BY 4.0) should be included for each asset.
- For scraped data from a particular source (e.g., website), the copyright and terms of service of that source should be provided.
- If assets are released, the license, copyright information, and terms of use in the package should be provided. For popular datasets, `paperswithcode.com/datasets` has curated licenses for some datasets. Their licensing guide can help determine the license of a dataset.
- For existing datasets that are re-packaged, both the original license and the license of the derived asset (if it has changed) should be provided.
- If this information is not available online, the authors are encouraged to reach out to the asset's creators.

13. **New assets**

Question: Are new assets introduced in the paper well documented and is the documentation provided alongside the assets?

Answer: [Yes]

Justification: The paper introduces the Ensemble++ algorithm and provides a codebase at `https://github.com/szrlee/Ensemble_Plus_Plus`. The algorithm is extensively documented in the paper, with pseudocode in Algorithm 1, detailed derivations in Appendix B, and hyperparameter specifications. The implementation details for reproduction are thoroughly described in Section 5 and Appendix F.

Guidelines:

- The answer NA means that the paper does not release new assets.
- Researchers should communicate the details of the dataset/code/model as part of their submissions via structured templates. This includes details about training, license, limitations, etc.
- The paper should discuss whether and how consent was obtained from people whose asset is used.
- At submission time, remember to anonymize your assets (if applicable). You can either create an anonymized URL or include an anonymized zip file.

14. **Crowdsourcing and research with human subjects**

Question: For crowdsourcing experiments and research with human subjects, does the paper include the full text of instructions given to participants and screenshots, if applicable, as well as details about compensation (if any)?

Answer: [NA]

Justification: This paper does not involve crowdsourcing or research with human subjects. The experiments are computational, using publicly available datasets and simulated environments.

Guidelines:

- The answer NA means that the paper does not involve crowdsourcing nor research with human subjects.
- Including this information in the supplemental material is fine, but if the main contribution of the paper involves human subjects, then as much detail as possible should be included in the main paper.
- According to the NeurIPS Code of Ethics, workers involved in data collection, curation, or other labor should be paid at least the minimum wage in the country of the data collector.

15. **Institutional review board (IRB) approvals or equivalent for research with human subjects**

Question: Does the paper describe potential risks incurred by study participants, whether such risks were disclosed to the subjects, and whether Institutional Review Board (IRB) approvals (or an equivalent approval/review based on the requirements of your country or institution) were obtained?

Answer: [NA]

Justification: This paper does not involve research with human subjects and therefore did not require IRB approval. The research focuses on algorithmic methods and computational experiments with existing datasets.

Guidelines:

- The answer NA means that the paper does not involve crowdsourcing nor research with human subjects.
- Depending on the country in which research is conducted, IRB approval (or equivalent) may be required for any human subjects research. If you obtained IRB approval, you should clearly state this in the paper.
- We recognize that the procedures for this may vary significantly between institutions and locations, and we expect authors to adhere to the NeurIPS Code of Ethics and the guidelines for their institution.
- For initial submissions, do not include any information that would break anonymity (if applicable), such as the institution conducting the review.

16. **Declaration of LLM usage**

Question: Does the paper describe the usage of LLMs if it is an important, original, or non-standard component of the core methods in this research? Note that if the LLM is used only for writing, editing, or formatting purposes and does not impact the core methodology, scientific rigorousness, or originality of the research, declaration is not required.

Answer: [Yes]

Justification: The paper uses foundation models (GPT-2 and Pythia14m) as feature extractors in the hate speech detection experiments described in Appendix G. This usage is clearly documented, explaining how the models are integrated into the Ensemble++ framework. The paper discusses both frozen and finetuned versions of these models and provides performance comparisons in Fig. 19.

Guidelines:

- The answer NA means that the core method development in this research does not involve LLMs as any important, original, or non-standard components.
- Please refer to our LLM policy (`https://neurips.cc/Conferences/2025/LLM`) for what should or should not be described.

# Contents

# A    Additional Discussions on Related Works

This appendix provides further background and motivation for the techniques discussed in the main text, focusing on Thompson Sampling and its limitations, local perturbation for Gaussian posteriors, and ensemble-based methods. We also compare Ensemble++ with advanced neural architectures such as Ensemble+ [Osband et al., 2018, 2019] and EpiNet [Osband et al., 2023b].

## A.1    Thompson Sampling (TS)

Thompson Sampling is a Bayesian approach for balancing exploration and exploitation in bandit or sequential decision-making problems [Thompson, 1933, Russo et al., 2018, Li and Luo, 2024]. It maintains a posterior distribution over unknown parameters (or functions) and selects actions by sampling from this posterior.

**Methodology.**    At each time step $t$, with history $\mathcal{H}_t$, TS does:

1. **Sample a model:** Draw a parameter $\theta_t \sim P(\theta \mid \mathcal{H}_t)$.
2. **Select action:** $X_t = \arg\max\limits_{x \in \mathcal{X}_t} f_{\theta_t}(x)$, where $f_\theta(\cdot)$ represents the expected reward under model $\theta$.
3. **Observe reward:** Receive $Y_t$.
4. **Update posterior:** Incorporate $(X_t, Y_t)$ into the posterior $P(\theta \mid \mathcal{H}_{t+1})$.

Because TS samples from a posterior that encodes the agent's uncertainty, it naturally allocates exploration to regions (actions) that are less well understood, while exploiting current knowledge of high-reward actions.

**Conjugate Settings.**    In special "conjugate" scenarios, posterior updates are tractable:

- **Beta-Bernoulli Bandits:** A Beta prior for each arm remains Beta after seeing Bernoulli rewards.
- **Linear-Gaussian Bandits:** With Gaussian priors and Gaussian noise, the posterior remains Gaussian with updated mean and covariance.

Here, each TS update is fast. However, in high-dimensional or non-conjugate (e.g., neural network) reward models, exact posterior inference becomes intractable.

## A.2    Challenges in Scaling Thompson Sampling

While Thompson Sampling has strong theoretical properties (e.g., near-optimal regret in finite action or linear bandit scenarios), it faces two main challenges when extended to large-scale or complex environments: **(1) intractability in non-conjugate models**, and **(2) the computational overhead of approximate inference methods**.

**Non-Conjugate Models.**    Many real-world applications (e.g., deep neural networks, highly structured rewards) do not admit closed-form updates, which is the first challenge of intractability. Direct posterior sampling in these cases requires approximate Bayesian techniques.

### A.2.1    Approximate Bayesian Inference

**Motivation.**    This leads to the second challenge: the computational overhead of such approximations. To preserve the essence of TS (sampling from a distribution over plausible models), various approximate inference methods aim to produce posterior-like samples at each step. However, many of these approaches suffer from high computational overhead.

**Prominent Approximate Methods.**

- **Laplace Approximation** [MacKay, 1992]: Approximates the posterior around its mode with a Gaussian. Scales poorly if the parameter dimension is large.
- **Variational Inference (VI)** [Blei et al., 2017]: Uses a parametric distribution and minimizes a KL divergence. Handles larger dimensions than Laplace but involves biases based on chosen family.
- **MCMC Methods** [Welling and Teh, 2011]: Iteratively generate samples from the true posterior. Accurate but often expensive in high dimensions or real-time tasks.

- **Langevin Monte Carlo (LMC)** [Xu et al., 2022]: A gradient-based MCMC approach adding noise to gradient steps. Its iterative nature can be costly in long horizons. More precisely, the per-step computation complexity grows linearly with the size of the history interaction data set.

**Key Limitations.**

- *Biased Uncertainty*: Approximate posteriors may misestimate uncertainty, harming TS's exploration.
- *Iterative Overheads*: Repeated passes over the entire history per step become impractical as $T$ grows large.
- *Scalability*: Quadratic/cubic scaling in model dimension is prohibitive for large networks.

Thus, while approximate methods broaden TS's applicability, their computational or memory costs remain problematic in large-scale, non-conjugate settings.

### A.3 Gaussian Sampling via Local Perturbation

An alternative for *linear–Gaussian* environments is **local perturbation** [Papandreou and Yuille, 2010], which incrementally updates posterior samples in $\mathcal{O}(d^2)$ per step—avoiding $\mathcal{O}(d^3)$ matrix factorizations.

**Idea.** Suppose $\theta^* \sim \mathcal{N}(\mu_0, \Sigma_0)$ and observations

$$Y_s = X_s^\top \theta^* + \omega_s^*, \quad \omega_s^* \sim \mathcal{N}(0,1).$$

Then the posterior after $t$ observations is $\mathcal{N}(\mu_t, \Sigma_t)$. Rather than factor $\Sigma_t$ at each step, local perturbation maintains

$$\tilde{A}_t = \Sigma_t \Big( \Sigma_0^{-1} \tilde{A}_0 + \sum_{s=1}^{t} X_s Z_s \Big),$$

with $\tilde{A}_0 \sim \mathcal{N}(0, \Sigma_0)$ and $Z_s \sim \mathcal{N}(0,1)$. Under a *fixed* (non-adaptive) design $\{X_s\}$, $\tilde{A}_t \sim \mathcal{N}(0, \Sigma_t)$, hence

$$\tilde{\theta}_t = \mu_t + \tilde{A}_t \quad \text{is an exact draw from } \mathcal{N}(\mu_t, \Sigma_t).$$

Both $\mu_t$ and $\tilde{A}_t$ update incrementally in $\mathcal{O}(d^2)$.

#### A.3.1 Distribution Matching Proof Outline

For completeness, we briefly sketch why local perturbation yields an *exact* posterior draw in the non-adaptive case.

Let $D_t = \{(X_s, Y_s)\}_{s=1}^{t}$. Then:

$$\mathbb{E}[\tilde{A}_t \mid D_t] = 0, \quad \mathrm{Cov}(\tilde{A}_t \mid D_t) = \Sigma_t,$$

implying $\mu_t + \tilde{A}_t \sim \mathcal{N}(\mu_t, \Sigma_t)$. The key steps:

**Mean argument.** For each $s$, $Z_s \sim N(0,1)$ is independent of $D_t$, so $\mathbb{E}[Z_s \mid D_t] = 0$. Similarly, $\tilde{A}_0 \sim N(0, \Sigma_0)$ is independent of $D_t$ and $\mathbb{E}[\tilde{A}_0 \mid D_t] = 0$. Hence,

$$\mathbb{E}[\tilde{A}_t \mid D_t] = \Sigma_t \Big( \Sigma_0^{-1} \mathbb{E}[\tilde{A}_0 \mid D_t] + \sum_{s=1}^{t} X_s \mathbb{E}[Z_s \mid D_t] \Big) = 0.$$

**Covariance argument.** Because $Z_s \sim N(0,1)$ i.i.d. and $\tilde{A}_0 \sim N(0, \Sigma_0)$,

$$\mathrm{Cov}[\tilde{A}_t \mid D_t] = \Sigma_t \Big( \Sigma_0^{-1} \mathrm{Cov}[\tilde{A}_0 \mid D_t] \Sigma_0^{-1} + \sum_{s=1}^{t} X_s X_s^\top \mathbb{E}[Z_s^2] \Big) \Sigma_t$$

$$= \Sigma_t \Big( \Sigma_0^{-1} \Sigma_0 \Sigma_0^{-1} + \sum_{s=1}^{t} X_s X_s^\top \Big) \Sigma_t$$

$$= \Sigma_t \big( \Sigma_t^{-1} \big) \Sigma_t = \Sigma_t.$$

Thus $\tilde{A}_t \sim \mathcal{N}(0, \Sigma_t)$ *conditionally on* $D_t$. Therefore, $\tilde{\theta}_t := \mu_t + \tilde{A}_t$ *has mean* $\mu_t$ *and covariance* $\Sigma_t$ *and matches exactly* $\mathcal{N}(\mu_t, \Sigma_t)$.

## A.4 Recursive Randomized Least Squares (RRLS)

**Motivation.** Motivated by bounded per-step computation requirement, one could attempt to update the parameter vector $\theta_t$ in an incremental, *recursive* manner:

$$\theta_t \;=\; \Sigma_t \Big( \Sigma_{t-1}^{-1} \theta_{t-1} + X_t \left( Y_t + Z_t \right) \Big), \tag{10}$$

where $Z_t \sim \mathcal{N}(0,1)$ is a fresh random perturbation at each time $t$. This yields the *Recursive RLS* (RRLS) algorithm:

---

**Algorithm 2** Recursive Randomized Least Squares (RRLS)

---

1: Initialize $\theta_0 \sim \mathcal{N}(\mu_0, \Sigma_0)$
2: **for** $t = 1$ to $T$ **do**
3:     $X_t = \arg\max_{x \in \mathcal{X}_t} \langle \theta_{t-1}, x \rangle$
4:     Observe $Y_t$
5:     Sample $Z_t \sim \mathcal{N}(0,1)$
6:     Update $\theta_t$ via equation 10
7: **end for**

---

**Sequential Dependency** However, RRLS introduces *sequential dependency* because the action $X_t$ chosen at time $t$ depends on the previous parameter estimate $\theta_{t-1}$, which itself depends on all past perturbations $Z_s$ and past actions $X_s$ for $s < t$. Due to this sequential dependency, the conditional expectation and covariance of $\theta_t$ no longer match those of the posterior distribution $\theta^* \mid D_t$. This is because when conditioning on $D_t$, the perturbations $Z_1, \ldots, Z_t$ are no longer independent and identically distributed (i.i.d.) as Normal random variables. This results in biased estimates and ineffective exploration, giving linear regret in some scenarios. This sequential dependency is illustrated in Figure 5.

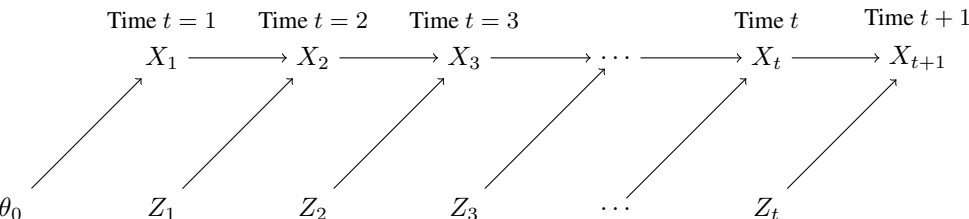

Figure 5: Sequential dependence due to the interplay between recursive updates and sequential decision-making. $Z_1, \ldots, Z_t$ are no longer independent and identically distributed (i.i.d.) when conditioned on the data $X_{t+1}$.

One potential workaround is to *resample* fresh random perturbations $\{Z_s\}_{s \le t}$ and re-fit from scratch each step (e.g., storing all historical data) to restore independence, but this is computationally and memory expensive in practice [Osband et al., 2019, Kveton et al., 2020a] and defeats the purpose of a cheap incremental method.

The next subsection describes an approach—*ensemble sampling*—that mitigates sequential dependency via multiple parallel parameter vectors, each incrementally updated.

## A.5 Ensemble Sampling (ES)

**Principle.** Ensemble Sampling [Osband and Van Roy, 2015, Osband et al., 2016, Lu and Van Roy, 2017] keeps $M$ independent parameter vectors (or models). At time $t$, it uniformly picks one model $m_t$ to decide $X_t$ and then updates all $M$ models incrementally and recursively. Intuitively, if these $M$ vectors approximate $M$ draws from the posterior, the overall policy resembles Thompson Sampling.

**Algorithm Outline.**

- *Initialization:* $\theta_{0,m} \sim \mathcal{N}(\mu_0, \Sigma_0)$ for $m = 1, \ldots, M$.

- *Action Selection:*

$$m_t \sim \text{Uniform}\{1, \ldots, M\}, \qquad X_t = \arg\max_{x \in \mathcal{X}_t} \langle \theta_{t-1,m_t}, x \rangle.$$

- *Model Updates:* Each $\theta_{t,m}$ is updated in an RRLS-like manner, but with fresh noise $Z_{t,m}$.

This balances memory usage ($M \ll T$) against sequential dependency.

Table 3: Comparison of methods for addressing sequential dependency.

| Method | Computation per Step | Memory Usage | Sequential Dependency |
|---|---|---|---|
| RLS ($M = T$) | $O(T)$ | High | None |
| ES ($M \ll T$) | $O(M)$ | Moderate | Reduced |
| RRLS ($M = 1$) | $O(1)$ | Low | High |

**Empirical Trade-offs.**

- *If $M = T$, and each model is selected exactly once at time $t$ (i.e., $m_t = t$), the Ensemble Sampling method becomes equivalent to original randomized least squares (RLS) with perturbations resampling and model retraining at each time step. This approach eliminates sequential dependency entirely but requires huge computation and memory overhead.*

- *If $M = 1$, it degenerates to RRLS with minimal memory but strong sequential dependency. Hence, by choosing $M \ll T$, one often obtains good practical performance (Fig. 6). This suggest, empirically, ES with moderate $M$ can achieve performance comparable to TS while only paying a factor of $M$ overhead in memory and a moderate per-step cost.*

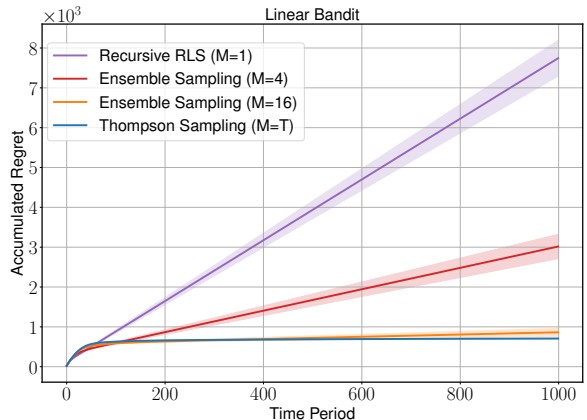

Figure 6: Ensemble Sampling (ES) with moderate $M$ achieves near-TS performance. Setup: $|\mathcal{X}| = 10,000$ and dimension $d = 50$.

**Theoretical Limitations**   Qin et al. [2022] provide a rigorous regret analysis for linear ensemble sampling that could match the regret order of exact Thompson sampling but require $M = O(|\mathcal{X}|T)$ to maintain $\sqrt{T}$ scaling in Bayesian regret, a major barrier in practical large-scale problems. This suggests *naively* we might need an ensemble size that scales linearly with $T$ or $|\mathcal{X}|$—infeasible for large action sets and long horizons tasks, contradicting with the empirical findings of a moderate size of ensembles.

**Remark A.1.** Qin et al. [2022] consider a $d$-dimensional linear bandit problem with an action set $\mathcal{X}$. When the true parameter follows a standard normal distribution $\theta^* \sim \mathcal{N}(0, I_d)$, the Bayesian regret is bounded by:

$$BR(T) \leq C\sqrt{dT \log |\mathcal{X}|} + CT\sqrt{\frac{|\mathcal{X}| \log(MT)}{M}}(d \wedge \log |\mathcal{X}|)$$

where $C > 0$ is a universal constant. This bound has two significant limitations:

1. To achieve the desired $\sqrt{T}$ scaling in Bayesian regret (ignoring constant and logarithmic factors), the ensemble size $M$ must grow linearly with the time horizon $T$. This requirement undermines the computational efficiency that ensemble sampling aims to achieve.

2. To maintain a logarithmic dependence on $|\mathcal{X}|$ in the bound, the ensemble size $M$ must scale linearly with the number of actions $|\mathcal{X}|$.

These limitations become particularly problematic when dealing with compact action spaces. For instance, consider a bandit problem where $\mathcal{X} = \mathbb{B}_2^d$ (the $d$-dimensional unit ball). To achieve a small discretization error, we need approximately $2^{d-1}$ discrete actions. Consequently, following Qin et al.'s bound, the required ensemble size $M$ would grow exponentially with dimension $d$.

**Ensemble Sampling Beyond Linear Models**   For a general function class $\mathcal{F}$, Ensemble Sampling can be extended to approximate the posterior distribution of the optimal function $f^* \in \mathcal{F}$, e.g. *Bootstrapped Ensemble* [Osband et al., 2016] and *Ensemble+* [Osband et al., 2018, 2019]. The agent maintains $M$ models, each representing a hypothesis about $f^*$ based on historical data. At each time step $t$, the agent samples a model $m_t$ uniformly from $\{1, 2, \ldots, M\}$ and selects an action: $X_t = \arg\max_{x \in \mathcal{X}_t} f_{\theta_{t,m_t}}(x)$, where $f_{\theta_{t,m_t}}(x)$ is the prediction of ensemble member $m_t$ for action $x$. After observing the reward $Y_t$, each ensemble member $m$ updates its parameters $\theta_{t+1,m}$ by performing stochastic gradient descent on the loss (Eq. (11)) starting from the previous iterate $\theta_{t,m}$:

$$L_m(\theta; D) = \sum_{s=1}^{t} \left(Y_s + Z_{s,m} - f_\theta(A_s)\right)^2 + \Psi(\theta) \tag{11}$$

where $D = \mathcal{H}_t$ and $Z_{s,m}$ are independent random perturbations added to encourage diversity among ensemble members, and $\Psi(\theta_{t+1,m})$ is a regularization term. This perturbed training procedure ensures that each ensemble member captures different aspects of the uncertainty in $f^*$, representing different plausible hypotheses consistent with the history. The random perturbations $Z_{s,m}$ are independent across time index $s$ and model index $m$. Once realized, $Z_{s,m}$ are fixed throughout the rest of the training, enabling incremental updates for real-time adaptation. This is a key computational feature compared to methods like Randomized Least Squares (RLS) or Perturbed History Exploration (PHE). In RLS and PHE, fresh independent perturbations for all historical data are introduced at each time $t$, and the model requires full retraining from scratch to ensure diverse exploration of different plausible hypotheses. Yet, it is important to note that the theoretical analysis of Ensemble Sampling beyond linear models remains an open research question.

## A.6   Concluding Remarks and Forward Outlook

Local perturbation methods (RLS, RRLS) and ensemble-based approximations collectively aim to solve large-scale or non-conjugate posterior sampling in an *online* manner. Yet:

- **Recursive RLS (RRLS)** is cheap to update but suffers from *sequential dependency* bias, often giving linear regret in adaptive settings.
- **Ensemble Sampling** lessens sequential dependency empirically with moderate size of ensembles. How, current theory suggest ensemble sampling may require $M \propto T$ or $|\mathcal{X}|$ in worst-case analyses, which is computationally or memory-intensive. Moreover, maintaining $M$ independent neural network ensembles is also computationally prohibitive for large models, even with moderate size $M = 10\,100$.

We propose Ensemble++, which addresses these drawbacks by maintaining a single shared factor for covariance approximation, with incremental updates in $\mathcal{O}(d^2 M)$ and a rigorous proof that $M \approx d \log T$ suffices to achieve near-optimal regret that matches exact Thompson sampling. Before concluding, we briefly compare with broader ensemble-based research, including Ensemble+ [Osband et al., 2018, 2019] and EpiNet [Osband et al., 2023a].

## A.7   Ensemble Methods in Broader Context

**History of Ensemble Approaches.**   Ensemble methods date back to the *Ensemble Kalman Filter* [Evensen, 1994, 2003] or Bayesian bootstrap [Rubin, 1981]. In modern literature, *Bootstrapped Ensemble* [Osband and Van Roy, 2015, Lu and Van Roy, 2017] introduced multiple models updated

with random perturbations or "bootstrap" samples of data. *Ensemble+* [Osband et al., 2018, 2019] introduced the randomized prior ensembles to enhance the exploration efficiency.

**Hypernetworks and EpiNet.** Some architectures, like *Hypermodels* [Dwaracherla et al., 2020] or *Epistemic Neural Networks (EpiNet)* [Osband et al., 2023a], treat the ensemble index or random seed as additional network inputs, effectively learning a mapping from random "epistemic index" to parameter space. Although conceptually appealing, they typically lack any rigorous understanding and proven regret bounds and may suffer from large parameter counts, as we discuss next.

### A.7.1 Detailed Comparison with EpiNet and Ensemble+

**EpiNet Overview.** EpiNet is designed to estimate epistemic uncertainty in neural networks by injecting an "epistemic index" $z \in \mathbb{R}^M$ into an MLP layer. Its final output is a combination of:

- A *base* prediction $\mu_\zeta(x)$ on the raw input $x$,
- An *epinet* MLP $\sigma_\eta^L([x, \tilde{x}, z])$ that processes the concatenation of raw input $x$, the hidden representation $\tilde{x} \in \mathbb{R}^D$ of $x$ and the random index $z$,
- A *fixed prior* $\sigma^P(x, z)$, typically a collection of $M$ small MLPs with raw input $x$ as the input, each producing a per-class offset and combined with random $z$ as the output.

Hence, the final model is

$$f_{\text{EpiNet}}(x, z) = \mu_\zeta(x) + \sigma_\eta^L([x, \tilde{x}, z]) + \sigma^P(x, z).$$

This architecture can learn uncertainty-aware predictions but often suffers from a large parameter footprint (due to multiple MLPs) and lacks any proven regret guarantees in bandit settings. Additionally, because both the *epinet* MLP and the *fixed prior* must take the raw input $x$, it is challenging to apply EpiNet to more complex networks such as Transformers. Therefore, we do not compare EpiNet in the Hate Speech Detection task.

**Ensemble+ Overview.** Ensemble+ extends *ensemble sampling* to deep neural networks using a *randomized prior* approach [Osband et al., 2018, 2019]. Concretely:

- A *shared* feature extractor processes inputs $x$ into some hidden representation $\tilde{x}$. There are $M$ heads $\{\theta_m\}$, each a simple linear layer that predicts the reward from $\tilde{x}$.
- Additionally, a *fixed prior network* is maintained as a separate feature extractor: $x \mapsto \hat{x}$. Also, there are $M$ unique random prior heads that fixed after random initialized, each predicting the additive prior reward from $\hat{x}$

This design helps capture model uncertainty by mixing learned features with a distinct randomized prior in each ensemble head. However, as with EpiNet, Ensemble+ can become large in parameter count (due to separate prior modules) and currently lacks theoretical regret bounds in deep or high-dimensional bandits.

**Parameter Counts.** We compare the number of parameters of each method. Assuming the number of parameters of the hidden feature extractor is $H$, we analyze how many additional parameters each method allocates beyond a single hidden feature extractor network.

- **EpiNet:**
  - The *epinet* MLP has hidden layers that receive $[x, \tilde{x}, z] \in \mathbb{R}^{d+D+M}$ as input and output $\mathbb{R}^{M \times C}$ (for $C$ classes or outputs). Following Osband et al. [2023b,a], we use 2-layer MLPs with 15 units and bias to construct this epinet MLP. Therefore, we count the parameters of this part as:
    $$(d + D + M + 1) \times 15 + (15 + 1) \times 15 + (15 + 1) \times (M + C)$$
    $$= 15(d + D + M + 1) + 16 \times 15 + 16 \times (M + C)$$
    $$= 15d + 15D + 31M + 15 + 240 + 16C$$
    $$= 15d + 15D + 31M + 255 + 16C.$$
  - The fixed prior $\sigma^P$ is composed of $M$ small MLPs, each adding parameters. Following Osband et al. [2023b,a], we use 2-layer MLPs with 5 units and bias to construct this prior network. It takes the raw input $x \in \mathbb{R}^d$ and each MLP outputs $\mathbb{R}^C$. Therefore, we count the parameters of this part as:
    $$M \times \big((d + 1) \times 5 + (5 + 1) \times 5 + (5 + 1) \times C\big)$$
    $$= M \times \big(5(d + 1) + 30 + 6C\big) = M \times (5d + 5 + 30 + 6C) = M \times (5d + 35 + 6C).$$

- Together, EpiNet can have a large overhead as $M$ small prior MLPs or the epinet's hidden size grow. We can calculate the total parameters as:
$$H + 15d + 15D + 31M + 255 + 16C + M \times (5d + 35 + 6C).$$

- **Ensemble+:**
  - $M$ *linear* heads, each taking the hidden representation $\tilde{x} \in \mathbb{R}^D$ as input, produce the main ensemble predictions $\mathbb{R}^C$, and each head has the same random prior network. Therefore, we count the parameters of this part as:
  $$2 \times M \times ((D+1) \times C) = 2MDC + 2MC.$$
  - A *separate* feature extractor for the $M$ linear prior network heads to form the prior offset. Therefore, we count the parameters of this part as $H$.
  - This leads to approximately $2M$ last-layer transforms (main + prior), plus the potential duplication of feature extractors. We can calculate the total parameters as:
  $$2H + 2 \times M \times ((D+1) \times C) = 2H + 2MDC + 2MC.$$

- **Ensemble++:**
  - There are $M$ *linear* heads without bias for the main ensemble for uncertainty estimation, each mapping $\mathbb{R}^D \to \mathbb{R}^C$ and equipped with the same prior networks. Therefore, we calculate the parameters of this part as:
  $$2 \times M \times D \times C.$$
  - One more *base* linear head with bias to estimate the mean. The parameters of this part are $(D+1) \times C$.
  - In total, this results in $(2M+1)$ linear layers of dimension $\mathbb{R}^D \to \mathbb{R}^C$, but each is relatively lightweight. We can calculate the total parameters as:
  $$H + 2 \times M \times D \times C + (D+1) \times C = H + (2M+1)DC + C.$$

**Computational Efficiency.**

- **EpiNet:** Concatenates $[x, \tilde{x}, z]$ of dimension $(d + D + M)$, driving up the input size for the epinet MLP. The fixed prior $\sigma^P$ also has multiple small MLPs. Training/inference cost grows significantly with $M$.

- **Ensemble+**: Combines a main network and a separate prior network, each with $M$ linear heads. While each head is relatively cheap, maintaining two feature extractors can be more expensive than Ensemble++'s single shared representation.

- **Ensemble++:** Each ensemble head is just a $\mathbb{R}^D \to \mathbb{R}^C$ linear map, combined additively with a base head. Training/inference overhead remains modest, as backprop only flows through linear heads plus one shared feature extractor. The stop-gradient trick can further reduce overhead.

**Practical Implications.** Empirical studies [Li et al., 2024] show that EpiNet's parameter overhead often slows training and can degrade exploration. Likewise, Ensemble+ can be parameter-heavy if the prior network is large or if $M$ grows. By contrast, Ensemble++ uses a single shared representation with relatively simple linear heads (for both ensemble and prior), yielding a smaller parameter footprint and faster training. Crucially, *Ensemble++* also provides a theoretical foundation guaranteeing near-optimal linear-bandit regret with $M = \widetilde{O}(d \log T)$, whereas EpiNet and Ensemble+ currently lack proven regret bounds.

**Conclusion.** In summary, EpiNet and Ensemble+ push ensemble-based methods toward richer neural function approximation but face large parameter counts and no *a priori* theoretical guarantees. Ensemble++ uses lightweight linear heads on top of a shared feature extractor—much more efficient in large-scale or real-time settings—and *does* come with rigorous regret analyses for the linear bandit case. Extending those theoretical insights to deep bandits is an ongoing research direction, but empirical results (§5) show strong performance of *Ensemble++* relative to EpiNet and Ensemble+.

# B    Ensemble++ Algorithm Details

Here we provide detailed derivations and design choices for the Ensemble++ algorithm. Let $x \in \mathcal{X}$ denote the input, and $h(x; w)$ be the shared feature extractor parameterized by $w$. The extracted features are denoted by
$$\tilde{x} = h(x; w).$$

The base network $\psi(\tilde{x}; b)$, parameterized by $b$, estimates the mean prediction based on the shared features. The ensemble components $\{\psi(\text{sg}(\tilde{x}); \theta_m)\}_{m=1}^M$, parameterized by $\theta_m$, capture the uncertainty in the prediction. The stop-gradient operator $\text{sg}(\cdot)$ prevents gradients from flowing through $\tilde{x}$ when computing gradients with respect to $\theta_m$, effectively decoupling the ensemble components from the shared layers. The prior ensemble components $\{\psi(\tilde{x}; \theta_{0,m})\}_{m=1}^M$ are fixed throughout the learning process, incentivizing diverse exploration with prior variations in the initial stage where the data region is under-explored. Put all together, $\theta = \{w, b, \theta_1, \ldots, \theta_M, \theta_{0,1}, \ldots, \theta_{0,M}\}$ are the model parameters. By default, we choose $\psi$ as a linear function:

$$\psi(\tilde{x}; \theta) = \langle \tilde{x}, \theta \rangle.$$

the Ensemble++ model predicts via random linear combinations of the base network and ensemble components, with the prior ensemble components fixed throughout the learning process. The model is defined as:

$$f_\theta^{++}(x, \zeta_t) = \psi(\tilde{x}; b) + \psi(\text{sg}(\tilde{x}); \sum_{m=1}^M \zeta_{t,m}\theta_m) + \text{sg}(\psi(\tilde{x}; \sum_{m=1}^M \zeta_{t,m}\theta_{0,m})), \tag{12}$$

where $\zeta_t = (\zeta_{t,1}, \ldots, \zeta_{t,M})^\top$ is a random vector sampled from an sampling distribution $P_\zeta$.

**Loss Function Derivation.** Starting from the loss function $L(\theta; D)$ with symmetric auxiliary variables:

$$\frac{1}{2M} \sum_{m=1}^M \sum_{s=1}^N \sum_{\beta \in \{1,-1\}} (Y_s + \beta \mathbf{z}_{s,m} - \psi(\tilde{x}_s; b) - \beta\psi(\text{sg}(\tilde{x}_s); \theta_m) - \beta \, \text{sg}(\psi(\tilde{x}_s; \theta_{0,m})))^2 + \Phi(\theta). \tag{13}$$

Expanding the square and summing over $\beta$:

$$\sum_{\beta \in \{1,-1\}} (Y_s + \beta \mathbf{z}_{s,m} - \psi(\tilde{x}_s; b) - \beta\psi(\text{sg}(\tilde{x}_s); \theta_m) - \beta \, \text{sg}(\psi(\tilde{x}_s; \theta_{0,m})))^2$$

$$= \sum_{\beta \in \{1,-1\}} ((Y_s - \psi(\tilde{x}_s; b)) + \beta(\mathbf{z}_{s,m} - \text{sg}(\psi(\tilde{x}_s; \theta_{0,m})) - \psi(\text{sg}(\tilde{x}_s); \theta_m)))^2$$

$$= 2 \left( (Y_s - \psi(\tilde{x}_s; b))^2 + (\mathbf{z}_{s,m} - \text{sg}(\psi(\tilde{x}_s; \theta_{0,m})) - \psi(\text{sg}(\tilde{x}_s); \theta_m))^2 \right),$$

since the cross terms cancel out due to summing over $\beta \in \{1,-1\}$. This leads to the simplified loss function $L(\theta; D)$:

$$\frac{1}{M} \sum_{m=1}^M \sum_{s=1}^N \left[ \frac{1}{2} (Y_s - \psi(\tilde{x}_s; b))^2 + \frac{1}{2} (\mathbf{z}_{s,m} - \text{sg}(\psi(\tilde{x}_s; \theta_{0,m})) - \psi(\text{sg}(\tilde{x}_s); \theta_m))^2 \right] + \Phi(\theta). \tag{14}$$

**Gradient Computations.** The gradients with respect to the shared parameters $(w, b)$ are derived solely from the base network loss:

$$\nabla_w L(\theta; D) = \sum_{s=1}^N (\psi(\tilde{x}_s; b) - Y_s) \nabla_{\tilde{x}_s} \psi(\tilde{x}_s; b) \nabla_w h(A_s; w), \tag{15}$$

$$\nabla_b L(\theta; D) = \sum_{s=1}^N (\psi(\tilde{x}_s; b) - Y_s) \nabla_b \psi(\tilde{x}_s; b). \tag{16}$$

The gradients with respect to the ensemble parameters $\theta_m$ are independent of the base network:

$$\nabla_{\theta_m} L(\theta; D) = \sum_{s=1}^N (\psi(\text{sg}(\tilde{x}_s); \theta_m) - Z_{s,m}) \nabla_{\theta_m} \psi(\text{sg}(\tilde{x}_s); \theta_m). \tag{17}$$

Note that due to the stop-gradient operator $\text{sg}(\cdot)$, the ensemble components do not contribute to the gradients of shared parameters.

**Classification Loss Function.** For classification tasks, we use the cross-entropy loss function instead of the squared loss function in Eq. (13):

$$L(\theta; D) = \frac{1}{2M} \sum_{m=1}^{M} \sum_{s=1}^{N} \sum_{\beta \in \{1, -1\}} \text{CE}\left(f^{++}(X_s, \beta e_m), [Y_s, 1 - Y_s]\right) + \Phi(\theta) \tag{18}$$

where $\text{CE}(X, Y) = -\sum_j Y_j (X_j - \log \sum_i \exp X_i)$ is the cross-entropy loss function, and $[Y_s, 1 - Y_s]$ is the one-hot encoding of the label $Y_s$.

**Hyperparameters.** For the practical implementation of Ensemble++, we utilize a 2-layer MLP with 64 units and ReLU activation to construct the feature extractor $h(x; w)$. The ensemble size is set to $M = 8$, and we use a symmetrized slack variable $\beta = 0.01$ and weight decay $\lambda = 0.01$ across all nonlinear bandit tasks. During training, a unified batch size of 128 and a learning rate of 0.0001 are employed for all tasks. Based on the ablation studies presented in Fig. 16, we adopt the Coordinate update distributions, Sphere reference distribution, and Sphere perturbation distribution to achieve optimal performance. Two key hyperparameters for training are the buffer capacity $C$ and the number of gradient steps $G$. As demonstrated in Fig. 15, we found that setting the buffer capacity to $C = 10,000$ achieves strong performance, even for tasks with $T = 100,000$ steps. For gradient steps $G$, task-specific optimal values are determined via parameter sweeps. In the case of synthetic bandits, such as the Quadratic Bandit and Neural Bandit, we use a smaller gradient step size of $G = 1$. For tasks involving UCI datasets, we increase the gradient steps to $G = 50$ in Mushroom task and $G = 100$ in Shuttle task.

**Summary.** We have provided an overview of Ensemble++ in Algorithm 1. Through comprehensive empirical studies detailed in Section 5 and Appendix F, Ensemble++ demonstrates strong performance in practice, even for complex and high-dimensional reward functions. Moreover, we observe that Ensemble++ exhibits consistent performance in the linear setting, where the agent's regret remains largely unaffected by the size of the decision set. This finding, illustrated in Fig. 4, strongly supports the effectiveness of our neural extension. For clarity and reproducibility, we provide the codebase of Ensemble++ at `https://anonymous.4open.science/r/EnsemblePlus2-1E54`.

## B.1 Design of Reference and Perturbation Distributions

In the Ensemble++ algorithm, we distinguish between two key distributions: the perturbation distribution $P_z$ and the reference distribution $P_\zeta$. While both play critical roles, they serve different purposes and operate under different theoretical constraints.

The perturbation distribution $P_z$ generates vectors $\mathbf{z}_t$ used in the incremental update of the ensemble matrix factor $\mathbf{A}_t$ through Equation equation 4. These vectors must satisfy strict theoretical requirements—they need to be almost surely unit-norm, and conditionally $\frac{1}{\sqrt{M}}$-sub-Gaussian—to ensure the covariance tracking guarantees established in Lemma 4.2.

In contrast, the reference distribution $P_\zeta$ is used during action selection to sample exploration vectors $\zeta_t$ for generating parameter samples according to $\theta_t = \hat{\theta}_t + \mathbf{A}_t \zeta_t$. The theoretical properties of $P_\zeta$ directly impact the exploration behavior and regret bounds of the algorithm. Specifically, as shown in Theorem 4.3, the regret bound depends on the ratio $\frac{\rho(P_\zeta)}{p(P_\zeta)}$, where:

- $\rho(P_\zeta)$ captures the concentration properties of $P_\zeta$, relating to how tightly the distribution concentrates around its mean
- $p(P_\zeta)$ quantifies the anti-concentration properties, which are crucial for ensuring adequate exploration in the action space

For effective implementation, the reference distribution $P_\zeta$ should satisfy zero-mean and unit-variance properties ($\mathbb{E}[\zeta_t] = 0$ and $\mathbb{E}[\zeta_t \zeta_t^\top] = I_M$) to maintain proper statistical interpretation when used with $\mathbf{A}_t$. The scaling factors in certain distributions (e.g., $\sqrt{M}$ for sphere distribution) ensure this variance normalization.

We consider five candidate distributions for $P_\zeta$, each with different theoretical guarantees for exploration and computational properties:

1. **Gaussian Distribution** ($\zeta_t \sim \mathcal{N}(0, I_M)$):

2. **Sphere Distribution** ($\zeta_t \sim \sqrt{M} \cdot \mathcal{U}(\mathbb{S}^{M-1})$):

3. **Cube Distribution** ($\zeta_t \sim \mathcal{U}(\{1, -1\}^M)$):

4. **Coordinate Distribution** ($\zeta_t \sim \mathcal{U}(\sqrt{M}\{\pm e_1, \ldots, \pm e_M\})$):

5. **Sparse Distribution** ($s$-sparse random vectors):

As shown in Table 1, continuous distributions (Gaussian, Sphere, and Cube) generally offer more favorable $\frac{\rho(P_\zeta)}{p(P_\zeta)}$ ratios compared to discrete distributions like the Coordinate distribution. This translates to tighter regret constants in Theorem 4.3. The anti-concentration property $p(P_\zeta)$ is particularly important as it directly influences how effectively the algorithm explores the parameter space, with larger values leading to more efficient exploration.

For sampling algorithms and the detailed theoretical analysis of the properties of these distributions (isotropy, concentration, and anticoncentration), see Appendix E.

## C Technical Details for Lemma 4.2 (Incremental Covariance Tracking)

This appendix provides a detailed proof for Lemma 4.2, which establishes that the Ensemble++, with a sufficiently large ensemble size $M$, can maintain an accurate approximation of the posterior covariance matrix incrementally. The core of the argument relies on adapting sequential Johnson-Lindenstrauss (JL) results to the context of adaptive data collection in bandits.

**Proof Sketch**   The recursive update rule for the ensemble matrix factor $\mathbf{A}_t$ in Ensemble++, given by Eq. (4) in the main paper, can be expressed in terms of the inverse covariance $\boldsymbol{\Sigma}_t^{-1}$ as:

$$\boldsymbol{\Sigma}_t^{-1}\mathbf{A}_t = \boldsymbol{\Sigma}_{t-1}^{-1}\mathbf{A}_{t-1} + X_t\mathbf{z}_t^\top. \tag{19}$$

Our proof strategy involves two main steps:

1. **Reduction to a Sequential Sum:** We first simplify the problem by analyzing the projection of Eq. (19) onto an arbitrary direction. This transforms the matrix update into a sum of vector products, resembling the structure required by sequential JL theorems.

2. **Application of Sequential JL and Discretization:** We then apply a sequential Johnson-Lindenstrauss theorem to bound the error in this sum. To extend this result from a single direction to all directions (i.e., to bound the operator norm of the error matrix), we employ a discretization argument over the unit sphere.

To build intuition, we first consider a simplified multi-armed bandit (MAB) setting where the feature vectors $X_t$ are standard basis vectors. In this scenario, the covariance matrix $\boldsymbol{\Sigma}_t$ becomes diagonal. The update rule in Eq. (19), when projected appropriately, reduces to the incremental update for the variance estimate of a single arm, as detailed in example 1. The core challenge, even in this simpler MAB case, is the sequential dependence: the choice of arm $X_t$ (and thus the data $x_s$ in Eq. (21)) depends on past random perturbations $\{\mathbf{z}_s\}_{s<t}$. Standard JL arguments do not directly apply under such adaptivity, necessitating the use of a sequential JL theorem (Theorem C.6).

**Example 1** (Approximate Posterior in a Multi-Armed Bandit)**.** Consider a multi-armed bandit with $K$ independent arms. Let the unknown mean reward of arm $i$ be $\theta_i^*$. We place a Gaussian prior on each arm's mean: $\theta_i^* \sim \mathcal{N}(\mu_0^i, (\sigma_0^i)^2)$. At each time step $t$, the algorithm selects an arm $X_t \in \{e_1, \ldots, e_K\}$, where $e_i$ is the $i$-th standard basis vector. The posterior variance $(\sigma_t^i)^2$ for arm $i$ is updated via Sherman-Woodbury formula, which simplifies to:

$$\frac{1}{(\sigma_t^i)^2} = \frac{1}{(\sigma_{t-1}^i)^2} + \mathbf{1}_{\{X_t = e_i\}},$$

and $(\sigma_k^i)^2$ remains unchanged for $k \neq i$ if arm $k$ was not chosen (assuming unit observation noise variance for simplicity).

Ensemble++ aims to approximate the posterior samples. It stores a factor $m_t^i \in \mathbb{R}^M$ for each arm $i$ such that its squared norm $\|m_t^i\|^2$ approximates the posterior variance $(\sigma_t^i)^2$. An approximate posterior sample for arm $i$ is then formed as $\mu_t^i + \langle m_t^i, \zeta \rangle$, where $\zeta \sim \mathcal{N}(0, I_M)$ is a random vector.

To maintain $m_t^i$ efficiently, an incremental update rule is derived from the general form in Eq. (19). For a specific arm $i$, this update becomes:

$$\frac{1}{(\sigma_t^i)^2} m_t^i = \frac{1}{(\sigma_{t-1}^i)^2} m_{t-1}^i + \mathbf{1}_{\{X_t = e_i\}} \mathbf{z}_t, \tag{20}$$

where $\mathbf{z}_t \in \mathbb{R}^M$ are fresh random perturbation vectors at each step $t$. For initialization, we set $m_0^i = \sigma_0^i \mathbf{z}_0^i$ for some random vector $\mathbf{z}_0^i$ (e.g., satisfying conditions in Theorem C.6), so that ideally $\|m_0^i\|^2 \approx (\sigma_0^i)^2$.

The crucial observation is that the choice of arm $X_t$ (and thus the indicator $\mathbf{1}_{\{X_t = e_i\}}$) depends on all past data and consequently on the past random vectors $\mathbf{z}_0, \ldots, \mathbf{z}_{t-1}$. Let $x_{s,i} := \mathbf{1}_{\{X_s = e_i\}}$ be the indicator that arm $i$ was chosen at step $s$. Rewriting Eq. (20) for a fixed arm $i$ by unrolling the recursion:

$$\frac{1}{(\sigma_t^i)^2} m_t^i = \sum_{s=0}^{t} x_{s,i} \mathbf{z}_s, \quad \text{while the true precision is} \quad \frac{1}{(\sigma_t^i)^2} = \sum_{s=0}^{t} x_{s,i}^2 = \sum_{s=0}^{t} x_{s,i}$$

(since $x_{s,i}$ is 0 or 1, so $x_{s,i}^2 = x_{s,i}$). The goal is to ensure that the squared norm of the sum of random vectors approximates the sum of squares of the adaptive coefficients:

$$\left\| \sum_{s=0}^{t} x_{s,i} \mathbf{z}_s \right\|^2 \approx \sum_{s=0}^{t} x_{s,i}^2 \quad \text{(uniformly over } t \in \{0, \ldots, T\} \text{ and all arms )} \tag{21}$$

Standard JL arguments typically require $x_{s,i}$ to be independent of $\mathbf{z}_s$, which is violated here due to the adaptive nature of $X_t$. This motivates the need for a sequential Johnson-Lindenstrauss theorem, such as Theorem C.6, which is designed to handle such sequential dependencies.

For the general linear bandit case, where $X_t$ are arbitrary bounded feature vectors (not just standard basis vectors), the approach is analogous but more involved. We cannot simply look at individual diagonal elements of $\mathbf{\Sigma}_t$. Instead, we analyze the behavior of the matrix update Eq. (19) by projecting it onto arbitrary directions $a \in \mathbb{S}^{d-1}$:

$$a^\top \mathbf{\Sigma}_t^{-1} \mathbf{A}_t = a^\top \mathbf{\Sigma}_{t-1}^{-1} \mathbf{A}_{t-1} + (a^\top X_t) \mathbf{z}_t^\top.$$

This expression has a similar sequential sum structure. The term $a^\top X_t$ is an adaptive scalar coefficient, and $\mathbf{z}_t^\top$ is the random vector. The core idea is to show that $\|a^\top \mathbf{\Sigma}_t^{-1} \mathbf{A}_t\|^2$ concentrates around its expectation, which is related to $a^\top \mathbf{\Sigma}_t^{-1} a$. The proof then requires a discretization argument (covering the unit sphere $\mathbb{S}^{d-1}$ with a finite set of representative directions) and a union bound to extend the concentration results from these discrete directions to the entire continuous space. This allows us to bound the operator norm $\|\mathbf{\Sigma}_t^{-1/2} \mathbf{A}_t \mathbf{A}_t^\top \mathbf{\Sigma}_t^{-1/2} - \mathbf{I}\|$, which is equivalent to the desired statement in Lemma 4.2.

In the subsequent subsections, we rigorously formalize each of these components, starting with the sequential Johnson-Lindenstrauss tools.

## C.1 Fundamental Probability Tool: The Sequential Johnson-Lindenstrauss Theorem

As outlined in the proof sketch, our analysis of the incremental updates in Ensemble++ under adaptive data collection hinges on a sequential version of the Johnson-Lindenstrauss (JL) lemma. This section introduces the necessary definitions and the specific sequential JL theorem we utilize. While the original tool from [Li, 2024a] was developed for scalar processes, our work extends these concepts to handle the high-dimensional vector processes inherent in linear bandits with ensemble methods, requiring the non-trivial discretization argument detailed in Appendix C.3.

We begin by defining some key concepts within a filtered probability space. Let $(\Omega, \mathcal{F}, \mathbb{F} = (\mathcal{F}_t)_{t \in \mathbb{N}}, \mathbb{P})$ be a complete filtered probability space, where $\mathbb{F}$ is the filtration.

**Definition C.1** (Adapted Stochastic Process). A stochastic process $(X_t)_{t \in I}$, where $I$ is a vector set (e.g., $\mathbb{N}$ or $\{t \in \mathbb{N} : t \geq t_0\}$), is said to be *adapted* to the filtration $\mathbb{F} = (\mathcal{F}_t)_{t \in I}$ if, for every $t \in I$, the random variable $X_t$ is $\mathcal{F}_t$-measurable.

**Definition C.2** (Conditionally $\sigma$-Sub-Gaussian Random Variable / Process). A random variable $X \in \mathbb{R}$ is $\sigma$-*sub-Gaussian* (for $\sigma \geq 0$) if its moment generating function satisfies:

$$\mathbb{E}[\exp(\lambda X)] \leq \exp\left(\frac{\lambda^2 \sigma^2}{2}\right), \quad \forall \lambda \in \mathbb{R}.$$

A stochastic process $(X_t)_{t \geq 1}$ taking values in $\mathbb{R}$ is *conditionally $\sigma_t$-sub-Gaussian* with respect to a filtration $(\mathcal{F}_t)_{t \geq 0}$ if $X_t$ is $\mathcal{F}_t$-measurable, and for a non-negative process $(\sigma_{t-1})_{t \geq 1}$ where $\sigma_{t-1}$ is $\mathcal{F}_{t-1}$-measurable, we have:

$$\mathbb{E}[\exp(\lambda X_t) \mid \mathcal{F}_{t-1}] \leq \exp\left(\frac{\lambda^2 \sigma_{t-1}^2}{2}\right), \quad \text{almost surely (a.s.),} \quad \forall \lambda \in \mathbb{R}.$$

If $\sigma_t = \sigma$ (a constant) for all $t$, the process is called conditionally $\sigma$-sub-Gaussian.

For a random vector $X \in \mathbb{R}^M$ (or a vector process $(X_t)_{t \geq 1} \subset \mathbb{R}^M$), it is said to be $\sigma$-*sub-Gaussian* (or conditionally $\sigma_t$-sub-Gaussian) if for every fixed vector $v \in \mathbb{S}^{M-1}$, the scalar random variable $\langle v, X \rangle$ (or the scalar process $(\langle v, X_t \rangle)_{t \geq 1}$) is $\sigma$-sub-Gaussian (or conditionally $\sigma_t$-sub-Gaussian, respectively).

**Definition C.3** (Almost Surely Unit-Norm). A random vector $X \in \mathbb{R}^M$ is said to be *almost surely (a.s.) unit-norm* if $\|X\|_2 = 1$ almost surely.

**Remark C.4** (Properties of Perturbation Distribution $P_{\mathbf{z}}$). The perturbation vectors $\mathbf{z}_t \in \mathbb{R}^M$ used in Ensemble++ are drawn from a distribution $P_{\mathbf{z}}$. For our theoretical analysis, particularly for applying Theorem C.6, these distributions are chosen and scaled such that specific properties are met. When *normalizing* all specific distributions discussed in Appendix E, both the spherical distribution $\mathcal{U}(\mathbb{S}^{M-1})$ and the uniform over scaled cube $\mathcal{U}(\frac{1}{\sqrt{M}}\{1, -1\}^M)$ satisfy:

1. Each $\mathbf{z}_t$ is almost surely unit-norm (Definition C.3). For the discrete cube, this holds because $\|(\frac{\pm 1}{\sqrt{M}}, \ldots, \frac{\pm 1}{\sqrt{M}})\|_2^2 = \sum_{j=1}^{M}(\frac{1}{\sqrt{M}})^2 = M \cdot \frac{1}{M} = 1$.

2. The process $(\mathbf{z}_t)_{t \geq 1}$ is conditionally $\sigma$-sub-Gaussian with $\sigma = \frac{1}{\sqrt{M}}$ according to Definition C.2.

While our ultimate goal is to establish concentration results for adaptive processes in bandits, we first review the classical Johnson-Lindenstrauss lemma which forms the foundation of our analysis. This will help contextualize the sequential variant we subsequently employ.

The essence of geometry preservation within the context of dimensionality reduction can be mathematically formulated as the challenge of designing a probability distribution over matrices that effectively retains the norm of any vector within a specified error margin after transformation. Specifically, for a given vector $x \in \mathbb{R}^n$, the objective is to ensure that with probability at least $1 - \delta$, the norm of $x$ after transformation by a matrix $\Pi = [\mathbf{z}_1, \ldots, \mathbf{z}_n] \in \mathbb{R}^{m \times n}$ drawn from the distribution $\mathcal{D}_{\varepsilon, \delta}$ remains an $\varepsilon$-approximation of its original norm, as shown below:

$$\mathbb{P}_{\Pi \sim \mathcal{D}_{\varepsilon, \delta}}\left(\|\Pi x\|_2^2 \in [(1-\varepsilon)\|x\|_2^2, (1+\varepsilon)\|x\|_2^2]\right) \geq 1 - \delta \tag{22}$$

A foundational result in this domain, the following Johnson-Lindenstrauss (JL) lemma, establishes a theoretical upper bound on the reduced dimension $m$, achievable while adhering to the above-prescribed fidelity criterion.

**Lemma C.5** (JL lemma [Johnson and Lindenstrauss, 1984]). *For any $0 < \varepsilon, \delta < 1/2$, there exists a distribution $\mathcal{D}_{\varepsilon, \delta}$ on $\mathbb{R}^{m \times n}$ for $m = O(\varepsilon^{-2} \log(1/\delta))$ that satisfies eq. equation 22.*

The standard Johnson-Lindenstrauss lemma (Lemma C.5) provides dimension reduction guarantees for fixed vectors (or a fixed set of vectors). However, this classical result is insufficient for our bandit setting, where vectors are chosen adaptively based on past observations. As illustrated in Example 1, the choice of arm $X_t$ depends on all past perturbation vectors $\mathbf{z}_0, \ldots, \mathbf{z}_{t-1}$, creating a sequential dependency that violates the independence assumptions of the standard JL lemma. To address this challenge, we require a sequential version of the JL lemma that can handle these adaptive processes. The following theorem provides exactly such a tool:

**Theorem C.6** (Sequential Johnson-Lindenstrauss Theorem, adapted from [Li, 2024a]). *Fix $\varepsilon \in (0, 1)$, and let $\{\mathcal{F}_t\}_{t \geq 0}$ be a filtration. Consider random vectors $\{\mathbf{z}_t\}_{t \geq 0} \subset \mathbb{R}^M$ adapted to $\{\mathcal{F}_t\}_{t \geq 0}$, satisfying:*

- ***Initial vector $\mathbf{z}_0$:*** *$\mathbf{z}_0$ is $\mathcal{F}_0$-measurable, $\mathbb{E}[\|\mathbf{z}_0\|_2^2] = 1$, and $\left|\|\mathbf{z}_0\|_2^2 - 1\right| \leq \varepsilon/2$ almost surely.*

- ***Subsequent vectors*** *$(\mathbf{z}_t)_{t\geq 1}$: For $t \geq 1$, the process $\{\mathbf{z}_t\}_{t\geq 1}$ is conditionally $\sqrt{c_0/M}$-sub-Gaussian (see Definition C.2), where $c_0 > 0$ is an absolute constant. Furthermore, each $\|\mathbf{z}_t\|_2 = 1$ almost surely (Definition C.3).*

*Let $\{x_t\}_{t\geq 1} \subset \mathbb{R}$ be adapted to $\{\mathcal{F}_{t-1}\}_{t\geq 1}$ and satisfy $x_t^2 \leq c_x$ a.s., where $c_x > 0$ is an absolute constant. For a fixed $x_0 \in \mathbb{R} \setminus \{0\}$, if*

$$M \geq \frac{16\,c_0\,(1+\varepsilon)}{\varepsilon^2}\left(\log\left(\tfrac{1}{\delta}\right) + \log\left(1 + \tfrac{c_x\,T}{x_0^2}\right)\right),$$

*then with probability at least $1 - \delta$,*

$$\forall t = 0, \ldots, T: \quad (1-\varepsilon)\sum_{i=0}^{t} x_i^2 \;\leq\; \left\|\sum_{i=0}^{t} x_i\,\mathbf{z}_i\right\|_2^2 \;\leq\; (1+\varepsilon)\sum_{i=0}^{t} x_i^2.$$

## C.2  Applying Sequential JL to the Incremental Update

This subsection details how the Sequential Johnson-Lindenstrauss Theorem (Theorem C.6) is applied to analyze the incremental update rule of Ensemble++. The goal is to show that for any fixed direction $a \in \mathbb{S}^{d-1}$, the squared norm of the projected ensemble factor, $\|a^\top \boldsymbol{\Sigma}_t^{-1}\mathbf{A}_t\|_2^2$, remains close to the true projected precision, $a^\top \boldsymbol{\Sigma}_t^{-1}a$, uniformly over time.

We aim to establish bounds on the approximation error for the posterior variance. For a chosen direction $a \in \mathbb{S}^{d-1}$ and a desired precision $\varepsilon \in (0, 1)$, we define the "good event" $\mathcal{G}_t(a, \varepsilon)$ at time $t \in \mathcal{T} := \{0, 1, \ldots, T\}$ as the event where the approximate posterior variance $a^\top \mathbf{A}_t\mathbf{A}_t^\top a$ is $\varepsilon$-close (multiplicatively) to the true posterior variance $a^\top \boldsymbol{\Sigma}_t a$:

$$\mathcal{G}_t(a, \varepsilon) := \left\{|a^\top \mathbf{A}_t\mathbf{A}_t^\top a - a^\top \boldsymbol{\Sigma}_t a| \leq \varepsilon(a^\top \boldsymbol{\Sigma}_t a)\right\}. \tag{23}$$

The overall good event at time $t$ for all directions is then

$$\mathcal{G}_t(\varepsilon) := \bigcap_{a \in \mathbb{S}^{d-1}} \mathcal{G}_t(a, \varepsilon). \tag{24}$$

Lemma 4.2 in the main paper essentially claims that $\mathcal{G}_t(1/2)$ holds with high probability for all $t \in \mathcal{T}$.

**Reduction to Scalar Sequences for Theorem C.6**   To utilize the Sequential Johnson-Lindenstrauss Theorem, we need to transform our matrix update equation into a form that involves sums of scalar coefficients multiplying random vectors. For a fixed $a \in \mathbb{S}^{d-1}$, we let $\mathbf{s}(a) = a^\top \boldsymbol{\Sigma}_0^{-1/2}\mathbf{Z}_0$ and $s(a)^2 = a^\top \boldsymbol{\Sigma}_0^{-1}a$. We define short notation $\mathbf{z}_0 := \mathbf{s}(a)^\top / s(a)$ and $x_0 := s(a)$. For $t \geq 1$, we define $x_t = a^\top X_t$.

With these definitions, we can relate the incremental update to scalar sequences:

$$a^\top \boldsymbol{\Sigma}_t^{-1}\mathbf{A}_t = \underbrace{a^\top \boldsymbol{\Sigma}_0^{-1/2}\mathbf{Z}_0}_{\mathbf{s}(a)=\mathbf{z}_0^\top x_0} + \sum_{i=1}^{t} \underbrace{a^\top (X_i)}_{x_i}\,\mathbf{z}_i^\top,$$

$$a^\top \boldsymbol{\Sigma}_t^{-1}a = \underbrace{a^\top \boldsymbol{\Sigma}_0^{-1}a}_{x_0^2} + \sum_{i=1}^{t} \underbrace{a^\top (X_i)(X_i)^\top a}_{x_i^2}$$

These equations transform the matrix update into sequences $(x_t)_{t\geq 0}$ and $(\mathbf{z}_t)_{t\geq 0}$ that can be analyzed using Theorem C.6.

**Filtration and Measurability**   To apply Theorem C.6, we need to define the filtration and verify the measurability conditions. Let $\mathcal{H}_t$ be the $\sigma$-algebra generated from history $(\mathcal{X}_1, X_1, Y_1, \ldots, \mathcal{X}_{t-1}, X_{t-1}, Y_{t-1}, \mathcal{X}_t)$. Denote $\mathcal{Z}_1 = \sigma(\mathbf{Z}_0)$ and $\mathcal{Z}_t = \sigma(\mathbf{Z}_0, \mathbf{z}_1, \ldots, \mathbf{z}_{t-1})$ for $t \geq 2$. We observe the following statistical relationships:

- $\mathbf{z}_t \perp\!\!\!\perp (\mathcal{H}_t, X_t, \mathcal{Z}_t)$, while $X_t$ depends on $\mathcal{H}_t, \mathcal{Z}_t$
- $\mathbf{A}_{t-1} \in \sigma(\mathcal{H}_t, \mathcal{Z}_t)$
- $\mu_{t-1}, \boldsymbol{\Sigma}_{t-1} \in \mathcal{H}_t$

For all $t \geq \mathbb{N}$, let us define the sigma-algebra $\mathcal{F}_t = \sigma(\mathcal{H}_{t+1}, \mathcal{Z}_{t+1}, X_{t+1})$. We can verify $\mathcal{F}_k \subseteq \mathcal{F}_l$ for all $k \leq l$. Thus $\mathbb{F} = (\mathcal{F}_t)_{t \in \mathbb{N}}$ is a filtration. With this construction, we can verify that $(\mathbf{z}_t)_{t \geq 0}$ is adapted to $(\mathcal{F}_t)_{t \geq 0}$ and $(x_t)_{t \geq 1}$ is adapted to $(\mathcal{F}_{t-1})_{t \geq 1}$, satisfying the conditions in Theorem C.6.

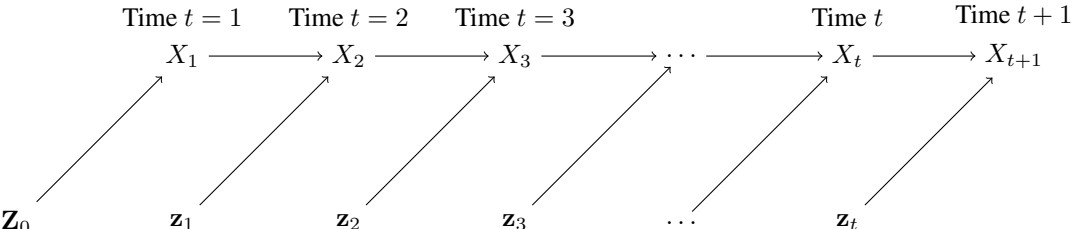

Figure 7: Sequential Dependence Structure: The choice of action $X_t$ depends on past random perturbations, illustrating why standard JL results are insufficient for this setting.

### C.2.1 Prior Approximation (Time $t = 0$)

First, we need to ensure that the initial condition for $\mathbf{z}_0(a)$ in Theorem C.6 holds. Standard covering argument for the unit sphere and the distributional Johnson-Lindenstrauss lemma (Lemma C.5, [Li, 2024b]) establish that when

$$M \geq M_1(\varepsilon, \delta) := 256\varepsilon^{-2}(d \log 9 + \log(2/\delta)), \tag{25}$$

the initial good event for prior approximation $\mathcal{G}_0(\varepsilon/2)$ holds with probability at least $1 - \delta$.

Under the event $\mathcal{G}_0(\varepsilon/2)$, we have:

$$(1 - \varepsilon/2)a^\top \boldsymbol{\Sigma}_0 a \leq \|a^\top \boldsymbol{\Sigma}_0^{1/2} \mathbf{Z}_0\|^2 \leq (1 + \varepsilon/2)a^\top \boldsymbol{\Sigma}_0 a, \quad \forall a \in \mathbb{S}^{d-1}$$

$$\Leftrightarrow \quad \|\mathbf{Z}_0 \mathbf{Z}_0^\top - \boldsymbol{I}\| \leq \varepsilon/2$$

$$\Leftrightarrow \quad (1 - \varepsilon/2)a^\top \boldsymbol{\Sigma}_0^{-1} a \leq \|a^\top \boldsymbol{\Sigma}_0^{-1/2} \mathbf{Z}_0\|^2 \leq (1 + \varepsilon/2)a^\top \boldsymbol{\Sigma}_0^{-1} a, \quad \forall a \in \mathbb{S}^{d-1}. \tag{26}$$

Recall $\mathbf{s}(a) = a^\top \boldsymbol{\Sigma}_0^{-1/2} \mathbf{Z}_0$ and $s(a)^2 = a^\top \boldsymbol{\Sigma}_0^{-1} a$. Using the short notation $\mathbf{z}_0 = \mathbf{s}(a)^\top/s(a)$, equation Eq. (26) implies that $|\|\mathbf{z}_0\|^2 - 1| \leq (\varepsilon/2)$, satisfying the initial condition required by Theorem C.6.

### C.2.2 Posterior Approximation (Time $t \geq 1$)

We now verify the remaining conditions needed to apply Theorem C.6 for $t \geq 1$.

Notice that $x_0^2 = a^\top \boldsymbol{\Sigma}_0^{-1} a \geq \inf_{a \in \mathbb{S}^{d-1}} a^\top \boldsymbol{\Sigma}_0^{-1} a = s_{\min}^2$. By the assumption of bounded features in Assumption 4.1, we have $x_t^2 = (a^\top X_t)^2 \leq 1$ for $t \geq 1$. That is, the sequence $(a^\top X_t)_{t \geq 1}$ is 1-square-bounded for any $a \in \mathbb{S}^{d-1}$.

We can also verify that $(\mathbf{z}_t)_{t \geq 1}$ is $1/\sqrt{M}$-sub-Gaussian and has unit-norm when the perturbation distribution $P_z$ is either the Cube $\mathcal{U}(\{1, -1\}^M)$ or the Sphere $\mathcal{U}(\mathbb{S}^{M-1})$.

Under the prior approximation event $\mathcal{G}_0(\varepsilon/2)$, we can apply Theorem C.6 to show that for any fixed $a \in \mathbb{S}^{d-1}$,

$$\forall t \in \mathcal{T}, E_t(a, \varepsilon) := \left\{ |a^\top \boldsymbol{\Sigma}_t^{-1} \mathbf{A}_t \mathbf{A}_t^\top \boldsymbol{\Sigma}_t^{-1} a - a^\top \boldsymbol{\Sigma}_t^{-1} a| \leq \varepsilon a^\top \boldsymbol{\Sigma}_t^{-1} a \right\} \tag{27}$$

holds with probability at least $1 - \delta$ when

$$M \geq \frac{16(1+\varepsilon)}{\varepsilon^2} \left( \log\left(\frac{1}{\delta}\right) + \log\left(1 + \frac{T}{s_{\min}^2}\right) \right). \tag{28}$$

## C.3 From Per-Direction to Uniform Guarantee: Variance-Aware Discretization

In Appendix C.2, we established that for any fixed direction $a \in \mathbb{S}^{d-1}$, the approximate precision $\|a^\top \boldsymbol{\Sigma}_t^{-1} \mathbf{A}_t\|_2^2$ is close to the true precision $a^\top \boldsymbol{\Sigma}_t^{-1} a$ with high probability (event $E_t(a, \varepsilon)$ in Eq. (27)). However, Lemma 4.2 requires a stronger guarantee: that the approximation holds uniformly for *all* directions $a \in \mathbb{S}^{d-1}$. This section details the discretization argument used to bridge this gap.

We first need some standard tools for covering the unit sphere:

**Lemma C.7** (Covering Number of a Sphere [Vershynin, 2012])**.** *There exists a set $\mathcal{C}_\iota \subset \mathbb{S}^{d-1}$ with $|\mathcal{C}_\iota| \leq (1 + 2/\iota)^d$ such that for all $x \in \mathbb{S}^{d-1}$ there exists a $y \in \mathcal{C}_\iota$ with $\|x - y\|_2 \leq \iota$.*

**Lemma C.8** (Computing Spectral Norm on a Covering Set [Vershynin, 2012])**.** *Let $\mathbf{A}$ be a symmetric $d \times d$ matrix, and let $\mathcal{C}_\iota$ be an $\iota$-covering of $\mathbb{S}^{d-1}$ for some $\iota \in (0, 1/2)$. Then,*

$$\|\mathbf{A}\|_{op} = \sup_{x \in \mathbb{S}^{d-1}} |x^\top \mathbf{A} x| \leq \frac{1}{1 - 2\iota} \sup_{x \in \mathcal{C}_\iota} |x^\top \mathbf{A} x|.$$

**Insufficiency of Standard Discretization**   Utilizing standard discretization for computing the spectral norm as in Lemma C.8, we could set $\iota = 1/4$ and show that

$$\bigcap_{a \in \mathcal{C}_{1/4}} E_t(a, \varepsilon/2T) \subseteq \mathcal{G}_t(\varepsilon).$$

This follows because:

$$
\begin{aligned}
\|\boldsymbol{\Sigma}_t^{-1/2} \mathbf{A}_t \mathbf{A}_t^\top \boldsymbol{\Sigma}_t^{-1/2} - \boldsymbol{I}\|_{op} &= \sup_{x \in \mathbb{S}^{d-1}} \frac{|x^\top (\boldsymbol{\Sigma}_t^{-1} \mathbf{A}_t \mathbf{A}_t^\top \boldsymbol{\Sigma}_t^{-1} - \boldsymbol{\Sigma}_t^{-1}) x|}{x^\top \boldsymbol{\Sigma}_t^{-1} x} \\
&\leq \frac{2}{\lambda_{\min}(\boldsymbol{\Sigma}_t^{-1})} \sup_{a \in \mathcal{C}_{1/4}} |a^\top (\boldsymbol{\Sigma}_t^{-1} \mathbf{A}_t \mathbf{A}_t^\top \boldsymbol{\Sigma}_t^{-1} - \boldsymbol{\Sigma}_t^{-1}) a| \\
&\leq 2\varepsilon' \frac{\sup_{a \in \mathcal{C}_{1/4}} a^\top \boldsymbol{\Sigma}_t^{-1} a}{\lambda_{\min}(\boldsymbol{\Sigma}_t^{-1})} \leq 2\varepsilon' \cdot \kappa(\boldsymbol{\Sigma}_t^{-1}) \leq 2T\varepsilon'.
\end{aligned}
$$

Then by union bound over $\mathcal{C}_{1/4}$ and plugging in $\varepsilon/2T$ to Eq. (28), we would require $M \geq \Omega(dT^2 \log T)$ to ensure $\bigcap_{a \in \mathcal{C}_{1/4}} E_t(a, \varepsilon/2T)$ holds with probability at least $1 - \delta$.

This result is not acceptable as the per-step computational complexity grows polynomially with the interaction steps $T$. We now introduce a variance-aware discretization approach to resolve this issue.

**Variance-Aware Discretization**   The key contribution here is that we choose a variance-weighted norm to measure discretization error. This variance-awareness, together with a specific choice of $O(1/\sqrt{T})$-discretization error and a constant approximation error $\varepsilon$, allows us to achieve an $M = \Theta(d \log T)$ bound.

Let $\mathbf{S}_t = \boldsymbol{\Sigma}_t^{-1} \mathbf{A}_t$ and $\boldsymbol{\Gamma}_t = \boldsymbol{\Sigma}_t^{-1/2} \mathbf{A}_t$. From Eq. (27), we know that the event

$$\forall t \in \mathcal{T}, E_t(a, \varepsilon') = \left\{ \frac{|a^\top \mathbf{S}_t \mathbf{S}_t^\top a - a^\top \boldsymbol{\Sigma}_t^{-1} a|}{a^\top \boldsymbol{\Sigma}_t^{-1} a} \leq \varepsilon' \right\}$$

holds with probability at least $1 - \delta'$ when

$$M \geq \frac{16(1 + \varepsilon')}{(\varepsilon')^2} \left( \log\left(\frac{1}{\delta'}\right) + \log\left(1 + \frac{T}{s_{\min}^2}\right) \right).$$

Let $\mathcal{C}_\iota \subset \mathbb{S}^{d-1}$ be the $\iota$-covering set from Lemma C.7, and assume the event $\bigcap_{a \in \mathcal{C}_\iota} E_t(a, \varepsilon')$ holds. Let $x \in \mathbb{S}^{d-1}$ and $y \in \mathcal{C}_\iota$ such that $\|x - y\| \leq \iota$. Define $u = \boldsymbol{\Sigma}_t^{-1/2} x$ and $v = \boldsymbol{\Sigma}_t^{-1/2} y$. We want to

bound the difference between the error for $x$ and the error for $y$:

$$\frac{|x^\top \mathbf{S}_t \mathbf{S}_t^\top x - x^\top \boldsymbol{\Sigma}_t^{-1} x|}{x^\top \boldsymbol{\Sigma}_t^{-1} x} - \frac{|y^\top \mathbf{S}_t \mathbf{S}_t^\top y - y^\top \boldsymbol{\Sigma}_t^{-1} y|}{y^\top \boldsymbol{\Sigma}_t^{-1} y}$$

$$= \frac{|u^\top \boldsymbol{\Gamma}_t \boldsymbol{\Gamma}_t^\top u - u^\top u|}{u^\top u} - \frac{|v^\top \boldsymbol{\Gamma}_t \boldsymbol{\Gamma}_t^\top v - v^\top v|}{v^\top v} = \frac{|\|\boldsymbol{\Gamma}_t u\|^2 - \|u\|^2|}{\|u\|^2} - \frac{|\|\boldsymbol{\Gamma}_t v\|^2 - \|v\|^2|}{\|v\|^2}$$

$$\leq \left| \frac{\|\boldsymbol{\Gamma}_t u\|^2}{\|u\|^2} - \frac{\|\boldsymbol{\Gamma}_t v\|^2}{\|v\|^2} \right| = \underbrace{\left| \frac{\|\boldsymbol{\Gamma}_t u\|^2 - \|\boldsymbol{\Gamma}_t v\|^2}{\|u\|^2} \right|}_{(I)} + \underbrace{\|\boldsymbol{\Gamma}_t v\|^2 \left| \frac{1}{\|u\|^2} - \frac{1}{\|v\|^2} \right|}_{(II)}.$$

We now bound terms $(I)$ and $(II)$ separately. Without loss of generality, assume $\|u\| \geq \|v\|$. Recall $s_{\max}^2 \geq a^\top \boldsymbol{\Sigma}_0^{-1} a \geq s_{\min}^2$ for all $a \in \mathbb{S}^{d-1}$. Since $\|u\| = x^\top \boldsymbol{\Sigma}_t^{-1} x = x^\top (\boldsymbol{\Sigma}_0^{-1} + \sum_{s=1}^t X_s X_s^\top) x$, we have $s_{\min}^2 \leq \|u\| \leq s_{\max}^2 + t$.

For term $(I)$:

$$(I) \leq \frac{(\|\boldsymbol{\Gamma}_t u\| - \|\boldsymbol{\Gamma}_t v\|)(\|\boldsymbol{\Gamma}_t u\| + \|\boldsymbol{\Gamma}_t v\|)}{\|u\|^2} \leq \frac{\|\boldsymbol{\Gamma}_t (u - v)\|}{s_{\min}} \left( \frac{\|\boldsymbol{\Gamma}_t u\|}{\|u\|} + \frac{\|\boldsymbol{\Gamma}_t v\|}{\|v\|} \right)$$

$$\leq \frac{\|\boldsymbol{\Gamma}_t\| \|u - v\|}{s_{\min}} (2\|\boldsymbol{\Gamma}_t\|) \leq \frac{2\|\boldsymbol{\Gamma}_t\|^2 \|\boldsymbol{\Sigma}_t^{-1/2}\| \iota}{s_{\min}} \leq \frac{2\|\boldsymbol{\Gamma}_t\|^2 \iota \sqrt{s_{\max}^2 + t}}{s_{\min}}.$$

For term $(II)$:

$$(II) \leq \frac{\|\boldsymbol{\Gamma}_t v\|^2}{\|v\|^2} \frac{\|u\|^2 - \|v\|^2}{\|u\|^2} \leq \|\boldsymbol{\Gamma}_t\|^2 \frac{\|u\|^2 - \|v\|^2}{\|u\|^2} \leq \|\boldsymbol{\Gamma}_t\|^2 \frac{(\|u\| - \|v\|)(\|u\| + \|v\|)}{\|u\|^2}$$

$$\leq \frac{2\|\boldsymbol{\Gamma}_t\|^2 \|u - v\|}{s_{\min}} \leq \frac{2\|\boldsymbol{\Gamma}_t\|^2 \|\boldsymbol{\Sigma}_t^{-1/2}\| \iota}{s_{\min}} \leq \frac{2\|\boldsymbol{\Gamma}_t\|^2 \iota \sqrt{s_{\max}^2 + t}}{s_{\min}}.$$

Combining terms $(I)$ and $(II)$, we get:

$$\|\boldsymbol{\Sigma}_t^{-1/2} \mathbf{A}_t \mathbf{A}_t^\top \boldsymbol{\Sigma}_t^{-1/2} - \boldsymbol{I}\|_{\text{op}} = \sup_{x \in \mathbb{S}^{d-1}} \frac{|x^\top (\boldsymbol{\Sigma}_t^{-1} \mathbf{A}_t \mathbf{A}_t^\top \boldsymbol{\Sigma}_t^{-1} - \boldsymbol{\Sigma}_t^{-1}) x|}{x^\top \boldsymbol{\Sigma}_t^{-1} x}$$

$$\leq \frac{4\|\boldsymbol{\Gamma}_t\|^2 \iota \sqrt{s_{\max}^2 + t}}{s_{\min}} + \sup_{y \in \mathcal{C}_\iota} \frac{|y^\top (\boldsymbol{\Sigma}_t^{-1} \mathbf{A}_t \mathbf{A}_t^\top \boldsymbol{\Sigma}_t^{-1} - \boldsymbol{\Sigma}_t^{-1}) y|}{y^\top \boldsymbol{\Sigma}_t^{-1} y}$$

$$\leq \frac{4\|\boldsymbol{\Gamma}_t\|^2 \iota \sqrt{s_{\max}^2 + t}}{s_{\min}} + \varepsilon'. \tag{29}$$

Let us set

$$\iota = \frac{\alpha s_{\min}}{4\sqrt{s_{\max}^2 + T}},$$

where $\alpha$ will be determined shortly.

An equivalent formulation of the norm is $\|\boldsymbol{\Gamma}_t\|^2 = \lambda_{\max}(\boldsymbol{\Gamma}_t \boldsymbol{\Gamma}_t^\top)$. Therefore:

$$\|\boldsymbol{\Sigma}_t^{-1/2} \mathbf{A}_t \mathbf{A}_t^\top \boldsymbol{\Sigma}_t^{-1/2} - \boldsymbol{I}\|_{\text{op}} = \max\{\lambda_{\max}(\boldsymbol{\Gamma}_t \boldsymbol{\Gamma}_t^\top) - 1, 1 - \lambda_{\min}(\boldsymbol{\Gamma}_t \boldsymbol{\Gamma}_t^\top)\}.$$

From Eq. (29), we get:

$$\lambda_{\max}(\boldsymbol{\Gamma}_t \boldsymbol{\Gamma}_t^\top) \leq \frac{1 + \varepsilon'}{1 - \alpha}, \quad \lambda_{\min}(\boldsymbol{\Gamma}_t \boldsymbol{\Gamma}_t^\top) \geq 1 - \varepsilon' - \alpha \lambda_{\max}(\boldsymbol{\Gamma}_t \boldsymbol{\Gamma}_t^\top) \geq 1 - \varepsilon' - \frac{\alpha(1 + \varepsilon')}{1 - \alpha}.$$

**Claim 1.** If $\frac{1+\varepsilon'}{1-\alpha} = 1 + \varepsilon$ and $\varepsilon' + \frac{\alpha(1+\varepsilon')}{1-\alpha} = \varepsilon$, then

$$1 - \varepsilon \leq \lambda_{\min}(\boldsymbol{\Gamma}_t \boldsymbol{\Gamma}_t^\top) \leq \lambda_{\max}(\boldsymbol{\Gamma}_t \boldsymbol{\Gamma}_t^\top) \leq 1 + \varepsilon.$$

For our target precision $\varepsilon = 1/2$, the parameter values $(\varepsilon', \alpha) = (1/4, 1/6)$ satisfy Claim 1. With these values, the discretization error $\iota$ becomes:

$$\iota = \frac{s_{\min}}{24\sqrt{s_{\max}^2 + T}}.$$

The covering number is $|\mathcal{C}_\iota| \leq (1 + 2/\iota)^d \leq (1 + (48/s_{\min})\sqrt{s_{\max}^2 + T})^d$. By union bound and defining $\delta' = \delta/(1 + (48/s_{\min})\sqrt{s_{\max}^2 + T})^d$, we have

$$\mathbb{P}\left(\bigcap_{t \in \mathcal{T}} \mathcal{G}_t(1/2) \mid \mathcal{G}_0(1/4)\right) \geq 1 - \delta,$$

when

$$M \geq M_2(\delta) := \frac{16(5/4)}{(1/4)^2}\left(d\log\left(\frac{1 + (48/s_{\min})\sqrt{s_{\max}^2 + T}}{\delta}\right) + \log\left(1 + \frac{T}{s_{\min}^2}\right)\right).$$

This gives us a constant of 320 multiplying the logarithmic terms.

## C.4   Putting Everything Together

We now consolidate the requirements on the ensemble size $M$ to ensure that Lemma 4.2 holds. We want to show that with probability at least $1 - \delta$, for all $t \in \{0, 1, \ldots, T\}$:

$$\|\boldsymbol{\Sigma}_t^{-1/2}\mathbf{A}_t\mathbf{A}_t^\top \boldsymbol{\Sigma}_t^{-1/2} - \boldsymbol{I}\|_{\mathrm{op}} \leq \varepsilon, \tag{30}$$

where we target $\varepsilon = 1/2$.

When

$$M \geq M_3 := \max\{M_1(1/4, \delta/2), M_2(\delta/2)\},$$

we have

$$\mathbb{P}\left(\bigcap_{t \in \mathcal{T}} \mathcal{G}_t(1/2)\right) = \mathbb{P}\left(\bigcap_{t \in \mathcal{T}} \mathcal{G}_t(1/2) \mid \mathcal{G}_0(1/4)\right)\mathbb{P}\left(\mathcal{G}_0(1/4)\right) \geq (1 - \delta/2)^2 \geq 1 - \delta.$$

With some calculations, we derive:

$$M_1(1/4, \delta/2) = 1024(d\log 9 + \log(4/\delta)),$$

and

$$M_2(\delta/2) = 320\left(d\log\left(\frac{2 + (96/s_{\min})\sqrt{s_{\max}^2 + T}}{\delta}\right) + \log\left(1 + \frac{T}{s_{\min}^2}\right)\right).$$

Since the total time periods $T$ is the dominant growing term, there exists a constant $T_0$ such that $M_3 = M_2(\delta/2)$ when $T > T_0$. This leads to the key insight that $M$ scales as $\tilde{\Omega}(d\log T)$, rather than polynomially with $T$.

**Remark C.9** (Constants and Parameters). The constants involved in the bounds depend on known properties of the algorithm's components (like the perturbation distribution). The parameters $s_{\min}, s_{\max}$ are related to the initial covariance properties, while $T$ is the horizon and $d$ is the feature dimension. The analysis crucially demonstrates that the ensemble size $M$ does not need to depend polynomially on $T$, which is essential for the practical efficiency of Ensemble++.

This completes the proof of Lemma 4.2, showing that with a suitable ensemble size $M = \Theta(d\log T)$, Ensemble++ can maintain accurate posterior uncertainty estimates throughout the learning process, despite the sequential dependencies introduced by the adaptive data collection.

# D    Technical Details in Regret Analysis for Theorem 4.3

## D.1    General Regret Bound

We start by providing a general analytical framework for agents, potentially randomized, operating in generic bandit environments. Let us introduce a few necessary definitions to facilitate the understanding and analysis. The confidence bound is used for uncertainty estimation over the true function $f^*$ given the history $\mathcal{H}_t$.

**Definition D.1** (Confidence bounds). Confidence bounds are a sequence of real-valued $\mathcal{H}_t$-measurable functions $L_t(\cdot)$ and $U_t(\cdot)$ for $t \in [T]$ such that, with probability at least $1 - \delta$, the joint event $\mathcal{E} = \cap_{t \in [T]} \mathcal{E}_t$ holds, where $\mathcal{E}_t := \{f^*(a) \in [L_t(a), U_t(a)], \forall a \in \mathcal{X}_t\}$.

The agent may not perform well unless it is well-behaved, defined by *reasonableness* and *optimism*. Intuitively, an agent that explores too much or too little will incur a high regret. Reasonableness and optimism are the mechanisms for controlling these potential flaws respectively.

**Definition D.2** (Reasonableness). Given confidence bounds $L_t(\cdot)$ and $U_t(\cdot)$ for $t \in [T]$, a (randomized) agent is called *reasonable* if it produces a sequence of functions $(\tilde{f}_t(\cdot), t \in [T])$ such that with probability at least $1 - \delta$, the joint event $\tilde{\mathcal{E}} = \cap_{t \in [T]} \tilde{\mathcal{E}}_t$ holds, where $\tilde{\mathcal{E}}_t := \{\tilde{f}_t(a) \in [L_t(a), U_t(a)], \forall a \in \mathcal{X}_t\}$.

In short, *reasonableness* ensures that the chosen action according to $\tilde{f}_t$ is close to the best action, which ensures the agent does not explore actions unnecessarily. The following *optimism* guarantees that the agent explores sufficiently.

**Definition D.3** (p-optimism). Let p be a sequence of positive real numbers $(p_t, t \in [T])$. We say a (randomized) agent is p-optimistic when it produces a sequence of functions $(\tilde{f}_t(\cdot), t \in [T])$ such that for all $t \in [T]$, $\tilde{f}_t(\cdot)$ is $p_t$-optimistic, i.e., $\mathbb{P}(\max_{a \in \mathcal{X}_t} \tilde{f}_t(a) \geq \max_{a \in \mathcal{X}_t} f^*(a) \mid \mathcal{H}_t) \geq p_t$.

The generic agent satisfying the conditions on *reasonableness* and *optimism* has desired behavior.

Building upon these definitions, we establish a general regret bound applicable to any agent satisfying these conditions.

**Theorem D.4** (General Regret Bound). *Given confidence bounds as defined in Definition D.1, and assuming the agent is reasonable and* p-*optimistic, the cumulative regret over $T$ time steps satisfies*

$$R(T) \leq \sum_{t=1}^{T} \frac{1}{p_t} \mathbb{E}\left[U_t(X_t) - L_t(X_t) \mid \mathcal{H}_t\right] + \sum_{t=1}^{T} \left(U_t(X_t) - L_t(X_t)\right), \tag{31}$$

*with probability at least $1 - \delta$.*

**Interpretation**    The regret bound in equation 31 decomposes into two main components:

1. **Exploration-Exploitation Trade-off:** The first term scales with $\frac{1}{p_t}$ and the expected width of the confidence bounds. A higher $p_t$ (i.e., greater optimism) reduces this component, promoting exploration.

2. **Confidence Bound Widths:** The second term aggregates the widths of the confidence intervals across all time steps, reflecting the uncertainty inherent in the agent's estimates.

For the regret to be sublinear in $T$, it is essential that the confidence bounds $U_t(a) - L_t(a)$ shrink appropriately as $t$ increases, ensuring that both terms grow slower than linearly with $T$.

*Proof.* Let $X_t = \arg\max_{a \in \mathcal{X}_t} \tilde{f}_t(a)$ be the action chosen by the agent and $A_t^* = \arg\max_{a \in \mathcal{X}_t} f^*(a)$ be the optimal action at time $t$. Let $B_t = \max_{a \in \mathcal{X}_t} L_t(a)$, which is $\mathcal{H}_t$-measurable.

Conditioned on the event $\mathcal{E} \cap \tilde{\mathcal{E}}$, both $f^*(A_t^*) \geq B_t$ and $\tilde{f}_t(X_t) \geq B_t$ hold.

By p-optimism and the fact that $(f^*(A_t^*) - B_t)$ is $\mathcal{H}_t$-measurable and non-negative:

$$p_t \leq \mathbb{P}(\tilde{f}_t(X_t) - B_t \geq f^*(A_t^*) - B_t \mid \mathcal{H}_t)$$

$$\overset{(*)}{\leq} \mathbb{E}[\tilde{f}_t(X_t) - B_t \mid \mathcal{H}_t]/(f^*(A_t^*) - B_t),$$

where $(*)$ is due to Markov's inequality.

Rearranging and using the fact $B_t \geq L_t(X_t)$ yields:

$$f^*(A_t^*) - \tilde{f}_t(X_t) \leq f^*(A_t^*) - B_t \tag{32}$$

$$\leq \frac{1}{p_t} \mathbb{E}[\tilde{f}_t(X_t) - B_t \mid \mathcal{H}_t] \tag{33}$$

$$\leq \frac{1}{p_t} \mathbb{E}[U_t(X_t) - L_t(X_t) \mid \mathcal{H}_t]. \tag{34}$$

By reasonableness, $\tilde{f}_t(X_t) \leq U_t(X_t)$. Then, from the definition of confidence bounds:

$$\tilde{f}_t(X_t) - f^*(X_t) \leq U_t(X_t) - L_t(X_t) \tag{35}$$

Putting equation 32 and equation 35 together, and then summing over the time index $t$ yields the general regret upper bound. $\qquad\square$

### D.2  Proof of Theorem 4.3 for linear contextual bandits

To make the proof easy to access, we restate the core results and a few notations that are needed for the proof of the propositions.

Table 4: The coefficients $\rho(P_\zeta)$ and $p(P_\zeta)$ related to reasonableness and optimism conditions. Restated from Table 1 with evidence from Appendix E.

| $P_\zeta$ | Gaussian $\mathcal{N}(0, I_M)$ | Sphere $\sqrt{M}\mathcal{U}(\mathbb{S}^{M-1})$ | Cube $\mathcal{U}(\{1, -1\}^M)$ | Coord $\sqrt{M}\mathcal{U}(\{\pm e_i\}_{i \in [M]})$ | Sparse |
|---|---|---|---|---|---|
| $\rho(P_\zeta)$ | $\rho_1 \wedge \rho_3$ | $\rho_2 \wedge \rho_3$ | $\rho_2 \wedge \rho_3$ | $\rho_2$ | $\rho_2$ |
| $p(P_\zeta)$ | $\frac{1}{4\sqrt{e\pi}}$ | $\frac{1}{2} - \frac{e^{1/12}}{\sqrt{2\pi}}$ | $7/32$ | $\frac{1}{2M}$ | N/A |

Adapting the results from [Abbasi-Yadkori et al., 2011b, Abeille and Lazaric, 2017], let $\beta_t = \sqrt{\lambda} + \sqrt{2\log(1/\delta) + \log\det(\Sigma_{t-1}^{-1}/\lambda^d)}$. Under Assumption 4.1, we define the confidence bounds as:

$$L_t(\cdot) = (-1) \vee (\langle \mu_{t-1}, \phi(\cdot) \rangle - \beta_t \|\phi(\cdot)\|_{\Sigma_{t-1}}),$$
$$U_t(\cdot) = 1 \wedge (\langle \mu_{t-1}, \phi(\cdot) \rangle + \beta_t \|\phi(\cdot)\|_{\Sigma_{t-1}})$$

For the purpose of analysis with various reference distributions, we define slightly inflated confidence bounds:

$$L_t(\cdot; P_\zeta) = (\langle \mu_{t-1}, \phi(\cdot) \rangle - \beta_t \rho(P_\zeta) \|\phi(\cdot)\|_{\Sigma_{t-1}}) \vee (-1),$$
$$U_t(\cdot; P_\zeta) = (\langle \mu_{t-1}, \phi(\cdot) \rangle + \beta_t \rho(P_\zeta) \|\phi(\cdot)\|_{\Sigma_{t-1}}) \wedge 1.$$

Here, $\rho(P_\zeta)$ is defined via:

$$\rho_1 = O(\sqrt{M \log(M/\delta)}),$$
$$\rho_2 = O(\sqrt{M}),$$
$$\rho_3 = O(\sqrt{\log(|\mathcal{X}|/\delta)})$$

and Table 4.

An immediate observation is that $[L_t(\cdot), U_t(\cdot)] \subset [L_t(\cdot; P_\zeta), U_t(\cdot; P_\zeta)]$. Thus, $L_t(\cdot; P_\zeta)$ and $U_t(\cdot; P_\zeta)$ are also valid confidence bounds.

We consider the following functional form for Ensemble++ under linear setup: for time $t$,

$$\tilde{f}_t(a) := f_{\theta_t}(a, \zeta_t) = \langle \phi(a), \beta_t \mathbf{A}_{t-1} \zeta_t + \mu_{t-1} \rangle, \quad \forall a \in \mathcal{X}, \tag{36}$$

where the parameters include $\theta_t = (\mathbf{A}_t, \mu_t)$.

The condition on the propositions and theorem for regret analysis is when equation 9 is satisfied, that is when $M = \Theta(d \log T)$, the Lemma 4.2 implies that with high probability, the good events $\mathcal{G} = \bigcap_{t=0}^{T} \mathcal{G}_t$ hold jointly, where

$$\mathcal{G}_t := \left\{ \frac{1}{2} x^\top \Sigma_t x \leq x^\top \mathbf{A}_t \mathbf{A}_t^\top x \leq \frac{3}{2} x^\top \Sigma_t x, \quad \forall x \in \mathbb{R}^d \right\}.$$

In the following section, we discuss the proof conditioned on the joint event $\mathcal{G}$ and also the confidence event that $f^*(a) \in [\mathrm{L}_t(a), \mathrm{U}_t(a)]$ for all $t \in [T]$ and $a \in \mathcal{X}$.

### D.2.1 Proof of Proposition D.5

**Proposition D.5.** *Under linear setups in equation 36 and equation 4, if equation 9 is satisfied, linear Ensemble++ is reasonable, i.e., $\forall t \in [T]$, $\tilde{f}_t(\cdot) = f_{\theta_t}(\cdot, \zeta_t) \in [\mathrm{L}_t(\cdot; P_\zeta), \mathrm{U}_t(\cdot; P_\zeta)]$ with probability at least $1 - \delta$.*

*Proof.* From equation 36, we derive:

$$\begin{aligned}
|\tilde{f}_t(a) - \langle \mu_{t-1}, \phi(a) \rangle| &= |\langle \phi(a), \beta_t \mathbf{A}_{t-1} \zeta_t \rangle| \\
&= \beta_t \sqrt{\phi(a)^\top \mathbf{A}_{t-1} \mathbf{A}_{t-1}^\top \phi(a)} \left| \left\langle \frac{\phi(a)^\top \mathbf{A}_{t-1}}{\|\phi(a)^\top \mathbf{A}_{t-1}\|}, \zeta_t \right\rangle \right| \\
&\leq (3/2) \beta_t \sqrt{\phi(a)^\top \Sigma_{t-1} \phi(a)} \left| \left\langle \frac{\phi(a)^\top \mathbf{A}_{t-1}}{\|\phi(a)^\top \mathbf{A}_{t-1}\|}, \zeta_t \right\rangle \right|,
\end{aligned}$$

where the last inequality is due to the good event $\mathcal{G}$.

For compact action sets, by the Cauchy–Schwarz inequality:

$$\left| \left\langle \frac{\phi(a)^\top \mathbf{A}_{t-1}}{\|\phi(a)^\top \mathbf{A}_{t-1}\|}, \zeta_t \right\rangle \right| \leq \|\zeta_t\|.$$

Using the concentration properties of $P_\zeta$ from Section E, we can bound $\|\zeta_t\|$:

- For Gaussian distribution: $\|\zeta_t\| \leq O(\sqrt{M \log(M/\delta)})$ with probability at least $1 - \delta$
- For Sphere, Cube, Coordinate, and Sparse distributions: $\|\zeta_t\| = \sqrt{M}$ by construction

For finite action sets $\mathcal{X}$, we leverage the concentration properties to bound:

$$\mathbb{P}\left( \left| \left\langle \frac{\phi(a)^\top \mathbf{A}_{t-1}}{\|\phi(a)^\top \mathbf{A}_{t-1}\|}, \zeta_t \right\rangle \right| \leq \sqrt{\log \frac{2|\mathcal{X}|}{\delta}} \,\Big|\, \mathcal{H}_t, \mathcal{Z}_t \right) \geq 1 - \delta,$$

since $\zeta_t$ is independent of the history $\mathcal{H}_t$ and $\mathcal{Z}_t$.

Combining these bounds, we have:

$$|\tilde{f}_t(a) - \langle \mu_{t-1}, \phi(a) \rangle| \leq \beta_t \rho(P_\zeta) \|\phi(a)\|_{\Sigma_{t-1}}$$

where $\rho(P_\zeta)$ is defined in Table 4 and incorporates both the factor from the good event and the appropriate concentration bound for each distribution.

Therefore, with probability at least $1 - \delta$:

$$\langle \mu_{t-1}, \phi(a) \rangle - \beta_t \rho(P_\zeta) \|\phi(a)\|_{\Sigma_{t-1}} \leq \tilde{f}_t(a) \leq \langle \mu_{t-1}, \phi(a) \rangle + \beta_t \rho(P_\zeta) \|\phi(a)\|_{\Sigma_{t-1}}$$

After applying the truncation to $[-1, 1]$, we conclude that $\tilde{f}_t(a) \in [\mathrm{L}_t(a; P_\zeta), \mathrm{U}_t(a; P_\zeta)]$ with probability at least $1 - \delta$, which establishes the reasonableness of linear Ensemble++. $\qquad \square$

### D.2.2 Proof of Proposition D.6

**Proposition D.6.** *Under linear setups in equation 36 and equation 4, if equation 9 is satisfied, linear Ensemble++ using reference distribution $P_\zeta$ is $p(P_\zeta)$-optimistic.*

*Proof.* Let $X_t = \arg\max_{a \in \mathcal{X}_t} \tilde{f}_t(a)$ and $A_t^* = \arg\max_{a \in \mathcal{X}_t} f^*(a)$. Conditioned on $\mathcal{G}$ and the confidence event:

$$
\begin{aligned}
\tilde{f}_t(X_t) - f^*(A_t^*) &\geq \tilde{f}_t(A_t^*) - f^*(A_t^*) \\
&\geq \tilde{f}_t(A_t^*) - \mathrm{U}_t(A_t^*) \\
&= \langle \phi(A_t^*), \beta_t \mathbf{A}_{t-1}\zeta_t + \mu_{t-1} \rangle - (\langle \mu_{t-1}, \phi(A_t^*) \rangle + \beta_t \|\phi(A_t^*)\|_{\Sigma_{t-1}}) \\
&= \langle \phi(A_t^*), \beta_t \mathbf{A}_{t-1}\zeta_t \rangle - \beta_t \|\phi(A_t^*)\|_{\Sigma_{t-1}} \\
&= \beta_t \sqrt{\phi(A_t^*)^\top \mathbf{A}_{t-1}\mathbf{A}_{t-1}^\top \phi(A_t^*)} \left\langle \frac{\phi(A_t^*)^\top \mathbf{A}_{t-1}}{\|\phi(A_t^*)^\top \mathbf{A}_{t-1}\|}, \zeta_t \right\rangle - \beta_t \|\phi(A_t^*)\|_{\Sigma_{t-1}} \\
&\geq \beta_t \|\phi(A_t^*)\|_{\Sigma_{t-1}} \left( \frac{1}{\sqrt{2}} \left\langle \frac{\phi(A_t^*)^\top \mathbf{A}_{t-1}}{\|\phi(A_t^*)^\top \mathbf{A}_{t-1}\|}, \zeta_t \right\rangle - 1 \right),
\end{aligned}
$$

where we used the good event $\mathcal{G}$ to relate $\mathbf{A}_{t-1}\mathbf{A}_{t-1}^\top$ to $\Sigma_{t-1}$.

Therefore, the agent is optimistic if:

$$
\left\langle \frac{\phi(A_t^*)^\top \mathbf{A}_{t-1}}{\|\phi(A_t^*)^\top \mathbf{A}_{t-1}\|}, \zeta_t \right\rangle \geq \sqrt{2}
$$

We consider the conditional probability:

$$
\mathbb{P}(\tilde{f}_t(X_t) \geq f^*(A_t^*) \mid \mathcal{H}_t, \mathcal{Z}_t) \geq \mathbb{P}\left( \left\langle \frac{\phi(A_t^*)^\top \mathbf{A}_{t-1}}{\|\phi(A_t^*)^\top \mathbf{A}_{t-1}\|}, \zeta_t \right\rangle \geq \sqrt{2} \mid \mathcal{H}_t, \mathcal{Z}_t \right) \tag{37}
$$

Since $v = \frac{\phi(A_t^*)^\top \mathbf{A}_{t-1}}{\|\phi(A_t^*)^\top \mathbf{A}_{t-1}\|}$ is a fixed unit vector in $\mathbb{R}^M$ (given $\mathcal{H}_t$ and $\mathcal{Z}_t$), we need the probability $\mathbb{P}(\langle v, \zeta_t \rangle \geq \sqrt{2})$.

By the anti-concentration properties derived in Section E and scaling arguments, we know that $\mathbb{P}(\langle v, \zeta_t \rangle \geq 1) \geq p(P_\zeta)$ for the distributions in Table 4. Since $\mathbb{P}(\langle v, \zeta_t \rangle \geq \sqrt{2}) \leq \mathbb{P}(\langle v, \zeta_t \rangle \geq 1)$, we have:

$$
\mathbb{P}(\tilde{f}_t(X_t) \geq f^*(A_t^*) \mid \mathcal{H}_t, \mathcal{Z}_t) \geq p(P_\zeta)
$$

This establishes that linear Ensemble++ with reference distribution $P_\zeta$ is $p(P_\zeta)$-optimistic. $\qquad\square$

### D.3 Proof of Theorem 4.3

*Proof.* The theorem follows directly from Propositions D.5, D.6, and Theorem D.4.

First, by Proposition D.5, linear Ensemble++ is reasonable with respect to the inflated confidence bounds $[\mathrm{L}_t(\cdot; P_\zeta), \mathrm{U}_t(\cdot; P_\zeta)]$.

Second, by Proposition D.6, linear Ensemble++ is $p(P_\zeta)$-optimistic.

Applying Theorem D.4 with these properties:

$$
\begin{aligned}
R(T) &\leq \sum_{t=1}^{T} \frac{1}{p(P_\zeta)} \mathbb{E}[\mathrm{U}_t(X_t; P_\zeta) - \mathrm{L}_t(X_t; P_\zeta) \mid \mathcal{H}_t] + \sum_{t=1}^{T} (\mathrm{U}_t(X_t; P_\zeta) - \mathrm{L}_t(X_t; P_\zeta)) \\
&\leq \frac{1}{p(P_\zeta)} \sum_{t=1}^{T} \mathbb{E}[2\beta_t \rho(P_\zeta)\|\phi(X_t)\|_{\Sigma_{t-1}} \mid \mathcal{H}_t] + \sum_{t=1}^{T} 2\beta_t \rho(P_\zeta)\|\phi(X_t)\|_{\Sigma_{t-1}} \tag{38}
\end{aligned}
$$

Additionally, by Azuma's inequality for the sum of bounded martingale differences (since $\mathrm{U}_t(\cdot) - \mathrm{L}_t(\cdot) \leq 2$ is bounded):

$$
\sum_{t \in [T]} \mathbb{E}[(\mathrm{U}_t(X_t; P_\zeta) - \mathrm{L}_t(X_t; P_\zeta)) \mid \mathcal{H}_t] - (\mathrm{U}_t(X_t; P_\zeta) - \mathrm{L}_t(X_t; P_\zeta)) \leq O(\sqrt{T \log(1/\delta)}),
$$

with probability at least $1 - \delta$.

Thus, it suffices to bound:

$$\sum_{t=1}^{T} 2\beta_t \rho(P_\zeta) \|\phi(X_t)\|_{\Sigma_{t-1}} \leq 2\rho(P_\zeta) \sqrt{T \sum_{t=1}^{T} \beta_t^2 \|\phi(X_t)\|_{\Sigma_{t-1}}^2}$$

$$\leq 2\rho(P_\zeta) \sqrt{T \cdot 2\beta_T^2 \log \frac{\det \Sigma_T}{\det \Sigma_0}} \tag{39}$$

where the first inequality is by Cauchy-Schwarz and the second applies the elliptical potential lemma (Lemma 19.4 in [Lattimore and Szepesvári, 2020] and [Abbasi-Yadkori et al., 2011a]).

Since

$$\beta_T \leq \sqrt{\lambda} + \sqrt{2 \log(1/\delta) + d \log \left( 1 + \frac{T}{d\lambda} \right)}$$

and

$$\log \frac{\det \Sigma_T}{\det \Sigma_0} \leq d \log \left( 1 + \frac{T}{d\lambda} \right),$$

we have:

$$\sum_{t=1}^{T} 2\beta_t \rho(P_\zeta) \|\phi(X_t)\|_{\Sigma_{t-1}} \leq O \left( \rho(P_\zeta) \sqrt{T \cdot \left( \log(1/\delta) + d \log \frac{T}{d\lambda} \right) \cdot d \log \frac{T}{d\lambda}} \right)$$

$$= O \left( \rho(P_\zeta) \sqrt{d^2 T \log^2(T)} \right)$$

$$= O \left( \rho(P_\zeta) d \sqrt{T} \log(T) \right) \tag{40}$$

**General case** Combining all terms, the regret bound becomes:

$$R(T) \leq O \left( \frac{\rho(P_\zeta)}{p(P_\zeta)} d\sqrt{T} \log(T) + \rho(P_\zeta) d\sqrt{T} \log(T) + \frac{1}{p(P_\zeta)} \sqrt{T \log(1/\delta)} \right)$$

$$= O \left( \left( \frac{\rho(P_\zeta)}{p(P_\zeta)} + \rho(P_\zeta) \right) d\sqrt{T} \log(T) \right) \tag{41}$$

**Compact action sets** Based on Table 4, for all distributions except Coordinate, $\rho(P_\zeta) = O(\sqrt{d \log T})$ (since $M = \Theta(d \log T)$) and $\frac{1}{p(P_\zeta)} = O(1)$. For the Coordinate distribution, $\frac{1}{p(P_\zeta)} = O(M) = O(d \log T)$.

Therefore, for Gaussian, Sphere, and Cube distributions:

$$R(T) = O(d^{3/2} \sqrt{T} \log^{3/2} T)$$

And for the Coordinate distribution:

$$R(T) = O(d^{5/2} \sqrt{T} \log^{5/2} T)$$

**Finite action sets** For finite action sets $|\mathcal{X}|$, we can achieve an improved regret bound. Recall from Table 4 that for Gaussian, Sphere, and Cube distributions, $\rho(P_\zeta) = \rho_1 \wedge \rho_3$ or $\rho_2 \wedge \rho_3$, where $\rho_3 = O(\sqrt{\log(|\mathcal{X}|/\delta)})$.

When $|\mathcal{X}|$ is not too large, specifically when $\log |\mathcal{X}| \ll d \log T$, we have $\rho_3 \ll \rho_1$ and $\rho_3 \ll \rho_2$. In this case:

$$\rho(P_\zeta) = O(\sqrt{\log(|\mathcal{X}|/\delta)}) \tag{42}$$

Substituting this into our regret bound:

$$R(T) = O \left( \left( \frac{\sqrt{\log(|\mathcal{X}|/\delta)}}{p(P_\zeta)} + \sqrt{\log(|\mathcal{X}|/\delta)} \right) d\sqrt{T} \log(T) \right) \tag{43}$$

$$= O \left( \frac{\sqrt{\log |\mathcal{X}|}}{p(P_\zeta)} \cdot d\sqrt{T} \log(T) \right) \tag{44}$$

For Gaussian, Sphere, and Cube distributions, where $\frac{1}{p(P_\zeta)} = O(1)$, this gives:

$$R(T) = O(d\sqrt{T \log |\mathcal{X}|} \log(T))$$

This represents a significant improvement over the bound for compact action sets when $|\mathcal{X}|$ is small, as the dependence on dimension changes from $O(d^{3/2})$ to $O(d)$.

This completes the proof. $\qquad\square$

# E   Sampling, Isotropy, Concentration and Anti-concentration

This appendix analyzes various probability distributions used in the Ensemble++ algorithm, focusing on their isotropy, concentration, and anti-concentration properties. We begin by clarifying the relationship between two key distributions in our framework.

In Ensemble++, we utilize two related yet distinct distributions:

1. **Reference Distribution** ($P_\zeta$): This is the base distribution from which we sample M-dimensional vectors $\zeta \in \mathbb{R}^M$. We analyze several options for this distribution, including Gaussian, Sphere, Cube, Coordinate, and Sparse.

2. **Perturbation Distribution** ($P_{\mathbf{z}}$): This distribution is derived from $P_\zeta$ (which exhibits 1-sub-Gaussianity) through a normalization process. For a vector $\zeta$ independently sampled from $P_\zeta$, the corresponding perturbation vector $\mathbf{z}$ is obtained as $\mathbf{z} = \zeta/\|\zeta\|_2$.

This normalization ensures that all perturbation vectors $\mathbf{z}$ have a unit norm (i.e., $\|\mathbf{z}\|_2 = 1$). This property is a crucial requirement for our theoretical analysis, specifically for Lemma 4.2. The characteristics of the reference distribution $P_\zeta$ directly influence the ensemble size $M$ required to achieve our theoretical guarantees.

The choice of reference distribution affects both statistical efficiency and computational properties of our algorithm. Different reference distributions lead to different constants in our theoretical bounds while offering various practical advantages.

**Definition E.1** (Isotropic). A distribution $P$ over $\mathbb{R}^M$ is called isotropic if $\mathbb{E}_{X \sim P}[X_i X_j] = \delta_{ij}$, i.e., $\mathbb{E}_{X \sim P}[XX^\top] = I$. Equivalently, $P$ is isotropic if $\mathbb{E}_{X \sim P}[\langle X, x \rangle^2] = \|x\|^2$, for all $x \in \mathbb{R}^M$.

The isotropy property (Definition E.1) is used for the update distribution and in proving equation 4. The sub-Gaussianity (Definition C.2) in concentration property is used for perturbation distributions and proving Lemma 4.2. The concentration and anti-concentration properties are used for reference distributions and in the discussion on the reasonableness condition (Proposition D.5) and optimism condition (Proposition D.6).

We now analyze each distribution case by case.

## E.1   Sphere Distribution $P_\zeta = \mathcal{U}(\sqrt{M}\mathbb{S}^{M-1})$

---
**Algorithm 3** Sampling from $\mathcal{U}(\sqrt{M}\mathbb{S}^{M-1})$

---
**Require:** Number of ensemble members $M$
1: Sample vector $v$: $v_i \sim \mathcal{N}(0, 1)$ for $i = 1, \ldots, M$
2: Construct vector: $\xi = \sqrt{M}v/\|v\|$
3: **Return** $\xi$

---

**Isotropy**. By the rotational invariance of the sphere distribution, for any fixed orthogonal matrix $Q$:

$$\langle \zeta, x \rangle \sim \langle Q\zeta, x \rangle = \langle \zeta, Q^\top x \rangle, \quad \forall x \in \mathbb{R}^d.$$

Then, for any fixed $x$, we can select $M$ orthogonal matrices $Q_1, \ldots, Q_M$ to rotate $x$ such that $Q_i^\top x = \|x\|e_i$ where $e_i$ is the $i$-th coordinate vector. With this construction, for any fixed $x$:

$$M\mathbb{E}[\langle \zeta, x \rangle^2] = \mathbb{E}[\sum_{i=1}^{M} \langle \zeta, x_i \rangle^2] = \mathbb{E}[\|x\|^2 \sum_{i=1}^{M} \zeta_i^2] = M\|x\|^2$$

and hence $\mathbb{E}[\langle \zeta, x \rangle^2] = \|x\|^2$, which is the definition of isotropy.

**Concentration**. By definition, $\|\zeta\| = \sqrt{M}$. For a random variable $\zeta \sim \mathcal{U}(\mathbb{S}^{M-1})$ and any fixed $v \in \mathbb{S}^{M-1}$, the inner product follows the transformed Beta distribution:

$$\langle \zeta, v \rangle \sim 2\,\text{Beta}\left(\frac{M-1}{2}, \frac{M-1}{2}\right) - 1.$$

As shown in Skorski [2023], Li [2024a], $P_\zeta = \mathcal{U}(\sqrt{M}\mathbb{S}^{M-1})$ is 1-sub-Gaussian. For a finite action set $\mathcal{A}$, using the concentration of Beta random variables with union bound:

$$\mathbb{P}\left(\forall a \in \mathcal{A}, \langle \zeta, \phi(a) \rangle \leq \|\phi(a)\|\sqrt{\log \frac{2|\mathcal{A}|}{\delta}}\right) \geq 1 - \delta.$$

**Anti-concentration**. Let's rewrite the problem in terms of the incomplete Beta function:

Given:

$$X \sim \text{Beta}\left(\frac{M-1}{2}, \frac{M-1}{2}\right)$$

We want:

$$\text{P}\left(\langle \zeta, v \rangle \geq 1\right) = \text{P}\left(2X - 1 > \frac{1}{\sqrt{M}}\right) = \text{P}\left(X > \frac{1}{2} + \frac{1}{2\sqrt{M}}\right).$$

**Theorem E.2.** *For all $d \geq 2$, the random variable $X \sim \text{Beta}\left(\frac{d-1}{2}, \frac{d-1}{2}\right)$ has the following anti-concentration behavior:*

$$P\left(X > \frac{1}{2} + \frac{1}{2\sqrt{d}}\right) \geq \frac{1}{2} - \frac{e^{1/12}}{\sqrt{2\pi}}.$$

**Remark E.3.** We did not find existing literature that provides such anti-concentration results for Beta distributions.

*Proof.* Using the incomplete Beta function $I_x(a, b)$, this probability can be expressed as:

$$\text{P}\left(X > \frac{1}{2} + \frac{1}{2\sqrt{d}}\right) = 1 - I_{\left(\frac{1}{2} + \frac{1}{2\sqrt{d}}\right)}\left(\frac{d-1}{2}, \frac{d-1}{2}\right)$$

For the regularized incomplete Beta function $I_x(a, b)$:

$$I_x(a, b) = \frac{B(x; a, b)}{B(a, b)}$$

where $B(x; a, b)$ is the incomplete Beta function and $B(a, b) := B(1; a, b)$ is the complete Beta function.

For $a = b = \frac{d-1}{2}$, the complete Beta function is:

$$B\left(\frac{d-1}{2}, \frac{d-1}{2}\right) = \frac{\Gamma\left(\frac{d-1}{2}\right)\Gamma\left(\frac{d-1}{2}\right)}{\Gamma(d-1)}$$

For the incomplete Beta function with $x = \frac{1}{2} + \frac{1}{2\sqrt{d}}$ and $a = b = \frac{d-1}{2}$:

$$B\left(\frac{1}{2} + \frac{1}{2\sqrt{d}}; \frac{d-1}{2}, \frac{d-1}{2}\right) = \int_0^{\frac{1}{2} + \frac{1}{2\sqrt{d}}} t^{\frac{d-3}{2}}(1-t)^{\frac{d-3}{2}}\,dt$$

Since $f(t) = t^{\frac{d-3}{2}}(1-t)^{\frac{d-3}{2}}$ is symmetric at $t = 1/2$ in the interval $[0, 1]$:

$$B\left(\frac{1}{2} + \frac{1}{2\sqrt{d}}; \frac{d-1}{2}, \frac{d-1}{2}\right) = \frac{1}{2}B\left(\frac{d-1}{2}, \frac{d-1}{2}\right) + \int_{\frac{1}{2}}^{\frac{1}{2} + \frac{1}{2\sqrt{d}}} t^{\frac{d-3}{2}}(1-t)^{\frac{d-3}{2}}\,dt.$$

As $f(t)$ achieves its maximum at $t = 1/2$, we can upper bound the incomplete Beta function:

$$\int_{\frac{1}{2}}^{\frac{1}{2}+\frac{1}{2\sqrt{d}}} t^{\frac{d-3}{2}} (1-t)^{\frac{d-3}{2}} \, dt \leq \left(\frac{1}{4}\right)^{\frac{d-3}{2}} \left(\frac{1}{2\sqrt{d}}\right) = \left(\frac{1}{2}\right)^{d-3} \left(\frac{1}{2\sqrt{d}}\right). \tag{45}$$

For the complete Beta function, we use Stirling's approximation for the Gamma function, which provides a strict lower bound [Nemes, 2015]:

$$\Gamma(z) \geq \sqrt{2\pi} z^{z-\frac{1}{2}} e^{-z},$$

and an upper bound [Gronwall, 1918]:

$$\Gamma(z) \leq \sqrt{2\pi} z^{z-\frac{1}{2}} e^{-z+\frac{1}{12z}}$$

for all $z > 0$.

This gives us the lower bound:

$$B\left(\frac{d-1}{2}, \frac{d-1}{2}\right) \geq \frac{\sqrt{2\pi}((d-1)/2)^{d-2} e^{-(d-1)}}{(d-1)^{d-\frac{3}{2}} e^{-d+1+\frac{1}{12(d-1)}}} = \sqrt{2\pi} \left(\frac{1}{2}\right)^{d-2} (d-1)^{-1/2} e^{-\frac{1}{12(d-1)}}.$$

Since $e^{-\frac{1}{12(d-1)}} \geq e^{-1/12}$ whenever $d \geq 2$:

$$B\left(\frac{d-1}{2}, \frac{d-1}{2}\right) \geq \sqrt{2\pi} e^{-1/12} \left(\frac{1}{2}\right)^{d-2} \frac{1}{\sqrt{d}}. \tag{46}$$

Combining equation 45 and equation 46:

$$I_{\left(\frac{1}{2}+\frac{1}{2\sqrt{d}}\right)}\left(\frac{d-1}{2}, \frac{d-1}{2}\right) \leq \frac{1}{2} + \frac{2e^{1/12}(\frac{1}{2\sqrt{d}})}{\sqrt{2\pi}\frac{1}{\sqrt{d}}}$$

$$= \frac{1}{2} + \frac{e^{1/12}}{\sqrt{2\pi}},$$

Therefore:

$$\mathbf{P}\left(X > \frac{1}{2} + \frac{1}{2\sqrt{d}}\right) \geq \frac{1}{2} - \frac{e^{1/12}}{\sqrt{2\pi}} \approx 0.0668.$$

$\square$

## E.2 Cube Distribution $P_\zeta = \mathcal{U}(\{1, -1\}^M)$

---

**Algorithm 4** Sampling from $\mathcal{U}(\{-1, 1\}^M)$

---

**Require:** Number of ensemble members $M$
1: Sample vector $\xi$: $\xi_i \sim \mathcal{U}(\{-1, 1\})$ for $i = 1, \dots, M$
2: **Return** $\xi$

---

**Isotropy**. Easy to verify by definition.

**Concentration**. By definition, $\|\xi\| = \sqrt{M}$. We sample the random vector $\zeta$ by independently sampling each entry from $\zeta_i \sim \mathcal{U}(\{1, -1\})$ for $i \in [M]$. Then, for any $v \in \mathbb{S}^{M-1}$, by independence:

$$\mathbb{E}[\exp(\lambda\langle v, \zeta\rangle)] = \prod_{i=1}^{M} \mathbb{E}[\exp(\lambda v_i \zeta_i)] \leq \prod_{i=1}^{M} \exp(\lambda^2 v_i^2) = \exp(\lambda^2 \sum_i v_i^2).$$

The inequality is due to the moment generating function of the Rademacher distribution [Wainwright, 2019]. This confirms that $P_\zeta = \mathcal{U}(\{1, -1\}^M)$ is 1-sub-Gaussian. For a finite action set $\mathcal{A}$, from the sub-Gaussian property:

$$\mathbb{P}\left(\forall a \in \mathcal{A}, \langle\zeta, \phi(a)\rangle \leq \|\phi(a)\|\sqrt{\log\frac{2|\mathcal{A}|}{\delta}}\right) \geq 1 - \delta.$$

**Anti-concentration**. Using the anti-concentration result from Hollom and Portier [2023], for any fixed unit vector $v$ in $\mathbb{R}^M$:

$$\mathbb{P}(\langle\zeta, v\rangle \geq 1) \geq 7/32 \approx 0.21875.$$

### E.3 Gaussian Distribution $P_\zeta = \mathcal{N}(0, I_M)$

---

**Algorithm 5** Sampling from $\mathcal{N}(0, I_M)$

---

**Require:** Number of ensemble members $M$
1: Sample vector $\xi$: $\xi_i \sim \mathcal{N}(0, 1)$ for $i = 1, \ldots, M$
2: **Return** $\xi$

---

**Isotropy**. Easy to verify by definition.

**Concentration**. The concentration property comes from the Chernoff bound for standard Gaussian random variables with the union bound. For any $\alpha > 0$:

$$\mathbb{P}(\|\zeta\| \leq \alpha\sqrt{M}) \geq \mathbb{P}(\forall 1 \leq i \leq M, |\zeta_i| \leq \alpha) \geq 1 - M\mathbb{P}(|\zeta_i| \geq \alpha).$$

Standard concentration inequality for Gaussian random variables gives, $\forall \alpha > 0$:

$$\mathbb{P}(|\zeta_i| \geq \alpha) \leq 2e^{-\alpha^2/2}.$$

With $\alpha = \sqrt{2\log\frac{2M}{\delta}}$:

$$\|\zeta\| \leq \sqrt{2M\log\frac{2M}{\delta}}, \quad \text{with probability at least } 1 - \delta.$$

For a finite action set $\mathcal{A}$:

$$\mathbb{P}\left(\forall a \in \mathcal{A}, \langle\zeta, \phi(a)\rangle \leq \|\phi(a)\|\sqrt{\log\frac{2|\mathcal{A}|}{\delta}}\right) \geq 1 - \delta.$$

**Anti-concentration**. For any fixed unit vector $v$ in $\mathbb{R}^M$, $\langle\zeta, v\rangle \sim \mathcal{N}(0, 1)$:

$$\mathbb{P}(\langle\zeta, v\rangle \geq 1) = \frac{1}{2}\text{erfc}\left(\frac{1}{\sqrt{2}}\right) \geq \frac{1}{4\sqrt{e\pi}} \approx 0.0856$$

### E.4 Coordinate Distribution $P_\zeta = \mathcal{U}(\sqrt{M}\{\pm e_1, \ldots, \pm e_M\})$

---

**Algorithm 6** Sampling from $\mathcal{U}(\sqrt{M}\{\pm e_1, \ldots, \pm e_M\})$

---

**Require:** Number of ensemble members $M$
1: Sample index: $i \sim \mathcal{U}(\{1, \ldots, M\})$
2: Sample sign: $s \sim \mathcal{U}(\{-1, 1\})$
3: Construct vector: $\xi = s\sqrt{M}e_i$ where $e_i$ is the $i$-th standard basis vector
4: **Return** $\xi$

---

**Isotropy**. By definition:

$$\mathbb{E}[\zeta\zeta^\top] = \frac{1}{2M}\sum_{i=1}^{M} 2Me_ie_i^\top = I.$$

**Concentration**. By definition, $\|\zeta\| = \sqrt{M}$.

**Anti-concentration**.

$$\mathbb{P}(\langle\zeta, v\rangle \geq 1) = \frac{1}{2M}\sum_{j\in[M]}\left(\mathbf{1}\left\{v_j \geq \frac{1}{\sqrt{M}}\right\} + \mathbf{1}\left\{-v_j \geq \frac{1}{\sqrt{M}}\right\}\right)$$

$$= \frac{1}{2M}\sum_{j\in[M]}\mathbf{1}\left\{|v_j| \geq \frac{1}{\sqrt{M}}\right\} \geq \frac{1}{2M},$$

where the last inequality uses the fact that for any $v \in \mathbb{R}^M$ with $\|v\| = 1$, there always exists at least one entry $j \in [M]$ with $|v_j| \geq \frac{1}{\sqrt{M}}$.

## E.5 Sparse Distribution $P_\zeta$

---
**Algorithm 7** Sampling from $s$-sparse random vector

---
**Require:** Number of ensemble members $M$, sparsity $s$
  1: Sample sign vector: $\omega_i \sim \mathcal{U}(\{-1,1\})$ for $i = 1,\ldots,M$
  2: Construct a set $\mathcal{S}$ by randomly picking $s$ elements from $\{1,\ldots,M\}$ without replacement
  3: Let $\eta_i = 1$ for $i \in \mathcal{S}$ and $\eta_{i'} = 0$ for $i' \in \{1,\ldots,M\} \setminus \mathcal{S}$
  4: Construct vector $\xi$: $\xi_i = \omega_i \cdot \eta_i \cdot \sqrt{\frac{M}{s}}$
  5: **Return** $\xi$

---

**Definition E.4** ($s$-sparse distribution)**.** The sparse vector is in the form $\zeta = \sqrt{\frac{M}{s}}\eta \odot \omega$ where $P_\omega := \mathcal{U}(\{1,-1\}^M)$, and $\eta$ is independently and uniformly sampled from all possible $s$-hot vectors, where an $s$-hot vector has exactly $s$ non-zero entries with value 1. This construction was introduced by Kane and Nelson [2014].

**Isotropy**. By definition:

$$\mathbb{E}[\zeta_j \zeta_k] = \frac{M}{s}\mathbb{E}[\eta_j \eta_k]\mathbb{E}[\omega_j \omega_k].$$

For $j \neq k$, $\mathbb{E}[\omega_j \omega_k] = 0$ since the signs are independent. For $j = k$, $\mathbb{E}[\omega_j^2] = 1$.

For $\eta$, we have $\mathbb{E}[\eta_j^2] = \frac{s}{M}$ since each coordinate has probability $\frac{s}{M}$ of being selected in the $s$-hot vector. For $j \neq k$, $\mathbb{E}[\eta_j \eta_k] = \frac{s(s-1)}{M(M-1)}$ since this is the probability that both coordinates are selected.

Therefore:

$$\begin{aligned}
\mathbb{E}[\zeta_j \zeta_k] &= \frac{M}{s}\mathbb{E}[\eta_j \eta_k]\mathbb{E}[\omega_j \omega_k] \\
&= \frac{M}{s}\mathbb{E}[\eta_j \eta_k]\delta_{jk} \\
&= \begin{cases} \frac{M}{s} \cdot \frac{s}{M} = 1 & \text{if } j = k \\ 0 & \text{if } j \neq k \end{cases} \\
&= \delta_{jk}
\end{aligned}$$

Thus, the sparse distribution in Definition E.4 is indeed isotropic.

**Concentration**. By construction, $\|\zeta\| = \sqrt{M}$.

**Anti-concentration**. The anti-concentration behavior of sparse distributions depends on the sparsity parameter $s$ and requires further investigation for precise bounds.

## E.6 Summary of Distribution Properties

Table 5 summarizes the key properties of each reference distribution $P_\zeta$ discussed in this appendix:

Table 5: Summary of Reference Distribution Properties

| Distribution | Isotropy | Sub-Gaussian | Anti-concentration | Computational Benefits |
|---|---|---|---|---|
| Sphere | Yes | $c_0 = 1$ | $\geq 0.0668$ | Standard implementation |
| Cube | Yes | $c_0 = 1$ | $\geq 0.21875$ | Simple sampling |
| Gaussian | Yes | $c_0 = 1$ | $\geq 0.0856$ | Standard implementation |
| Coordinate | Yes | Not characterized | $\geq 1/(2M)$ | Sparse matrix operations |
| $s$-sparse | Yes | Not characterized | Not characterized | $O(s)$ non-zero entries |

These properties directly influence the behavior of the derived perturbation distribution $P_z$ and ultimately affect the exploration-exploitation trade-off in the Ensemble++ algorithm. The choice of distribution can be tailored to the specific requirements of the application, balancing theoretical guarantees with computational efficiency.

# F  In-depth Empirical and Ablation Studies

In this section, we dive into the intricacies of each evaluation testbed. Through a comprehensive set of empirical results, we'll further illuminate the benefits afforded by Ensemble++. All experiments are conducted on P40 GPUs to maintain processing standardization.

## F.1  Additional Experiments on Linear Bandit

We begin by examining Linear Ensemble++ Sampling in linear bandits. In this section, we focus on studying the impact of perturbation and reference distributions, and we provide detailed results under varying numbers of ensembles $M$.

**Environment Settings.**  We use the action feature set $\mathcal{X}$ to denote the set of features $\phi(a) : a \in \mathcal{A}$ induced by action set $\mathcal{A}$ and feature mapping $\phi(\cdot)$. We build two linear bandit environments with different action distribution as follow:

- **Finite-action Linear Bandit**: We construct the finite set $\mathcal{X}$ by uniformly sampling a set of action features from the range $[-1/\sqrt{5}, 1/\sqrt{5}]^d$ where $d$ is the ambient dimension of the linear reward function. This environment builds upon prior research Russo and Van Roy [2018]. We vary the action size $|\mathcal{X}|$ over a set of $\{100, 1000, 10000\}$, and the ambient dimension across $\{10, 50\}$.

- **Compact-action Linear Bandit**: Let the action feature set $\mathcal{X} = \mathbb{S}^{d-1}$ be the unit sphere. In this environment, we vary the ambient dimension $d$ over a set of $\{10, 50, 100\}$.

In both bandits, the reward for feature $X_t \in \mathbb{R}^d$ is computed as $r_t = X_t^\top \theta + \epsilon$, where $\theta \sim \mathcal{N}(0, 10\mathrm{I})$ is drawn from the multivariate Gaussian prior distribution, and $\epsilon \sim \mathcal{N}(0, 1)$ is an independent additive Gaussian noise term. At each step $t$, only the reward from the chosen feature $X_t$ is discernible. To ensure robust results, each experiment is executed 1000 time steps and repeated 200 times.

**Impact of Reference and Perturbation Distributions.**  We investigated all 25 combinations of perturbation and reference distribution under different scales of the linear bandit environments and numerous #ensembles $M$. As depicted in Figs. 8 to 10, the outcomes across diverse problem scales corroborate each other. **The use of a Gaussian reference distribution significantly enhances performance when the $M$ is relatively small, such as when $M$ is 2 or 4.** As the #ensembles $M$ grows, all combinations show an analogous performance under varying problem scales. However, it is worth noting that for extremely large $M$, such as 512 or 1024, combinations involving the Coordinate perturbation and Coordinate reference distribution significantly underperform compared to other combinations. Given that Coordinate distributions are used in the Ensemble+, the results prompt a compelling argument. Linear Ensemble++ Sampling equipped with a continuous reference distribution presents a superior performance, suggesting its potential for surpassing traditional Linear Ensemble Sampling. These findings strongly support the superior advantage of our sampling method, validating our theoretical analysis.

**Analysis of Computational Efficiency.**  We delve deeper into the effects of varying #ensembles $M$ within Linear Ensemble++ Sampling. We assess its performance across different combinations of perturbation and reference distributions using an assortment of $M \in \{4, 8, 16, 32, 64, 128, 256, 512, 1024\}$. The outcomes, visualized in Figs. 11 and 12, are consistent with the findings illustrated in Figs. 8 to 10. We observe that for large $M$, the Coordinate perturbation and Coordinate reference distributions degrade performance, indicating that the sampling method employed by Ensemble+ lacks efficiency. However, when Linear Ensemble++ Sampling utilizes Gaussian or Sphere reference distributions, it achieves satisfactory performance, comparable to Thompson Sampling with small $M$.

> **Remark F.1** (Limitation of Theorem 4.3.). Notice that Theorem 4.3 suggest that when $M \geq O(d \log T)$, the regret bound of Linear Ensemble sampling would increase with factor $M^{3/2}$, which contradicts with our empirical evidence in Figs. 8 to 12.

> **Remark F.2** (Good prediction of Theorem 4.3.). Our empirical evidence in Figs. 8 to 12 confirms the Theorem 4.3 in finite decision set setting for continuous-support reference distributions: when $M$ is larger then a threshold $O(d \log T)$, the regret is independence on $M$.

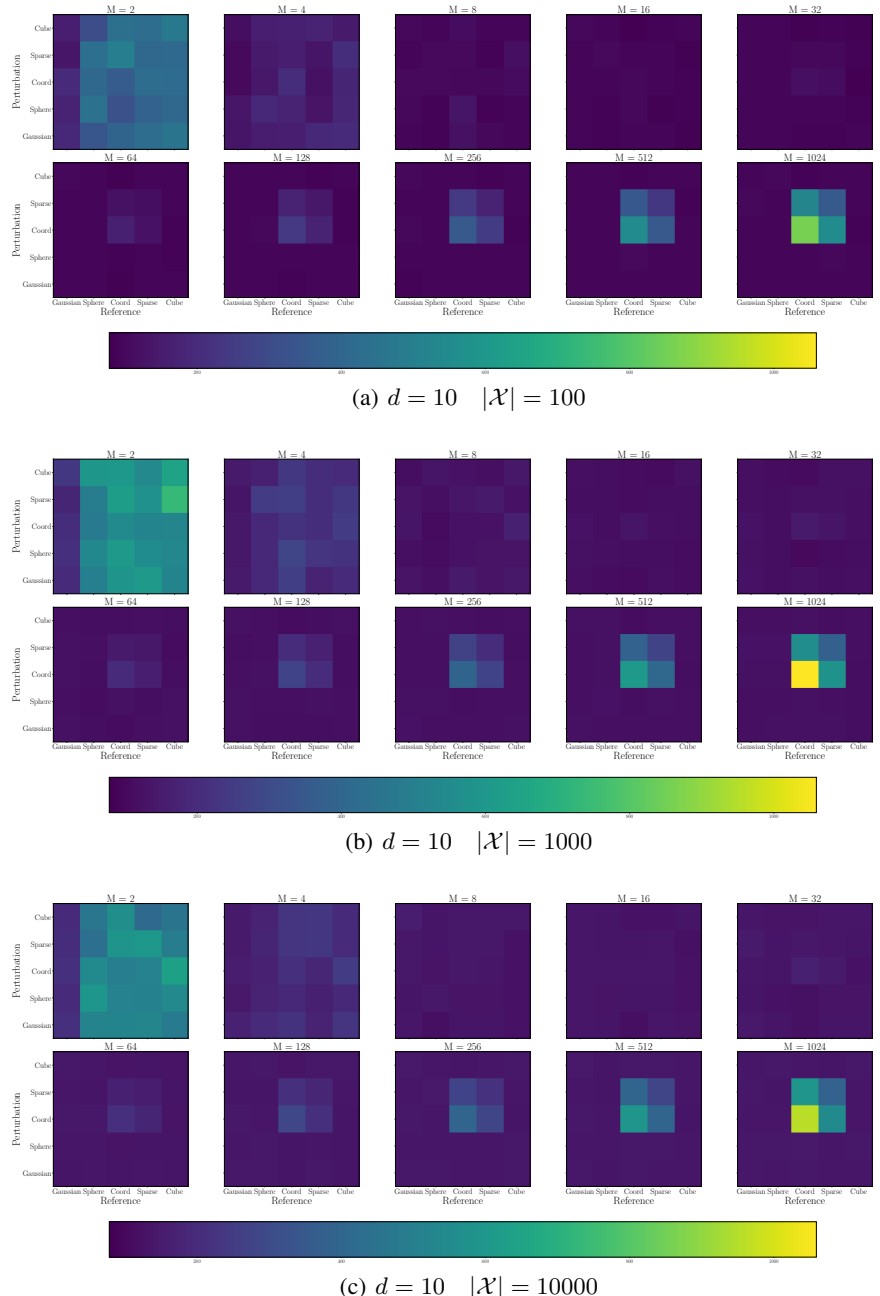

Figure 8: Results on the combinations of perturbation and reference distribution in Finite-action Linear Bandit under action dimension $d = 10$. A deeper color signifies lower accumulated regret and hence superior performance. Gaussian reference distribution significantly enhances performance.

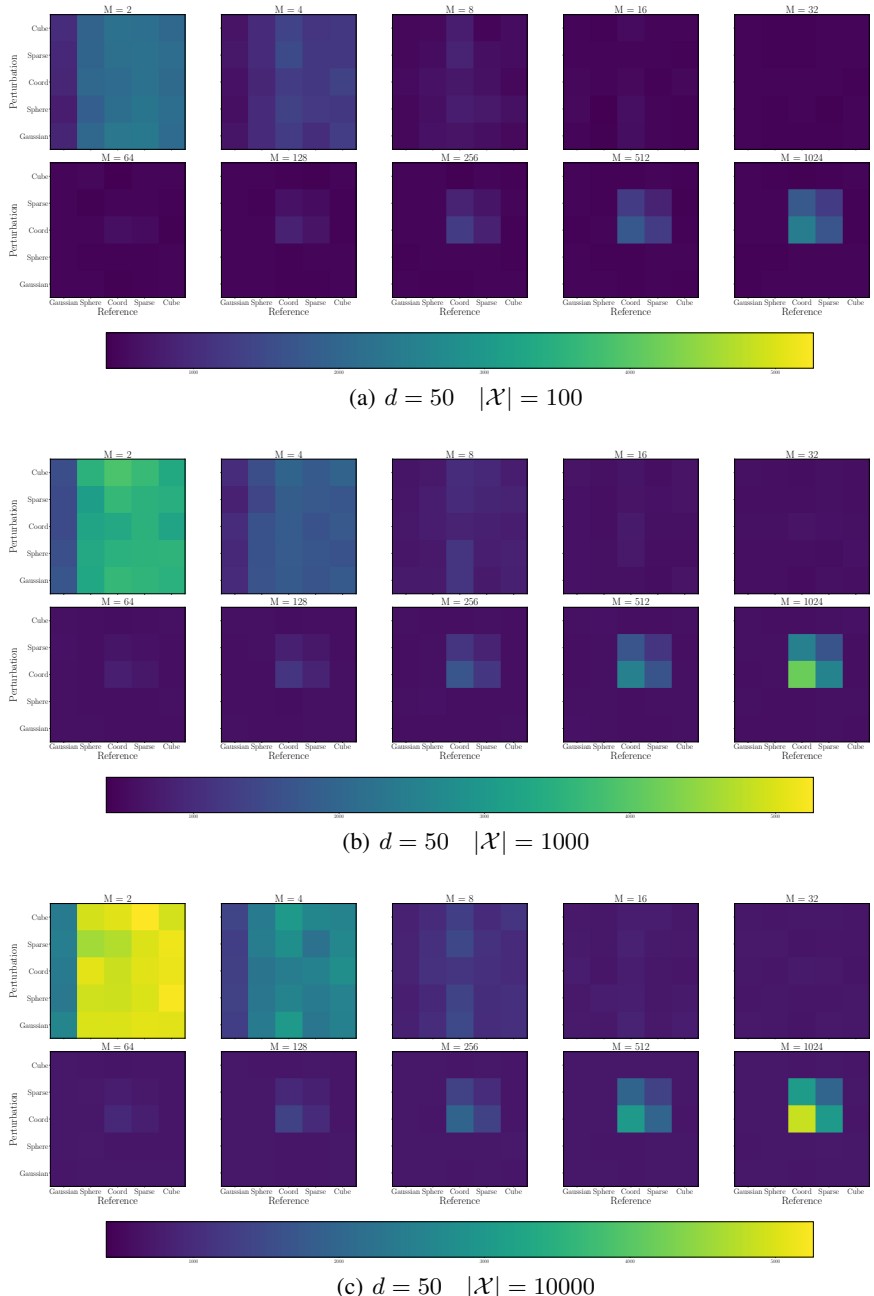

Figure 9: Results on the combinations of perturbation and reference distribution in Finite-action Linear Bandit under action dimension $d = 50$. A deeper color signifies lower accumulated regret and hence superior performance. Gaussian reference distribution significantly enhances performance.

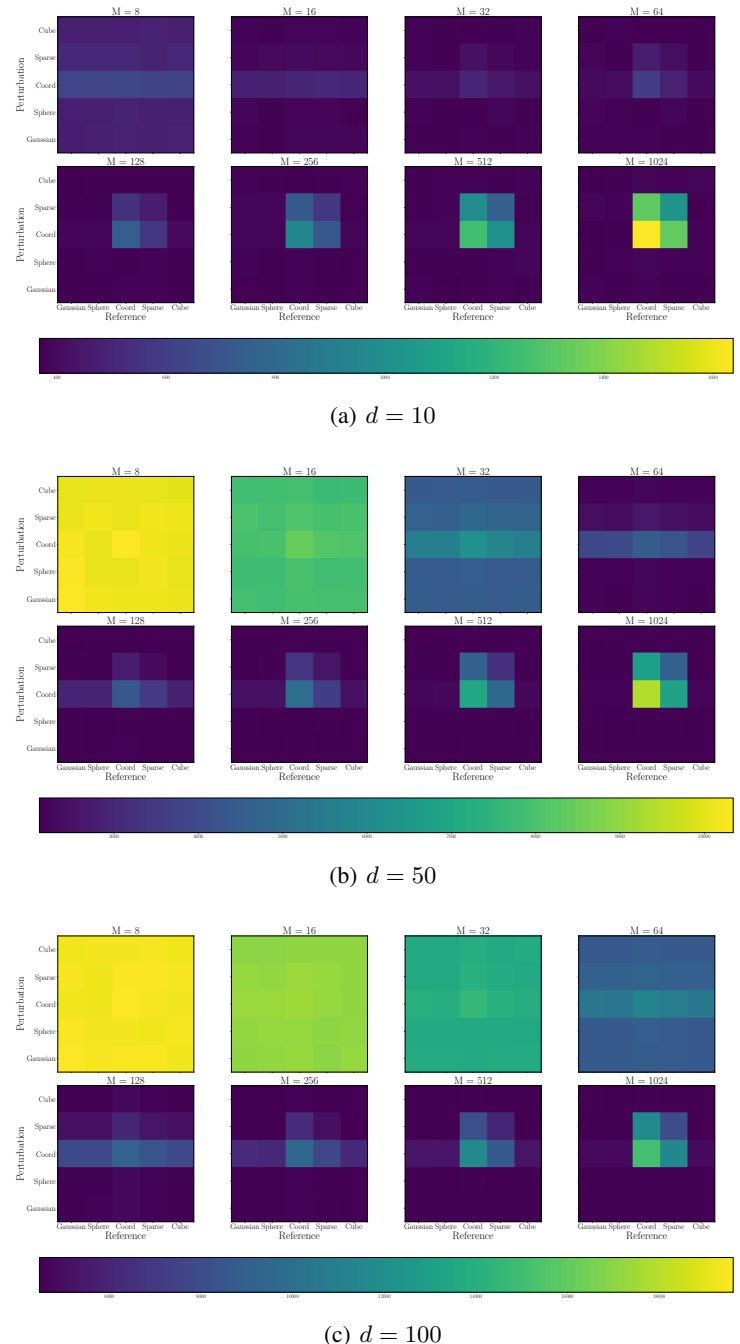

(a) $d = 10$

(b) $d = 50$

(c) $d = 100$

Figure 10: Results on the combinations of perturbation and reference distribution in Compact-action Linear Bandit. A deeper color signifies lower accumulated regret and hence superior performance. Gaussian reference distribution significantly enhances performance.

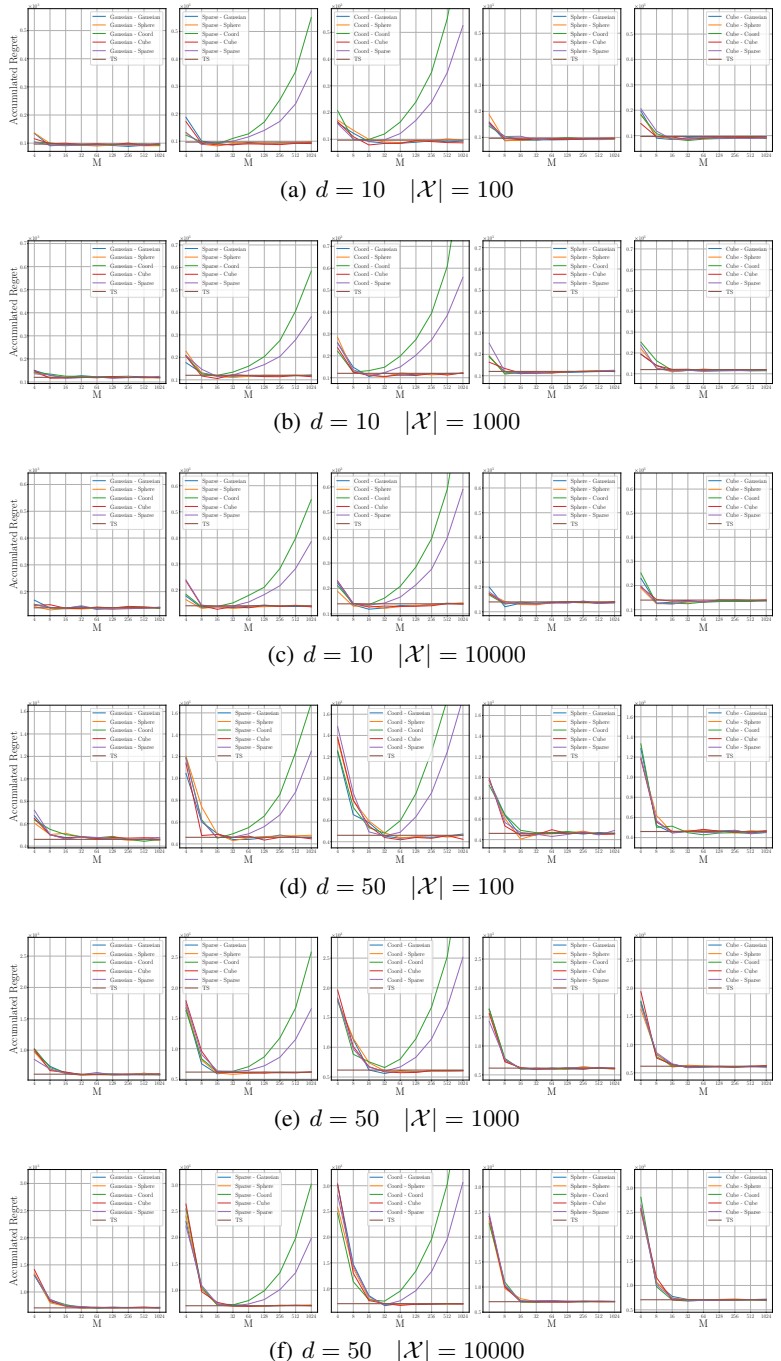

Figure 11: Results on regret under various #ensembles $M$ in Finite-action Linear Bandit. The label $A - B$ indicates that Ensemble++ uses A as the reference distribution and B as the perturbation distribution. Ensemble++ with Gaussian or Sphere reference distribution could achieve comparable performance with that of Thompson sampling under same $M$ for different action spaces $|\mathcal{X}|$.

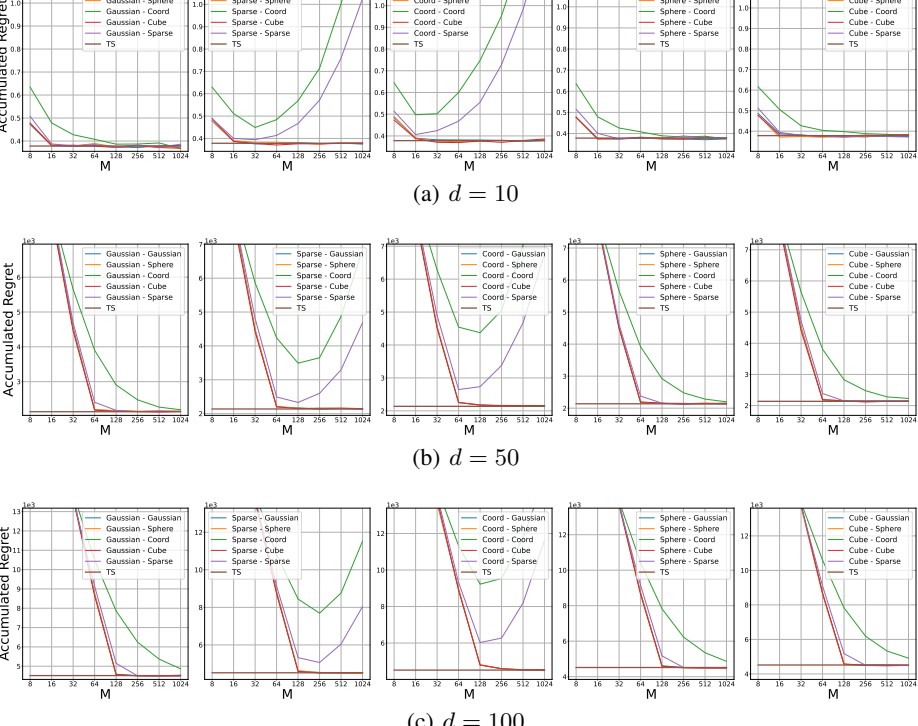

Figure 12: Results on regret under various #ensembles $M$ in Compact-action Linear Bandit. The label $A - B$ indicates that Ensemble++ uses A as the reference distribution and B as the perturbation distribution. Ensemble++ with Gaussian or Sphere reference distribution could achieve comparable performance with that of Thompson sampling under small $M$.

### F.2 Additional Experiments on Nonlinear Bandit

We conduct more comprehensive comparison of Ensemble++ with several baselines that utilize approximate posterior sampling across a wide range of nonlinear bandits.

**Environments Settings.** We formulate several nonlinear contextual bandit environments, with rewards generated by nonlinear functions in each.

- **Quadratic Bandit**: Its reward generation mechanism is built on a quadratic function, expressed as $f_1(x) = 10^{-2}(x^\top \Theta \Theta^\top x)$. Here, $x \in \mathbb{R}^d$ stands for the action, while $\Theta \in \mathbb{R}^{d \times d}$ is a matrix filled with random variables originating from $\mathcal{N}(0, 1)$. This task is used as the testbed in Zhou et al. [2020].

- **Vector Quadratic Bandit**: Its reward generation mechanism is built on a different quadratic function, expressed as $f_2(x) = 10(x^\top \theta)^2$. Here, $a \in \mathbb{R}^d$ stands for the action, while $\theta \in \mathbb{R}^d$ is a vector filled with random variables generated from a uniform distribution over the unit ball. This task is utilized as the testbed in Zhou et al. [2020], Xu et al. [2022].

- **Neural Bandit**: This bandit employs a nonlinear neural network built on 2-layer MLPs with 50 units and ReLU activations, producing two output logits. We apply the softmax function with a temperature parameter $p = 0.1$ to the two output logits to obtain probabilities. Subsequently, we use binomial sampling based on the second probability to generate the reward. The temperature parameter $p$ is used to control the signal-to-noise ratio. This task is used as the testbed in Osband et al. [2022, 2023a].

- **UCI Dataset**: Following prior works [Riquelme et al., 2018, Kveton et al., 2020b], we conduct contextual bandits with $N$-class classification using the UCI datasets [Asuncion et al., 2007] Mushroom and Shuttle. Specifically, given a data feature $x \in \mathbb{R}^d$ in the dataset, we construct context vectors for $N$ arms, such as $x^{(1)} = (x, 0, \cdots, 0), \cdots, x^{(N)} =$

$(0, 0, \cdots, x) \in \mathbb{R}^{Nd}$. Only the arm $x^{(j)}$ where $j$ matches the correct class of this data $x$ has a reward of 1, while all other arms have a reward of 0.

- **Online Hate Speech Detection**: The motivation, problem formulation and environment setups of the automated content moderation task are detailed in Appendix G.

In all tasks except the **Neural Bandit**, the original reward $r$ is disrupted by additive Gaussian noise $\epsilon$ drawn from $\mathcal{N}(0, 0.1)$. In the **Neural Bandit**, we use the temperature parameter $p$ to introduce noise into the reward. For the first three tasks, we set the action dimension $d$ to 100 and generate a total of 1000 candidated actions, randomly sampling 50 actions in each round. Each experiment is repeated with 10 distinct random seeds to ensure robust results.

**Comparison Results with Baselines.** We set the Sphere reference distribution, Coordinate update distribution, and Sphere perturbation distribution for Ensemble++ to compare with baselines. When comparing with Ensemble+ [Osband et al., 2018] and EpiNet [Osband et al., 2023a], we use the same hyperparameters, such as prior scale, learning rate, and batch size. Additionally, we employ the same network backbone for feature extraction to ensure fairness. As shown in Fig. 13(a) and (b), **Ensemble++ achieves sublinear regret and consistently outperforms these baselines across all tasks, demonstrating superior data efficiency.**

For comparison with LMCTS [Xu et al., 2022], we use its official implementation[5] to ensure credible results. As illustrated in Fig. 13(c), Ensemble++ consistently outperforms LMCTS. Notably, LMCTS uses the entire buffer data to update the network per step, which incurs significant computational costs. In contrast, **Ensemble++ achieves better performance with bounded computational steps, requiring only a minibatch to update the network.** As LMCTS was already demonstrated in its original work to outperform classical baselines (e.g., LinearUCB[Chu et al., 2011], NeuralUCB[Zhou et al., 2020]), we omit redundant validation of this result to avoid duplicating prior findings. These findings highlight the effective exploration and computational efficiency of Ensemble++.

**Additional Comparison on Trade-off between Regret and Computation.** We have demonstrated that Ensemble++ can achieve sublinear regret with moderate computational cost in the Quadratic Bandit, as shown in Fig. 1. Here, we further investigate the frontier relationship between regret and computation in the Neural Bandit. As shown in Fig. 14, we observe similar findings: Ensemble++ achieves minimal cumulative regret with the lowest computational cost. These results substantiate the scalability and efficiency of Ensemble++ when combined with neural networks.

**Ablation Studies on Storage Requirement.** As discussed in Section 3.3, Ensemble++ does not require storing the entire history. We tested different buffer capacity over 100,000 time steps (Fig. 15). Using smaller buffer leads to only a slight performance decrease, and Ensemble++ maintains comparable performance even with buffer capacity significantly smaller than the total time horizon.

**Ablation Study on Update, Reference and Perturbation Distribution.** To further evaluate the impact of different design of distributions, we perform an ablation study on the Quadratic Bandit. When fixing the Sphere reference distribution, we find that discrete update distributions such as Coordinate, Cube, and Sparse achieve similar better performance, as shown in Fig. 16(a). Conversely, when fixing the Coordinate update distribution, continuous reference distributions like Sphere and Gaussian also yield comparable better performance, as depicted in Fig. 16(b). Regarding the perturbation distribution, our findings indicate that it does not significantly influence performance when the neural network is involved in Ensemble++. This is evidenced in Fig. 16(c), where all different perturbation distributions achieve similar performance.

## G  Ensemble++ in Real-World Decision-Making: A Case Study on Content Moderation

This section demonstrates how *Ensemble++* can be integrated with large *foundation models* (e.g. GPTs) to address real-time decision-making under uncertainty. We focus on the high-stakes domain of **content moderation** on social media platforms, where rare or borderline hateful content arises frequently. By fusing GPT-style feature extraction with Ensemble++, uncertainty-driven sampling selectively allocates human review to ambiguous posts. This yields a scalable, adaptive pipeline that reduces moderator workload while improving overall safety and accuracy.

---

[5]`https://github.com/devzhk/LMCTS`

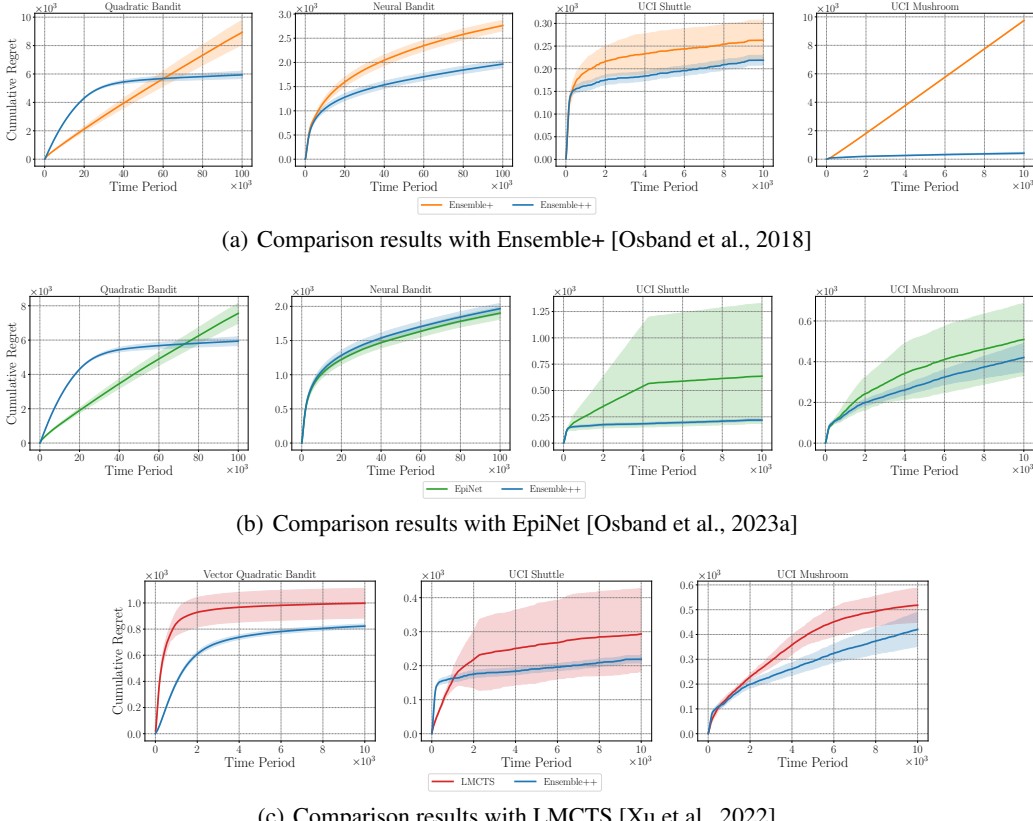

(a) Comparison results with Ensemble+ [Osband et al., 2018]

(b) Comparison results with EpiNet [Osband et al., 2023a]

(c) Comparison results with LMCTS [Xu et al., 2022]

Figure 13: Results on different bandits with various baselines. Ensemble++ could achieve better performance compared to other methods.

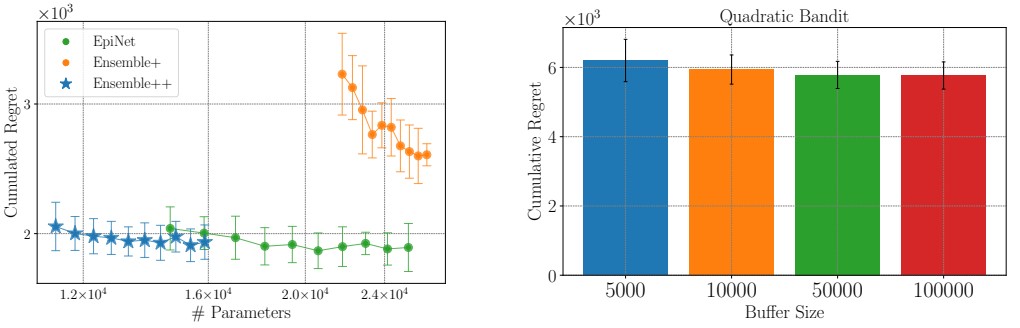

Figure 14: The regret-computation trade-off in Neural Bandit. Ensemble++ beats the SOTA baselines, e.g. Ensemble+ and EpiNet.

Figure 15: Performance of Ensemble++ under varying replay buffer capacity $C$.

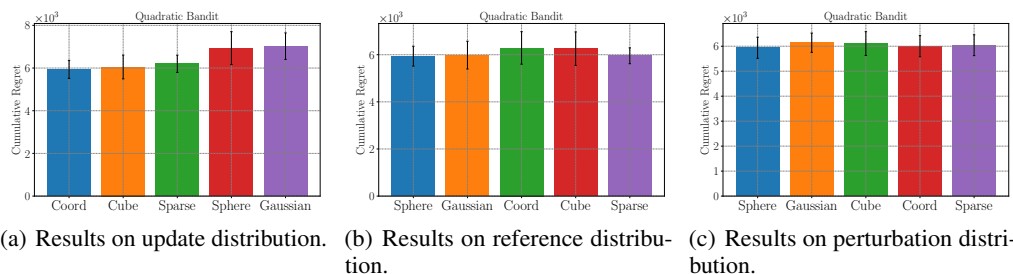

(a) Results on update distribution. (b) Results on reference distribution. (c) Results on perturbation distribution.

Figure 16: Ablation studies about different distributions on the Quadratic Bandit.

## G.1 Motivation: Content Moderation in Real-Time

Modern social-media platforms handle a vast volume of user-generated content every second, creating a *critical need* for automated moderation [Gorwa et al., 2020, Roberts, 2019]. Historically, human reviewers manually inspected each post to detect policy violations. However, as platforms like Facebook [Meta, 2024], Twitter [Corp., 2024], and Reddit [Reddit, 2024] expanded to hundreds of millions of users, *fully manual moderation* became infeasible. Consequently, *AI-driven moderation systems* emerged, often leveraging *foundation models* [Weng et al., 2023] (large pretrained language or vision models) for *real-time* filtering.

Despite robust performance on the distributions seen during training, these large models often face **high uncertainty** in *novel* or *rare* content: emergent slang, subtle or borderline hate speech, or newly formed harassment styles [Markov et al., 2023]. A purely *deterministic* policy (e.g., the model's single best guess) can err severely by

- **over-blocking** benign content (harmful to user experience), or
- **under-blocking** hateful material (a safety hazard).

Hence, **human feedback** remains indispensable for correcting the system, especially on ambiguous or boundary cases. The key dilemma is *when* to rely on human reviewers (which yields better learning but increases workload) versus *auto-removing* content (which saves labor but risks higher error).

**Human-AI Collaboration.** Figure 17 depicts a *human-in-the-loop* moderation pipeline:

1. A new post $x_t$ arrives.
2. The AI system either *auto-removes* it or *requests a human review*.
3. If reviewed, a moderator provides a corrective label $y_t$ (e.g., "hate" or "benign"), and the AI system updates its internal policy.
4. Over time, decisions become more accurate, reducing human intervention.

Balancing *exploitation* (avoiding unnecessary reviews) and *exploration* (gathering feedback on uncertain content) is central to improving moderation quality while minimizing reviewer workload. This tension aligns well with a *contextual bandit* formulation.

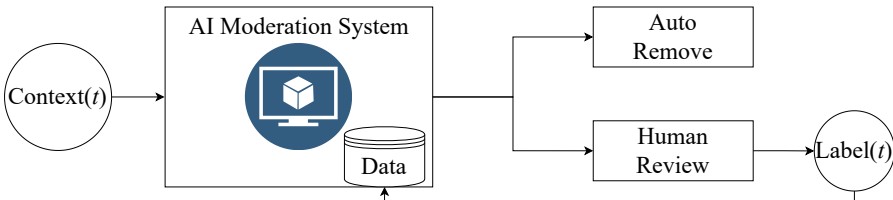

Figure 17: Real-time decision-making pipeline for content moderation. At each time $t$, the AI moderation system receives a post $x_t$, decides to **auto-remove** or **request human review**, then obtains feedback (if reviewed) to update its policy. This setup inherently involves uncertainty about *borderline* or *novel* content.

## G.2 Contextual Bandit Formulation

We model moderation as a **contextual bandit** [Wang et al., 2005, Langford and Zhang, 2007]:

- **Context** $x_t$: the textual (or multimedia) representation of the post at time $t$.
- **Action set** $\mathcal{A}_t = \{\text{auto-remove, human-review}\}$.
- **Reward** $r_t \in \mathbb{R}$: quantifying correctness vs. cost. For example:
  - $+1$ for correctly publishing benign content,
  - $-0.5$ for inadvertently publishing hateful content,
  - $+0.5$ for blocking any post (safer fallback, but potentially suboptimal if content was benign).

At each step, the agent chooses an action based on the context $\{x_1, \ldots, x_{t-1}\}$ and partial knowledge gained so far. Crucially, the agent must *explore* suspicious or uncertain contexts (requesting reviews) to learn from human labels, while *exploiting* confident predictions (auto-removing) to conserve human effort.

This *exploration-exploitation* trade-off, typical of contextual bandits, poses a significant challenge for large-scale moderation pipelines. As we show next, *foundation models alone* are not sufficient to address this adaptivity, motivating the need for an ensemble-based approach like *Ensemble++*.

### G.3 Challenges of Foundation Models in Online Decision-Making

Large foundation models (e.g. GPT series) have shown remarkable generalist capability but lack intrinsic **uncertainty modeling and adaptive exploration** [Krishnamurthy et al., 2024]. Indeed, even top-tier LLMs can fail in multi-armed bandit or contextual-bandit scenarios if not provided with explicit "memory" or "sampling" mechanisms [Krishnamurthy et al., 2024]. Hence, **foundation models alone** often struggle in large-scale, real-time moderation because:

- **Uncertainty Estimation.** Large models do not, by default, provide robust estimates of *how uncertain* they are on out-of-distribution content. As a result, they can incur high misclassification rates for novel forms of hate speech, rapidly changing memes, or new harassment tactics.

- **Incremental Adaptation at Scale.** The moderation stream is both continuous and high-volume. We need an approach that updates quickly (in near-constant or modest cost per step) to keep pace with new data, while preserving strong overall performance.

**In short**, to address *rare* or *emerging* forms of hateful content, a model must **actively explore** uncertain contexts and incorporate *human corrections* with minimal overhead. This is exactly where **Ensemble++** comes in.

### G.4 GPT-Ensemble++ for Content Moderation

We now introduce **GPT-Ensemble++**, adapting the *Ensemble++* agent (cf. Section 3.3) to text-based moderation scenarios with a foundation model backbone.

**1. LLM Feature Extractor.** We define $\phi(x; w)$, mapping a post $x$ into $\mathbb{R}^d$ using a GPT-2 (or Pythia14m) backbone, with $w$ either *frozen* or *partially finetuned*. This captures context and semantic cues.

**2. Ensemble++ Decision Head.** For each action $a$, we define a base parameter $b^a \in \mathbb{R}^d$ and an ensemble factor $\mathbf{A}^a \in \mathbb{R}^{d \times M}$. At each time step, we sample a random reference vector $\zeta \sim P_\zeta \subset \mathbb{R}^M$ (e.g. Gaussian). Then the *action-value* is:

$$f_\theta(x, \zeta)[a] = \langle \phi(x; w), b^a \rangle + \langle \mathrm{sg}[\phi(x; w)], \mathbf{A}^a \zeta \rangle.$$

Hence, the agent picks $\arg\max_a f_\theta(x, \zeta)[a]$. Drawing $\zeta$ each round fosters *randomized* (Thompson-like) exploration around uncertain or borderline posts.

**3. Incremental Updates.** If the system chooses `human-review` for a post $x_t$, we obtain a corrective label $y_t$ (hate vs. free) which implies a reward $r_t$. As describe in Algorithm 1, we then update $\theta = \{(b^a, \mathbf{A}^a), w\}$ using the *symmetrized* objective with bounded gradient steps. This step yields a fast, incremental refinement of the policy, allowing GPT-Ensemble++ to adapt quickly whenever new borderline cases arise in production.

### G.5 Experiments: Hate-Speech Detection

**Dataset and Setup.** We employ a hate-speech dataset[6] of about 135k posts, each assigned a continuous "hate" score. Thresholding at 0.5 yields "hate" vs. "free." At round $t$, the agent sees $x_t$ (text), chooses **publish** ($A_t = 1$) or **block** ($A_t = 2$), and receives:

$$\begin{cases} +1 & \text{if publishes a free post,} \\ -0.5 & \text{if publishes a hate post,} \\ +0.5 & \text{if blocks any post.} \end{cases}$$

---

[6]`https://huggingface.co/datasets/ucberkeley-dlab/measuring-hate-speech`

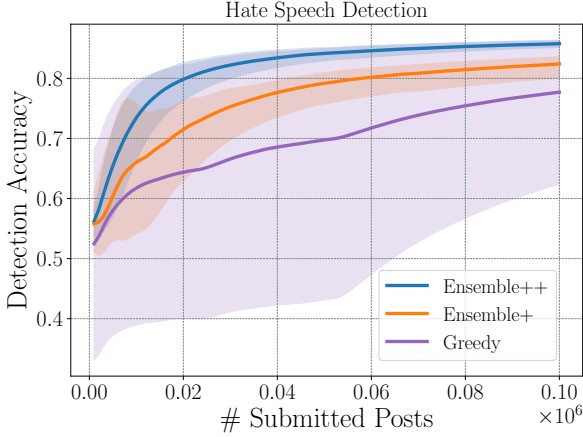

Figure 18: *Detection accuracy* over time in hateful vs. free moderation, averaged across random seeds, as the number of submitted posts increases. **Ensemble++** (blue) outperforms Greedy (purple) and Ensemble+ (orange) with lower variance.

We embed text with GPT-2 or Pythia14m in either frozen or partially finetuned mode, then feed into Ensemble++ or baselines.

**Comparative Baselines.** We consider:

1. **Greedy**: A single LLM-based classifier with no ensemble factor, i.e. $\mathbf{A}^a = 0$,

2. **Ensemble+** [Osband et al., 2018]: multiple ensemble heads jointly upated,

3. **Ensemble++** (ours): separated ensemble updates plus partial or full LLM finetuning.

We also vary **frozen vs. finetuned** embeddings $w$ for GPT-based models.

### G.5.1 Results and Analysis

**Uncertainty-Aware Gains.** Fig. 18 shows that *Ensemble++* significantly outperforms Greedy in cumulative reward, clarifying borderline expressions faster and reducing error variance.

**Frozen vs. Finetuned.** In Fig. 19, together with Ensemble++, full finetuning of GPT-based features yields further gains compared to frozen embeddings. This suggests that *active adaptation* of the LLM backbone is crucial for handling evolving content.

**Reduced Human Overhead.** Although not depicted, Ensemble++ quickly pinpoints which posts are certain vs. borderline, leading to $\sim 80\%$ fewer "human-review" actions after $10^4$ steps compared to naive or deterministic triggers (e.g., Greedy).

### G.6 Conclusions & Implications

In this chapter, we showed how **Ensemble++** can be integrated with *foundation models* like GPT-2 for large-scale content moderation—a domain rife with domain shifts, ambiguous inputs, and costly feedback. Our key findings:

- *Uncertainty quantification*: Ensemble++ better identifies borderline or novel forms of hate speech, enabling more selective human intervention.

- *Incremental adaptation*: The ensemble updates per step remain bounded, even with partial LLM finetuning.

- *Reduced moderator workload*: By focusing reviews on genuinely uncertain posts, Ensemble++ drastically cuts human oversight needs.

Overall, these results highlight *Ensemble++* as a powerful approach for real-time, uncertain tasks in industrial settings where foundation models alone lack the uncertainty-awareness needed for adaptive exploration.

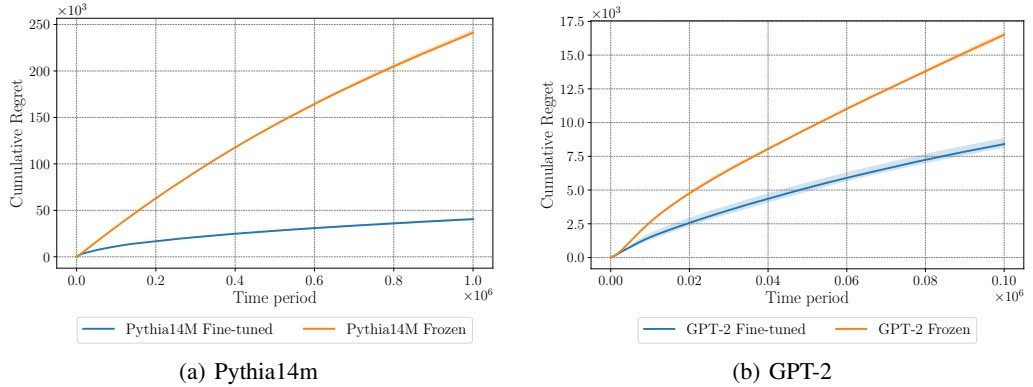

(a) Pythia14m  (b) GPT-2

Figure 19: Ablation in hateful-content moderation. (a,b) Fully finetuning yields stronger improvements in uncertain areas than freezing GPT (Pythia14m) backbone.

Potential future directions include:

- *Multi-modal moderation*: Extending Ensemble++ to handle multimedia content (e.g., images, videos) for more comprehensive moderation.
- *Adversarial robustness*: Investigating how Ensemble++ can adapt to adversarial attacks or adversarial examples in moderation tasks.
- *Scalable crowdsourcing*: Integrating Ensemble++ with scalable human-in-the-loop systems to further reduce human moderation costs.

