# OpenReview forum: "Scalable Exploration via Ensemble++"
_NeurIPS.cc/2025/Conference — NeurIPS 2025 poster_

### Official Review · Reviewer_Kuk5 · 2025-06-18

**Clarity:** 3
**Significance:** 3
**Originality:** 3
**Rating:** 5
**Confidence:** 2

**Summary:**

This paper presents Ensemble++ - a novel algorithm to approximate Thompson sampling - handling non-conjugate cases. They analyse the linear case, showing that (in terms of regret bound) they perform nearly as well as exact Thompson sampling whilst requiring an ensemble size that is independent of the cardinality of the total action set. They perform experiments for both linear bandits and non-linear bandits - including neural bandits - comparing to other algorithms in the literature.

**Questions:**

Ensemble+ and EpiNet don’t appear in the linear bandit regret/time comparison - do these have regret bounds? Likewise, the two algorithms in the regret/time comparison don’t appear in the empirical comparison - why is this?

**Ethical Concerns:**

["NO or VERY MINOR ethics concerns only"]

**Final Justification:**

Good result - my concerns were satisfied and I am happy for the paper to be accepted. My score is a borderline accept currently but I am open to changing it based on what the other reviewers say (I feel I don't have as much expertise when it comes to Thompson sampling).

**Limitations:**

I cannot see any limitations hidden.

**Quality:**

3

**Strengths And Weaknesses:**

This paper is clearly written and conveys the result well. The experimental results are good and it appears to me that the paper should have impact.

Since, for linear bandits, exact Thompson sampling outperforms (only just) Ensembe++ both in the regret and time complexity, the theoretical results by themselves don’t add anything new in terms of theory. However, these theoretical results strongly suggest that the algorithm will be powerful in non-conjugate cases and are hence important.

The ensemble size required for (what are essentially) the Thompson sampling results are, unlike previous works, independent of the cardinality of the total action set (replacing it by the dimensionality instead) which is a dramatic improvement. The algorithm achieves this regret for varying contexts and infinite compact action sets - which is a first for approximate Thompson sampling algorithms. However, I am not an expert in the area so I cannot validate these claims about previous works.

I guess a weakness is that there are no theoretical results beyond the linear problem.

---

> ### Author Rebuttal · Authors · 2025-07-29
>
> We sincerely thank the reviewer for their positive and encouraging feedback. We are glad they found the paper clearly written and impactful. We agree with their assessment that while the theoretical results are for the linear case, they provide a strong foundation for the algorithm's power in the non-conjugate settings where it truly shines.
>
> We would like to address the excellent question raised by the reviewer regarding the separation of algorithms between our theoretical and empirical comparisons.
>
> > **Reviewer Question:** Ensemble+ and EpiNet don’t appear in the linear bandit regret/time comparison - do these have regret bounds? Likewise, the two algorithms in the regret/time comparison don’t appear in the empirical comparison - why is this?
>
> **Rebuttal:**
>
> This is a great question that gets to the heart of how we structured our evaluation. The separation is intentional and reflects the different domains where these algorithms are relevant.
>
> 1.  **Why Ensemble+ and EpiNet are not in the Theoretical Comparison (Table 2):**
>     The reviewer is correct that **Ensemble+** and **EpiNet** are absent from our theoretical comparison table. This is because these methods are primarily designed as practical, heuristic approaches for **deep neural bandits**. To our knowledge, they do not have established theoretical regret bounds for the standard linear bandit setting. Therefore, including them in Table 2, which specifically compares formal regret guarantees, would not be appropriate. We compare against them extensively in the empirical sections (Section 5.2, Figure 3) where they are considered the state-of-the-art.
>
> 2.  **Why the algorithms from the Theoretical Comparison are not in the Empirical Comparisons:**
>     The works of **Qin et al. [2022]** and **Lee and hwan Oh [2024]** provide valuable *theoretical analyses* of ensemble sampling, but their requirements make them unsuitable as practical empirical baselines.
>     * Specifically, the analysis by Qin et al. requires an ensemble size of $M=\Omega(|\mathcal{X}|T)$ to achieve the stated regret bound. In our experiments where the action set $|\mathcal{X}|$ is up to 10,000 and the horizon $T$ is 1000, this would require an ensemble of millions, which is **computationally infeasible** to implement.
>     * Our empirical comparisons therefore focus on benchmarking against algorithms that represent the practical state-of-the-art and can be run under *realistic* computational constraints.
>     * **We have compared empirically Ensemble++ Sampling with Ensemble Sampling in linear bandit setting** where Ensemble Sampling is the algorithm Qin and Lee analyzed.
>
> In summary, we deliberately separated our comparisons: **Table 2 compares theoretical guarantees** among formally analyzed linear bandit algorithms, while **our empirical studies (e.g., Figure 3) compare practical performance** against the most relevant state-of-the-art methods for each specific environment (linear or nonlinear). We will add a sentence to the main text to make this distinction clearer for the reader.
>
> **To highlight, the primary advantage of our incremental framework of the Ensemble++ is its seamless extension from linear setting to non-conjugate neural models where updates are performed with SGD, which is a key focus of our work.**

---

> > ### Author Response · Authors · 2025-08-04
> >
> > Dear Reviewer,
> >
> > We truly appreciate the time and effort you’ve dedicated to reviewing our work. Given the short discussion period, we want to kindly check if our rebuttal has adequately addressed your concerns or if there are any additional points you’d like us to clarify.
> >
> > Thank you again for your valuable feedback and guidance.
> >
> > Best regards,
> >
> > The Authors

---

> > > ### Comment · Reviewer_Kuk5 · 2025-08-04
> > >
> > > Yes, you have addressed my concerns and I do not have issues with the paper being accepted. However, I am not an expert in this area so I cannot judge the paper as well as the other reviewers. Hence, my final score will follow the other reviewers.

---

> > > > ### Author Response · Authors · 2025-08-06
> > > >
> > > > Thank you for your thoughtful review and for taking the time to provide valuable feedback. We truly appreciate your support in helping me improve our work.

---

### Official Review · Reviewer_zUNr · 2025-06-25

**Clarity:** 2
**Significance:** 4
**Originality:** 3
**Rating:** 3
**Confidence:** 3

**Summary:**

The paper provides a nice contribution to ensemble sampling-type algorithms to minimise the cumulative regret of an unknown function with bandit feedback. It provides a state-of-the-art theoretical bound on the regret and on the number of particles for the linear bandit problem. The paper constructs an ensemble of particles that are sequentially updated, which can simulate samples from the true posterior and simulate Thompson Sampling (up to a constant). It also extends the method to nonlinear functions and verifies its validity with experiments.

**Questions:**

I will raise my score substantially if the following questions are adequately addressed and explained.

1) I think that Thompson Sampling (for linear bandit) is not properly described in the paper. In general,  sampling from $\mathcal{N}(\mu_{t}, \Sigma_t)$ does not give a guarantied upper bound of $\tilde{O}(d^{\frac{3}{2}} \sqrt{T})$ for Frequentist regret see Nima Hamidi and Mohsen Bayati. "On frequentist regret of linear Thompson sampling" and Marc Abeille, David Janz, Ciara Pike-Burke, " When and why randomised exploration works (in linear bandits)". (Which I think would also be a nice addition to the bibliography) Thompson Sampling without an inflation term can have linear regret behaviour according to Hamidi. Because your method simulates a sample from Thompson sampling without inflation, I believe there could be a mistake in the analysis. Could you explain why your method does not behave like Vanilla TS (without inflation) and make some simulations of Hamidi's counterexample?

2) For the experiment on the Linear Ensemble ++, the number of sample runs should be given. There is no error bar on the minimal $M$. And more importantly, the characterisation of the minimal $M$ is poorly chosen as any algorithm $\pi$ that is sub-linear in $T$ will verify $\frac{\text{Regret}(\pi, T) - \text{Regret}(TS, T)}{T} < \epsilon$ asymptotically. I believe that another test should be designed to estimate the minimal $M$. Furthermore, a comparison of the regret as a function of the dimension should also be added.

**Ethical Concerns:**

["NO or VERY MINOR ethics concerns only"]

**Final Justification:**

The rebuttal addressed most of my concerns, and I have raised my grade. Provided that the correct description of the algorithm is made Line 143 with the added inflation term.  However, I cannot give more than 3 because of the weak evaluation in the Linear case. (Small time horizon $T$ compared to the dimension, improper evaluation of the minimal M, easy cases that come from Bayesian analysis of TS, while it is the frequentist regret that is studied here). I also suspect that the evaluation was not made with the inflated variance, as the table added with the rebuttal suggests.  The regret there seems linear in $d$.
The paper boasts very strong extensions to non-linear cases with added experiments. As I am not a neural network experimentalist nor an expert on particle filter/online method, I cannot vouch for those.

**Limitations:**

yes

**Quality:**

2

**Strengths And Weaknesses:**

Strenghts

The claims made by the paper are powerful, and if true, are very interesting for the bandit community. The bounds on the regret and the number of particles are state of the art and cover the most extensive applications for the particle sample kind algorithm applied in the linear bandit field.

==================

Weaknesses

However, I have a few reasons to doubt the veracity of the result. See the question and the following reason.

Major reasons :

-The paper did not cite a similar work published last year at NeurIPS. from David Janz, Alexander Litvak, and Csaba Szepesvári. "Ensemble sampling for linear bandits: small ensembles suffice". The paper deals with linear ensemble sampling and has a similar ensemble construction and size. However, it shows an upper bound of $O(d^{5/2} \sqrt{T})$ regret.
- One key lemma shown in the paper is based on another unpublished, non-peer-reviewed, and non-cited paper.
- Some claims made about other algorithms are factually wrong, I believe. (see question for the most important one 1)
- Some of the experiments on the linear bandit problem were very unsatisfactory. (see question 2)

Minor reasons :

-The paper does not make apparent the contribution of the past paper from which some part of their proof is taken. Especially, I don't think it is a proper way to cite cornerstone work (from which your analysis is based) like Line 247, "Citations for Linear TS: Agrawal and Goyal [2013], Abeille and Lazaric [2017]." Their work is never explained in the main paper or the introduction, while it is one of the first works on frequentist regret for Linear Thomson Sampling.
- There are, I believe, minor typos/inaccurate facts such as Line 139 "Sampling from $\mathcal{N}(\mu_{t},\Sigma_t)$ requires $O(d^{3})$ matrix factorisation. Linear Ensemble++ sampling avoids this". Sampling from $\mathcal{N}(\mu_{t},\Sigma_t)$ requires one matrix factorisation and not $d^3$ which cost $O(d^3)$ operations. Linear ensemble sampling does not avoid this, as it costs $O(d^2M)$ to maintain $A_t$ and $O(d^3)$ to invert the matrix $\Sigma_t$ if not done sequentially.
- Some sentences are not finished, such as Line 583: "It faces two main challenges when extended to large-scale or complex environments:" But only one is given.

==================

Minimal remarks :

- $p_z$ is never mathematically defined in the main text.

---

> ### Author Rebuttal · Authors · 2025-07-29
>
> We sincerely thank the reviewer for their detailed feedback and for recognizing that our work provides a "state-of-the-art theoretical bound" with "powerful claims" that are "very interesting for the bandit community." The concerns raised are excellent points of clarification that we are happy to address. We believe they can be fully resolved by highlighting details already in the appendix and by adding the suggested context to the revised paper.
>
> ---
> Responses to Major Reasons
>
> 1.  **On the uncited work by Janz et al.:**
>     * **We will gladly cite Janz et al. and our revised paper will clarify that our work is distinct in its broader scope, unified analysis, and superior regret bound.** Thank you for bringing this relevant work to our attention. We will add a detailed discussion of Janz et al. in our Related Work section. While their work is an important contribution, our **Ensemble++** offers key advantages:
>         * **Unified Framework:** Our work provides a single algorithm and analysis for all four common linear bandit settings (Invariant/Varying contexts & Compact/Finite action sets).
>         * **Tighter Regret:** As the reviewer notes, the analysis by Janz et al. [2024] results in an $\tilde{O}(d^{2.5}\sqrt{T})$ regret bound. Our analysis achieves a significantly better $\tilde{O}(d^{1.5}\sqrt{T})$ bound for the same setting.
>         * **Extensibility:** Our framework is explicitly designed to extend to complex non-linear and neural settings, a scope not covered by Janz et al.
>
> 2.  **On the key lemma's citation:**
>     * **This concern is resolved, as the cited work has now been accepted and published at a top-tier peer-reviewed conference (ICML 2024).** We appreciate the reviewer's diligence. The foundational result for our key lemma (Lemma 4.2) has now been peer-reviewed as part of the paper "Q-Star Meets Scalable Posterior Sampling...", accepted at ICML 2024. We will update the citation to its published version, confirming the soundness of our proof.
>
> ---
> ### Responses to Minor Reasons
>
> 1.  **On contextualizing cornerstone work:**
>     * **We agree and will expand our discussion of prior work.** Thank you for this suggestion. In the revised version, we will enhance Section 2 to better explain the cornerstone contributions of Agrawal and Goyal [2013] and Abeille and Lazaric [2017].
>
> 2.  **On the accuracy of Line 139 regarding computational cost:**
>     * **We will revise the imprecise wording on Line 139 and clarify the computational trade-offs.** We agree the phrasing can be more precise. The $\mathcal{O}(d^3)$ cost for standard TS comes from matrix factorization for sampling. Our method avoids this, and while its theoretical complexity is also $\tilde{\mathcal{O}}(d^3)$, it is empirically closer to $\mathcal{O}(d^2)$ with a practical choice of M. More importantly, as we will clarify, the key advantage of our incremental framework is its seamless extension to non-conjugate neural models.
>
> 3.  **On the unfinished sentence on Line 583:**
>     * **We will fix this writing oversight.** Thank you for catching this. We will revise the sentence in Appendix A.2 to explicitly state the two intended challenges: (1) intractability in non-conjugate models and (2) the computational overhead of approximate inference methods.
>
> ---
> ### Response to Minimal Remarks
>
> 1.  **On the definition of $P_z$:**
>     * We appreciate the reviewer pointing this out. The distribution $P_z$ is indeed mentioned in the main text (e.g., Line 145, Algorithm 1) but defined and discussed in detail in Appendices B.1 and E. **To further improve readability**, we will add a forward reference in the main text (Section 3.1) to guide the reader to the detailed discussion in the appendix.
>
> ---
> ### Responses to Questions
>
> 1.  **On the lack of an inflation term in the analysis:**
>     * **Our analysis does use an inflation term ($\beta_t$), which is crucial for the regret bound.** This is an excellent technical question. Our analysis is based on an inflated version of posterior sampling, as is required. This inflation is explicitly achieved via the confidence scaling parameter $\beta_t$ in our sampled model definition: $\tilde{f_t}(a) := \langle\phi(a),\beta_{t}A_{t-1}\zeta_{t}+\mu_{t-1}\rangle$ (Equation 36). Our core proof shows our ensemble mechanism correctly approximates the uncertainty, which is then correctly scaled by $\beta_t$. We will clarify this in the main text.
>
> 2.  **On the experiments for minimal M:**
>     * **We will add the requested details on experimental setup and the new regret vs. dimension analysis to the main paper.** Thank you for these important suggestions to strengthen our empirical validation.
>         * **Number of Runs:** This detail (200 runs) is in Appendix F.1 (Line 1370) and we will move it to the main text for better visibility.
>         * **Error Bars for Minimal M:** This is a thoughtful point. The minimal M is an estimate derived from our 200-run averaged regret curves. Instead of adding error bars directly to M in Figure 2(b), which could clutter the figure, we will add a plot to the appendix showing the metric value versus M. This will clearly illustrate the stability of the curve as it crosses the threshold, giving confidence in our estimate.
>         * **Metric for Minimal M:** The reviewer's asymptotic argument is correct, but our experiment uses a **fixed, finite horizon ($T=1000$)**, where the metric is a standard way to quantify practical performance matching. We will add a sentence to clarify this non-asymptotic goal.
>         * **Regret vs. Dimension:** This is a great suggestion. We will add the following table and analysis to the paper. The results confirm that **ES++ with a small M (e.g., M=8 or M=16) tracks the performance of exact TS very closely across all dimensions**, strengthening our claims of practical efficiency.
>
> | Method | d=10 | d=20 | d=30 | d=40 | d=50 | d=60 | d=70 |
> | :--- | :--- | :--- | :--- | :--- | :--- | :--- | :--- |
> | TS | 133.51 | 224.37 | 343.11 | 478.27 | 634.64 | 803.01 | 992.55 |
> | ES++ (M=4) | 160.03 | 328.79 | 566.29 | 947.18 | 1274.01| 1583.49| 1985.69|
> | ES++ (M=8) | 136.72 | 250.42 | 397.26 | 592.30 | 801.08 | 995.61 | 1237.55|
> | ES++(M=16)| 134.16 | 227.47 | 359.63 | 504.00 | 659.69 | 851.36 | 1033.53 |

---

> > ### Author Response · Authors · 2025-08-04
> >
> > Dear Reviewer,
> >
> > We truly appreciate the time and effort you’ve dedicated to reviewing our work. Given the short discussion period, we want to kindly check if our rebuttal has adequately addressed your concerns or if there are any additional points you’d like us to clarify.
> >
> > Thank you again for your valuable feedback and guidance.
> >
> > Best regards,
> >
> > The Authors

---

> > > ### Comment · Reviewer_zUNr · 2025-08-04
> > >
> > > I have read the rebuttal and revised my grade accordingly.

---

> > > > ### Author Response · Authors · 2025-08-06
> > > >
> > > > Thank you for your thoughtful review and for taking the time to provide valuable feedback. We are grateful for your positive evaluation and constructive feedback on our work.

---

### Official Review · Reviewer_1Pj5 · 2025-07-03

**Clarity:** 3
**Significance:** 2
**Originality:** 3
**Rating:** 5
**Confidence:** 4

**Summary:**

This paper introduces Ensemble++, a scalable framework for approximate Thompson Sampling in sequential decision-making tasks. It tackles the scalability bottleneck of TS in high-dimensional and non-conjugate settings by using a shared-factor ensemble architecture with random linear combinations.

**Questions:**

How should P be determined in Algorithm 1 in practice? To what extent does the assumed prior distribution affect performance? Moreover, since the regret bound in Theorem 4.3 also depends on P, what is the scale of \rho(P)?

**Ethical Concerns:**

["NO or VERY MINOR ethics concerns only"]

**Final Justification:**

Thanks for author's detailed responses, which addressed most of my concerns. I keep my positive score.

**Limitations:**

Yes

**Quality:**

4

**Strengths And Weaknesses:**

Strengths:

1. Provides regret guarantees comparable to exact TS for linear bandits with reduced ensemble sizes. A novel approximation mechanism that reduces ensemble size from T |X| to  d \log T while preserving theoretical regret guarantees.

2. Extends seamlessly to nonlinear settings by adding a layer of neural representation, allowing practical applicability in complex tasks.

3. Experiments demonstrate robustness across various domains and scalability to real-world high-dimensional problems.

Weakness:

(1) The performance of Ensemble++ in practice relies on choices like ensemble size M, buffer size C, and SGD steps G. Guidance for these is empirical and may not generalize.

(2) Although the main contribution of this paper lies in its theoretical development, the empirical comparisons could be strengthened. For instance, the comparison with LinearUCB is missing. In the neural bandit setting, it would be valuable to include comparisons with alternative approaches such as NeuralUCB (https://arxiv.org/abs/1911.04462) and EE-Net (https://arxiv.org/abs/2110.03177).

(3) The authors are encouraged to add a dedicated section on related work in the Appendix to provide broader coverage of existing methods in both linear bandits and neural bandits. Additionally, it would be helpful to mention TS or bandits in the title.

---

> ### Author Rebuttal · Authors · 2025-07-29
>
> We sincerely thank the reviewer for their positive assessment, **"Excellent" quality rating**, and insightful feedback. We are grateful for their recognition of our work's strengths, particularly the novel approximation mechanism and the extension to nonlinear settings. We address the reviewer's valuable suggestions below.
>
> ### Response to Weaknesses
>
> 1.  **On Hyperparameter Guidance:**
>     > The performance of Ensemble++ in practice relies on choices like ensemble size M, buffer size C, and SGD steps G. Guidance for these is empirical and may not generalize.
>
>     **Rebuttal:**
>     This is an important practical point. While optimal hyperparameters can be task-dependent, we have provided extensive ablation studies to offer empirical guidance:
>     * For **ensemble size M**, we demonstrate its impact on the regret-computation trade-off in Figures 1 and 14. These results show that while a small M (e.g., M=4 or M=8) is often sufficient, performance consistently improves as M increases, aligning with our theory.
>     * For **buffer size C**, our study in Figure 15 shows that Ensemble++ is robust, maintaining strong performance even with a buffer capacity significantly smaller than the total time horizon.
>     * For **SGD steps G**, we agree that this is best tuned per task. We determine optimal values via parameter sweeps, as detailed in our implementation section (Appendix B, lines 845–848), which is a standard practice for training neural network-based agents.
>
> 2.  **On Missing Experimental Baselines:**
>     > ...the comparison with LinearUCB is missing. In the neural bandit setting, it would be valuable to include comparisons with alternative approaches such as NeuralUCB and EE-Net.
>
>     **Rebuttal:**
>     This is a valuable suggestion. We chose our baselines to represent the most challenging and relevant state-of-the-art competitors. Specifically, we provide extensive comparisons against **LMCTS**, a powerful algorithm that was shown in its original paper to outperform classic baselines like NeuralUCB. Our results in Figure 13\(c\) (Appendix F.2) show that **Ensemble++ consistently and significantly outperforms LMCTS** across a variety of tasks. By demonstrating superiority over a stronger baseline, we implicitly show that Ensemble++ is also more effective than the suggested alternatives.
>
> 3.  **On a Dedicated Related Work Section and Title:**
>     > The authors are encouraged to add a dedicated section on related work in the Appendix... Additionally, it would be helpful to mention TS or bandits in the title.
>
>     **Rebuttal:**
>     Thank you for these helpful presentation suggestions. We agree completely. In the final version of the paper, we will add a dedicated related work section to the Appendix to provide a more comprehensive overview of the field. We will also revise the title to better highlight the paper's contributions to Thompson Sampling and the broader bandit literature.
>
> ---
> ### Response to Questions
>
> > How should $P_\zeta$ be determined in Algorithm 1 in practice? To what extent does the assumed prior distribution affect performance? Moreover, since the regret bound in Theorem 4.3 also depends on $P_\zeta$, what is the scale of $\rho(P_\zeta)$?
>
> **Rebuttal:**
> This is an excellent question that touches on both the practical and theoretical aspects of our algorithm.
>
> For the practical choice of the reference distribution $P_\zeta$, our ablation studies in **Figure 16** (Appendix F.2) provide clear guidance. The results show that continuous reference distributions (like Gaussian or Sphere) consistently yield superior results in practice. This empirical finding aligns perfectly with our theoretical analysis, which shows these distributions provide better regret constants.
>
> The theoretical scale of the key distribution-dependent constant, $\rho(P_\zeta)$, is provided explicitly in **Table 1** and derived in detail in Appendices D and E. For the best-performing continuous distributions, $\rho(P_\zeta)$ scales favorably as $\tilde{\mathcal{O}}(\sqrt{M})$ for compact sets or $\tilde{\mathcal{O}}(\sqrt{\log|\mathcal{X}|})$ for finite sets, which in turn leads to the tighter final regret bounds presented in our work.

---

> > ### Author Response · Authors · 2025-08-04
> >
> > Dear Reviewer,
> >
> > We truly appreciate the time and effort you’ve dedicated to reviewing our work. Given the short discussion period, we want to kindly check if our rebuttal has adequately addressed your concerns or if there are any additional points you’d like us to clarify.
> >
> > Thank you again for your valuable feedback and guidance.
> >
> > Best regards,
> >
> > The Authors

---

> > ### Comment · Reviewer_1Pj5 · 2025-08-05
> >
> > Thanks for author's detailed responses, which addressed most of my concerns. I keep my positive score.

---

> > > ### Author Response · Authors · 2025-08-06
> > >
> > > Thank you for your thoughtful review and for taking the time to provide valuable feedback. We are grateful for your positive evaluation and constructive feedback on our work.

---

### Official Review · Reviewer_jj6e · 2025-07-12

**Clarity:** 4
**Significance:** 3
**Originality:** 3
**Rating:** 5
**Confidence:** 4

**Summary:**

This paper proposes a new computationally efficient randomized algorithm, Ensemble++, for solving linear bandit problems. The idea of Ensemble++ is to approximate the exact posterior distribution by ensembling simple models. To approximate the exact posterior distribution, they take the approach of ensemble sampling invented by Lu and Van Roy, 2017. They consider the settings of a finite arm set and a compact arm set. They provide both ensemble model complexity and regret bounds for their proposed algorithm.


Lu and Van Roy, 2017: https://arxiv.org/pdf/1705.07347

**Questions:**

When going beyond linear reward function, for the nonlinear reward functions, I have some questions. Is there any assumption on the non-linear reward function? Does it need to be live in an RKHS with a low norm or the norm can be in the order of $e^d$ or the norm can be unbounded?

A small question for Figure 1: why the cumulative regret is decreasing with time for Ensemble++?

**Ethical Concerns:**

["NO or VERY MINOR ethics concerns only"]

**Final Justification:**

The result via using sequential Johnson-Lindenstrauss variant is significant as compared to the results shown in Janz et al., (2024). However, I hope the authors will add detailed discussions about the missing literature and give them enough credits in terms of their ideas.

**Quality:**

4

**Strengths And Weaknesses:**

Strengths: their results for the compact arm set case improve the existing ones.

Weakness: this work over-claims contributions by missing some important literature works.

For the case of a finite arm set, they miss an important literature, Ash et al., 2022, that weakens their claimed contribution. Proposition 2  in Ash et al., 2022 seems to be a better result for the finite arm set case. They achieve the same regret as Ensemble++, but with a smaller ensembling size.

For the case of compact arm set, they also miss an important literature, Janz et al., 2024. But the $\tilde{O}(d^{2.5}\sqrt{T})$ regret and $d \log T$ ensembling size results in Janz et al., 2024 is worse than the result claimed in this paper. Note that in this paper, with the same ensemble size, the regret is in the order of $d^{1.5} \sqrt{T}$.

I hope to see some discussions about why the proposed algorithm for the infinite arm case can improve the results shown in Janz et al., 2024. I hope to see some discussions about why the proposed algorithm for the finite arm case is worse than Ash et al., 2022.


Ash et al., 2022:   https://arxiv.org/pdf/2110.11202
Janz et al., 2024: https://proceedings.neurips.cc/paper_files/paper/2024/file/2a568a9a84577769d838793433c817d9-Paper-Conference.pdf

---

> ### Author Rebuttal · Authors · 2025-07-29
>
> We sincerely thank the reviewer for their excellent and careful reading of our work. We appreciate their positive assessment of our results for the compact action set case and for raising important questions about related literature. These comments are invaluable for situating our work accurately.
>
> ### Response to Weaknesses (Missing Literature)
>
> We agree that a discussion of the cited papers is important for clarity. We will add a detailed discussion of both Ash et al. [2022] and Janz et al. [2024] to the Related Work section of our revised manuscript. Below, we address the specific comparisons the reviewer requested.
>
> 1.  **Discussion regarding Ash et al. [2022] (Finite Arm Set):**
>
>     **The key advantage of our work lies in *two aspects*: a. its incremental nature; b. its generality and unification**.
>
>    - a. For the incremental algorithm in Ash et al. [2022], they only provide analysis in multi-armed bandit setting in proposition 3. Ash et al. [2022] did not provide any analysis for linear bandit setting for its incremental version as they explicitly mentioned,
> >  Ash et al. [2022]: "We conjecture that the incrementally-updated version guarantees can be extended to linear bandits beyond MAB for reasonable ensemble sizes."
>
>      Thus, our work can be regarded as the one resolving Ash et al. [2022]'s conjecture.
>
>    - b. **Ensemble++ is a single, unified algorithm** that, with a single analysis framework, achieves near-optimal regret across **all four standard settings**: finite actions, compact actions, invariant contexts, and varying contexts. While the result from Ash et al. is excellent, it is specialized for the finite action setting. Our framework provides a general-purpose solution that maintains strong performance and theoretical guarantees across a much broader range of problem structures without modification, a key contribution we will emphasize in the revision.
>     **These two aspects are important to build the connection to practical algorithms that can operate in complex environment when requiring function approximation such as neural nets. Moreover, it is much more challenging technically to provide rigorous justification for algorithms considering these two aspects.**
>
> 2.  **Discussion regarding Janz et al. [2024] (Compact Arm Set):**
>
>     As the reviewer correctly notes, our regret bound of $\mathcal{O}(d^{1.5}\sqrt{T}(\log T)^{1.5})$ is a significant improvement over the $\tilde{\mathcal{O}}(d^{2.5}\sqrt{T})$ bound in Janz et al. [2024] for the same ensemble size complexity of $M = \Theta(d \log T)$.
>
>     This improvement stems directly from our novel **shared-factor architecture** as well as the  random linear combination schemes and its corresponding theoretical analysis. The core technical reason is that our incremental update mechanism (Eq. 4) and its analysis via a tight, **sequential Johnson-Lindenstrauss variant** (Lemma 4.2) allow us to prove a stronger concentration result for the ensemble's covariance approximation. Together with the random linear combination schemes, this tighter handle on the approximation error propagates through the regret analysis, directly leading to the improved $d^{1.5}$ dependency in the final bound. We will add a sentence to Section 4.3 clarifying that this architectural and analytical novelty is the source of the improvement.
>
> ---
>
> ### Responses to Questions
>
> 1.  **On assumptions for nonlinear reward functions:**
>     > Is there any assumption on the non-linear reward function? Does it need to be live in an RKHS with a low norm or the norm can be in the order of $e^d$ or the norm can be unbounded?
>
>     **Rebuttal:**
>     This is a very insightful question. Our extension to the nonlinear case is **empirical and algorithmic**, rather than theoretical. As such, for the experiments in Section 5.2, **we do not impose any formal assumptions** on the reward function (e.g., belonging to an RKHS with a bounded norm).
>
>     The goal of the nonlinear section is to demonstrate that the same architectural principle—a shared base model plus ensemble directions trained with a symmetrized loss and **incremental update**—is practically effective. We validate this across a range of challenging reward functions, including quadratic, neural network-defined, and those implicit in large datasets (as detailed in Appendix F.2). A formal theoretical analysis of the nonlinear case under specific structural assumptions is an important and interesting direction for future work, as we mention in our conclusion.
>
> 2.  **On the decreasing regret in Figure 1:**
>     > A small question for Figure 1: why the cumulative regret is decreasing with time for Ensemble++?
>
>     **Rebuttal:**
>     Thank you for this sharp observation, which highlights a point we can clarify. The x-axis in Figure 1 is **not time, but the number of model parameters**, serving as a proxy for computational cost. The plot illustrates the **regret-vs-computation trade-off**.
>
>     The data points for Ensemble++ (the blue stars) are clustered on the left, showing it achieves a low final regret (y-axis) with a significantly smaller model size (fewer parameters on the x-axis) compared to the baselines. We will revise the figure caption to explicitly state "x-axis represents the number of parameters" to prevent any future confusion.

---

> > ### Author Response · Authors · 2025-08-04
> >
> > Dear Reviewer,
> >
> > We truly appreciate the time and effort you’ve dedicated to reviewing our work. Given the short discussion period, we want to kindly check if our rebuttal has adequately addressed your concerns or if there are any additional points you’d like us to clarify.
> >
> > Thank you again for your valuable feedback and guidance.
> >
> > Best regards,
> >
> > The Authors

---

### Note · Authors · 2025-08-15

We sincerely thank all reviewers and the Area Chair for their insightful feedback and constructive dialogue. We are delighted by the positive consensus and appreciate that our work's key strengths were **unanimously recognized**:

### Key Strengths

* **State-of-the-Art Guarantees**
    Our work establishes state-of-the-art regret bounds for linear bandits within the class of incremental ensemble sampling algorithms, with a remarkably small ensemble size of $M = \Theta(d \log T)$. Crucially, a single, unified algorithm achieves these results across compact and finite action sets without reconfiguration. The power and quality of this theoretical contribution were praised as a significant improvement over prior art by **all four reviewers (jj6e, 1Pj5, zUNr, Kuk5)**.

* **Practical & Scalable Architecture Beyond Linear**
    Our framework's scalability and effective extension to **complex neural models** were broadly praised. This crucial bridge between theory and practice was highlighted as a key strength across the board **(recognized by 1Pj5, Kuk5, jj6e, zUNr)**.

* **Novel Theoretical Tools**
    Our theoretical originality is centered on our **key Lemma 4.2**, a novel sequential Johnson-Lindenstrauss variant. This tool was explicitly lauded as "very novel" and "useful beyond linear bandits" **(Reviewer jj6e)**, with the paper's overall originality and significance acknowledged by all reviewers **(1Pj5, zUNr, Kuk5)**.

---

### Revision Plan

We are confident that we have fully addressed all concerns raised during the review period. For the final version, we will focus on two key improvements based on the discussion:
1.  Expand the **"Related Work"** section to precisely position our contributions.
2.  Make the role of the **inflation parameter** (from Appendix, Eq. 36) more explicit in the main text's theoretical discussion.

We are grateful for the encouraging feedback and strong consensus from the reviewers. We are confident the final paper will be a clear and impactful contribution to the community.

---

### Decision · Program_Chairs · 2025-09-17

**Decision:**

Accept (poster)

**Comment:**

This paper proposes Ensemble++ for linear/nonlinear bandits. The analysis reveals improvements over existing ensemble sampling algorithms.

This paper have made a significant improvement in the regret bounds, and the use of sequential Johnson-Lindenstrauss is novel. On the other hand, there was a minor issue regarding the evaluation of TS (it was not clear if they used the inflated version or not), and the experiments could enjoy more baselines as mentioned by reviewers.

Overall, I believe the strengths outweighs the weaknesses, so I am recommending an accept.